# Guaranteeing Privacy in Hybrid Quantum Learning through Theoretical Mechanisms

## Abstract

Quantum Machine Learning (QML) is becoming increasingly prevalent due to its potential to enhance classical machine learning (ML) tasks, such as classification. Although quantum noise is often viewed as a major challenge in quantum computing, it also offers a unique opportunity to enhance privacy. In particular, intrinsic quantum noise provides a natural stochastic resource that, when rigorously analyzed within the differential privacy (DP) framework and composed with classical mechanisms, can satisfy formal $(\varepsilon, \delta)$-DP guarantees. This enables a reduction in the required classical perturbation without compromising the privacy budget, potentially improving model utility. However, the integration of classical and quantum noise for privacy preservation remains unexplored. In this work, we propose a hybrid noise-added mechanism, HYPER-Q, that combines classical and quantum noise to protect the privacy of QML models. We provide a comprehensive analysis of its privacy guarantees and establish theoretical bounds on its utility. Empirically, we demonstrate that HYPER-Q outperforms existing classical noise-based mechanisms in terms of adversarial robustness across multiple real-world datasets.

## 1 Introduction

Quantum Machine Learning (QML) has emerged as a compelling paradigm that integrates the computational advantages of quantum systems with the modeling power of machine learning (ML). A fundamental feature of quantum systems is quantum noise, the inherent randomness and decoherence that arise due to interactions with the environment. Although quantum noise is typically considered to be a barrier to achieving fault-tolerant quantum computing, it provides an opportunity to serve as a natural and intrinsic source of randomness for privacy-preservation.

In classical ML, Differential Privacy (DP) (Dwork, 2006) has become the standard framework for providing formal privacy guarantees. DP ensures that the output of an algorithm does not change significantly when a single individual's data is added or removed from the input dataset, thereby protecting individual privacy. Beyond its role in privacy preservation, DP has also been extended to certify the robustness of ML models against adversarial attacks (Lecuyer et al., 2019; Cohen et al., 2019). Privacy in DP is typically achieved by injecting carefully calibrated random noise, such as Gaussian or Laplacian, into the learning process (Geng & Viswanath, 2012; Balle & Wang, 2018; Ji & Li, 2024). Furthermore, the overall privacy guarantee can be amplified through additional stochastic techniques such as subsampling (Balle et al., 2018), iterative composition (Feldman et al., 2018), and diffusion-based mechanisms (Balle et al., 2019a). Nevertheless, theoretical privacy amplification is not guaranteed under arbitrary combinations of stochastic techniques.

Recent studies extend the notion of DP to the quantum domain, leading to Quantum Differential Privacy (QDP) (Du et al., 2021b; Hirche et al., 2023). However, several key challenges remain unaddressed. First, existing efforts primarily focus on defining privacy guarantees for quantum data. However, most practical, near-term QML applications are hybrid models that operate on classical data and use the quantum circuit only as an intermediate processing component. This hybrid architecture presents a critical privacy challenge: a DP guarantee applied only to the intermediate quantum layer does not ensure end-to-end privacy for the full model, especially if the preceding classical components are sensitive. Second, the interaction between classical noise (e.g., Gaussian, Laplacian) and intrinsic quantum noise has not yet been investigated. This research gap is critical because certain types of quantum noise, such as depolarizing noise, can naturally inject randomness into the learning process

without significantly degrading the performance of models (Du et al., 2021b). This raises a crucial open question: can this intrinsic quantum randomness be formally utilized as a stochastic technique to amplify the privacy guarantee originating from a preceding classical mechanism? To date, no work has theoretically established how to compose the privacy guarantees of classical and quantum noise sources within these hybrid models. In addition, understanding this relationship is crucial to control the preset privacy budget, especially considering that quantum noise in physical devices is inherently dynamic and difficult to precisely control.

**Contributions.** The key contributions and insights of this work can be highlighted as follows:

1. **Hybrid Privacy-Preserving Mechanism**. We propose HYPER-Q, a **HY**brid **P**rivacy-pres**ER**ving mechanism for **Q**uantum Neural Networks (QNNs). To the best of our knowledge, this is the first work to investigate the joint effect of classical and quantum noise in amplifying DP within quantum hybrid models. Specifically, HYPER-Q composes a classical input perturbation (e.g., Gaussian noise) with the intrinsic depolarizing noise of a quantum circuit, forming a dual-noise framework compatible with a broad class of QNNs.

2. **Privacy Guarantee Analysis**. We provide a rigorous analysis of HYPER-Q's DP guarantees. Our mechanism is a composition $Q^{(\eta)} \circ A$ where $A$ is a classical mechanism satisfying an original $(\varepsilon, \delta)$-DP and $Q^{(\eta)}$ is the quantum post-processing operation with the depolarizing noise factor of $\eta$. We analyze how this composition achieves new amplified privacy parameters $(\varepsilon', \delta')$. We provide three main analytical results:

   - First (Theorem 1): We show that quantum post-processing in a $d$-dimensional Hilbert space acts as a privacy amplifier by strictly reducing the failure probability (achieving $\delta' = \left[ \frac{\eta(1-e^\varepsilon)}{d} + (1-\eta)\delta \right]_+ < \delta$), while the privacy loss remains fixed ($\varepsilon' = \varepsilon$). This result directly implies stricter certifiable adversarial robustness.

   - Second (Theorem 2): We demonstrate that under a certain condition, it is possible to simultaneously amplify both parameters, $\varepsilon'$ and $\delta'$. This analysis yields two crucial insights. First, we show how to select Positive Operator-Valued Measures (POVMs) to maximize the privacy gain: the bound on $\delta'$ is minimized (i.e., the guarantee is strongest) when all POVM elements have equal trace. Second, we derive the explicit threshold that the quantum noise $\eta$ must exceed to guarantee the strict amplification of both privacy parameters.

   - Third (Theorems 1.1 and 1.2): We generalize the privacy amplification framework to asymmetric noise channels by identifying trace distance contraction as the core mechanism. We derive strict privacy amplification for Generalized Amplitude Damping (GAD) based on thermal relaxation ($\delta' = (2\sqrt{\eta} - \eta)\delta$) and for Generalized Dephasing (GD) under the assumption of product equatorial encoding, where the suppression of phase coherences scales the failure probability to $\delta' = |1 - 2\eta|\delta$.

3. **Utility Analysis**. We derive a formal utility bound (Theorem 3) that quantifies the model's performance. Specifically, we characterize the total error as a high-probability trade-off between the classical noise variance ($\sigma$) and the quantum depolarization probability ($\eta$).

4. **Empirical Experiments.** We empirically demonstrate that, under a fixed end-to-end privacy budget, HYPER-Q achieves significantly greater adversarial robustness than standard classical-only DP mechanisms across multiple datasets. These results indicate that replacing classical noise with quantum depolarizing noise can yield higher performance without weakening the privacy guarantee.

## 2 PRELIMINARY

### 2.1 QUANTUM INFORMATION BASICS

**Qubits and States.** Quantum computing systems operate on quantum bits (*qubits*). Unlike classical bits, qubits can exist in superpositions of 0 and 1. An $n$-qubit system resides in a $2^n$-dimensional Hilbert space $\mathcal{H}$. While ideal (pure) states are represented by vectors $|\psi\rangle$, general (possibly noisy) states are described by density matrices $\rho$: $d \times d$ positive semi-definite matrices with a trace of one (i.e., $\text{Tr}[\rho] = 1$).

**Quantum Channels.** The evolution of a quantum state, including noise effects, is modeled by a quantum channel. For example, the depolarizing channel, denoted as $f_{\text{dep}}^{(\eta)}$, replaces the state $\rho$ with the maximally mixed state $\frac{I}{d}$ with probability $\eta$ and leaves it unchanged with probability $1 - \eta$:

$$f_{\text{dep}}^{(\eta)}(\rho) = (1 - \eta)\rho + \eta\frac{I}{d}$$

where $\eta \in [0, 1]$ is the probability, $I$ is the identity matrix and $d$ is the dimension of the Hilbert space.

Classical information is extracted from a quantum state via measurement. A general measurement is defined by a set of operators $E_k$ forming a Positive Operator-Valued Measure (POVM). For a state $\rho$, the probability of observing the outcome $k$ is:

$$\Pr(\text{outcome} = k) = \text{Tr}[E_k\rho].$$

## 2.2 DIFFERENTIAL PRIVACY

Differential Privacy (DP) provides a formal guarantee that the presence or absence of any individual sample in a dataset has limited impact on the output (Dwork, 2006). More formally:

**Definition 1** (($\varepsilon, \delta$)-Differential Privacy). *A randomized mechanism $\mathcal{M} : \mathcal{D} \to \mathcal{R}$ satisfies ($\varepsilon, \delta$)-differential privacy if for any two adjacent datasets $D_1$ and $D_2$ that differs by a single element, , and for any subset of outputs $S \subseteq \mathcal{R}$, the following inequality holds:*

$$\Pr[\mathcal{M}(D_1) \in S] \leq e^\varepsilon \Pr[\mathcal{M}(D_2) \in S] + \delta$$

,

Here, $\varepsilon \geq 0$ is the privacy loss parameter while $\delta \in [0, 1)$ is the failure probability. The smaller $\varepsilon$ or the smaller $\delta$ implies stronger privacy.

An equivalent characterization of DP can be formulated using the **hockey-stick divergence**. For two distributions $P$ and $Q$, the hockey-stick divergence is defined as:

$$\text{D}_{e^\varepsilon}(P\|Q) = \int \max(0, P(x) - e^\varepsilon Q(x))dx$$

A mechanism $\mathcal{M}$ satisfies ($\varepsilon, \delta$)-DP if and only if $\text{D}_{e^\varepsilon}(\mathcal{M}(D_1)\|\mathcal{M}(D_2)) \leq \delta$ for all adjacent $D_1, D_2$.

This framework extends to the quantum setting (Hirche et al., 2023), where the quantum hockey-stick divergence for states $\rho, \rho'$ is defined as:

$$\text{D}_{e^\varepsilon}^{(q)}(\rho\|\rho') = \text{Tr}\left[(\rho - e^\varepsilon\rho')_+\right]$$

A quantum mechanism $\mathcal{E}$ satisfies ($\varepsilon, \delta$)-quantum DP if for any adjacent states $\rho, \rho'$, the divergence is bounded by $\delta$ where $\text{D}_{e^\varepsilon}^{(q)}(\mathcal{E}(\rho)\|\mathcal{E}(\rho')) \leq \delta$.

**Noise-added Mechanisms.** A standard way to achieve DP is by adding noise proportional to the sensitivity of a function, which is the maximum output change from altering one data point. The Gaussian mechanism adds noise $\eta_{\text{cdp}} \sim \mathcal{N}(0, \sigma^2 I)$ to a function $f : \mathcal{D} \to \mathbb{R}$ based on the function's $L_2$ sensitivity:

$$\Delta_2(f) = \max_{D_1, D_2} \|f(D_1) - f(D_2)\|_2$$

This mechanism outputs $f(x) + \eta_{\text{cdp}}$. For appropriate choices of $\sigma$, this mechanism satisfies ($\varepsilon, \delta$)-DP. Additional background on hybrid quantum machine learning, the connection between differential privacy and adversarial robustness, and classical noise mechanisms for achieving DP is provided in Appendix A.

## 3 RELATED WORKS

**Differential Privacy in Classical Machine Learning.** Differential Privacy (DP) has been established as a leading framework for protecting data in ML workflows. DP provides formal guarantees (Dwork

et al., 2006) that ensure that the inclusion or exclusion of a single data point has a limited impact on the output of an algorithm, thus minimizing the risk of information leakage. In machine learning, the most common way to achieve DP in practice is by injecting calibrated random noise into the learning process. This noise can be introduced at various stages, such as perturbing the input data (Lecuyer et al., 2019; Phan et al., 2019; Cohen et al., 2019), the gradients during optimization (Abadi et al., 2016; Ghazi et al., 2025), or the final model parameters (Yuan et al., 2023).

Input perturbation is particularly effective for providing instance-level privacy and is a key technique for certifying the adversarial robustness of a model's predictions (Lecuyer et al., 2019; Cohen et al., 2019). Standard mechanisms, such as the Gaussian or Laplacian mechanism, add noise scaled to the function's sensitivity to provide $(\varepsilon, \delta)$-DP guarantee (Dwork & Roth, 2014). To mitigate the degradation in model performance which is often caused by noise injection, a crucial line of research focuses on privacy amplification. The core idea is that certain stochastic processes can strengthen the final privacy guarantee without requiring additional initial noise. Privacy amplification can also be achieved through established techniques such as subsampling (Bun et al., 2015; Balle et al., 2018; Wang et al., 2019; Koga et al., 2022), shuffling (Cheu et al., 2018; Erlingsson et al., 2019; Balle et al., 2019b), iterative composition (Feldman et al., 2018), and specialized forms of post-processing (Balle et al., 2019a; Ye & Shokri, 2022). In particular, post-processing is fundamental: while standard post-processing can never weaken a privacy guarantee (Dwork, 2006), certain stochastic transformations can actively enhance it. However, not all combinations of stochastic sources yield amplification. For example, post-processing a Gaussian mechanism with an additional Gaussian transformation can amplify privacy, whereas composing a Gaussian mechanism with a Laplacian transformation does not yield such an effect.

**Differential Privacy in Quantum Settings.** The notion of DP has recently been extended to quantum settings, reflecting the growing interest of privacy-preserving quantum computing and quantum machine learning (QML). The foundational concept was introduced by (Zhou & Ying, 2017), who proposed a definition of QDP that is a direct quantum analogue of classical DP. Building on this, (Du et al., 2021a) demonstrated a practical application for QML by showing that inherent quantum noise could be leveraged to achieve QDP in quantum classifiers. Specifically, they analyzed the depolarizing noise channel as a privacy-preserving mechanism and derived the mathematical relationship between the noise strength and the resulting $(\varepsilon, \delta)$-QDP guarantee. They also proved that this privacy mechanism simultaneously enhances the model's adversarial robustness. Later, (Hirche et al., 2023) developed a comprehensive theoretical framework for QDP. Using tools such as quantum relative entropy, their work provides a more general and rigorous foundation for QDP. More recent works(Bai et al., 2024; Watkins et al., 2023; Song et al., 2025) have examined how various quantum noise sources, such as depolarizing, bit-flip, and phase-flip channels, affect the QDP budget.

Despite this progress in defining privacy for either purely quantum or purely classical systems, a critical gap remains for the hybrid quantum-classical architectures that are essential for near-term quantum advantage. These models are paramount for applying quantum computation to real-world problems. However, to date, no work has theoretically established how to compose the privacy guarantees of classical and quantum noise sources within hybrid quantum models. This significant gap highlights the importance of our proposed HYPER-Q and the need for further exploration of hybrid approaches that combine traditional DP mechanisms with the privacy properties innate to quantum systems.

## 4 HYBRID NOISE-ADDED MECHANISM

In this section, we present our privacy-preserving mechanism that integrates classical and quantum noise to achieve differential privacy (DP) in QNN models. We first describe the structure of the hybrid mechanism, then analyze its DP guarantees, and finally provide a utility bound that characterizes the impact of noise on model performance.

### 4.1 MECHANISM OVERVIEW

The proposed mechanism is designed to mitigate privacy leakage at two levels. First, classical data can be vulnerable to reconstruction attacks before it enters the quantum circuit. To prevent such exposure, we introduce classical noise mechanisms to perturb the input. Second, we leverage inherent

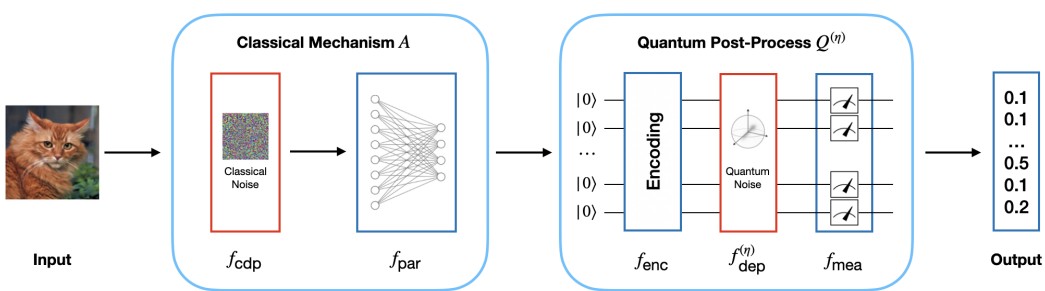

Figure 1: Overview of the proposed hybrid noise-added mechanism, HYPER-Q.

quantum depolarizing noise to enhance privacy after encoding. This noise has been shown to preserve utility in the ideal case of infinite measurements (Du et al., 2021b). By combining classical and quantum noise, our dual-layer approach reduces reliance on excessive classical noise, achieving stronger privacy with minimal utility loss.

We formally describe each stage of the mechanism using a modular function-based representation (see an overview in Figure 1):

**Classical Noise Function** $f_{\mathrm{cdp}} : \mathbb{X} \to \mathbb{X}$. This function adds calibrated classical noise to the input, providing an initial DP guarantee.

$$f_{\mathrm{cdp}}(x) = x + \eta_{\mathrm{cdp}}, \quad \text{where } \eta_{\mathrm{cdp}} \sim \mathcal{N}(0, \sigma^2 I)$$

Here, the noise $\eta_{\mathrm{cdp}}$ is drawn from a multivariate Gaussian distribution with covariance $\sigma^2 I$.

**Parameterized Linear Transformation** $f_{\mathrm{par}} : \mathbb{X} \to \mathbb{Y}$. This function serves as a learnable classical layer, transforming the input data into a feature space. The weights $W$ and biases $b$ are learnt during model training.

$$f_{\mathrm{par}}(x') = Wx' + b = y$$

**Quantum Encoding Function** $f_{\mathrm{enc}} : \mathbb{Y} \to \mathcal{H}$. This function encodes the classical feature vector $y$ into a quantum state $\rho$ within a $d$-dimensional Hilbert space $\mathcal{H}$ composed of $n$ qubits ($d = 2^n$). Let $|\psi_y\rangle = \prod_{j=1}^{n} e^{-iy_j H_j} |0\rangle^{\otimes n}$ be the encoded pure state vector, where $H_j$ are Hermitian operators. The function's output is the corresponding density matrix:

$$f_{\mathrm{enc}}(y) = |\psi_y\rangle\langle\psi_y| = \rho$$

**Depolarizing Noise Channel** $f_{\mathrm{dep}}^{(\eta)} : \mathcal{H} \to \mathcal{H}$. This quantum channel adds a second layer of randomness by applying noise directly to the encoded state $\rho$. This process will be shown to amplify the initial privacy guarantee from the classical noise layer in the subsequent analysis.

$$f_{\mathrm{dep}}^{(\eta)}(\rho) = (1 - \eta)\rho + \eta\frac{I}{d} = \tilde{\rho}$$

Here, $\eta \in [0, 1]$ is the depolarization probability, and $I$ is the identity operator on $\mathcal{H}$.

**Measurement Function** $f_{\mathrm{mea}} : \mathcal{H} \to \mathbb{Z}$. This final stage maps the noisy quantum state $\tilde{\rho}$ to a single classical class label $z$ from the output space $\mathbb{Z} = \{0, 1, \dots, K - 1\}$. This mapping is inherently stochastic and is formally defined as:

$$\Pr(f_{\mathrm{mea}}(\tilde{\rho}) = k) = \mathrm{Tr}[E_k \tilde{\rho}], \quad \forall k$$

This hybrid approach allows independent tuning of classical and quantum noise for flexible privacy-utility trade-offs. Its modular design also supports theoretical analysis of privacy guarantees and performance impact, as detailed below.

## 4.2 Differential Privacy Bound

We now define the concepts used in our DP analysis. Specifically, our proposed mechanism can be expressed as the composition $Q^{(\eta)} \circ A$, where $A = f_{\mathrm{par}} \circ f_{\mathrm{cdp}}$ is a classical mechanism satisfying

$(\varepsilon, \delta)$-DP, and $Q^{(\eta)} = f_{\text{meas}} \circ f_{\text{dep}}^{(\eta)} \circ f_{\text{enc}}$ is a quantum post-processing operation controlled by a noise parameter $\eta$. Assuming the random process $f_{\text{cdp}}$ satisfies $(\varepsilon, \delta)$-DP, it follows from the post-processing theorem (Dwork, 2006) that the mechanism $A$ also satisfies $(\varepsilon, \delta)$-DP.

Our goal is to analyze how the composed mechanism $Q^{(\eta)} \circ A$ achieves new privacy parameters $(\varepsilon', \delta')$, and how these parameters amplify the original guarantees $(\varepsilon, \delta)$. Specifically, we provide two analytical results for the proposed mechanism. In the first analysis, we show that $Q^{(\eta)} \circ A$ can improve the failure probability by establishing that $\varepsilon' = \varepsilon$ and $\delta' < \delta$. In the second analysis, we demonstrate that under certain conditions, $Q^{(\eta)} \circ A$ can amplify both the privacy loss and the failure probability, achieving $\varepsilon' < \varepsilon$ and $\delta' < \delta$. All proofs are presented in Appendix B.

### 4.2.1 FIRST ANALYSIS — AMPLIFYING THE FAILURE PROBABILITY

We investigate how the failure probability is amplified under quantum post-processing, assuming a fixed privacy loss parameter $\varepsilon$. Theorem 1 formalizes this by establishing a new bound on the failure probability $\delta'$ of the composed mechanism $Q^{(\eta)} \circ A$, while keeping the privacy loss fixed at $\varepsilon' = \varepsilon$. The proof for this theorem bridges the classical and quantum divergence measures by involving two key steps: (1) establishing that the classical hockey-stick divergence of the final, measured probabilities is upper-bounded by the quantum hockey-stick divergence of the quantum states before measurement , and (2) proving that this quantum divergence contracts under the depolarizing channel $f_{\text{dep}}^{(\eta)}$ by a factor of $(1 - \eta)$. The detailed derivation of Theorem 1, along with its corresponding proofs, is provided in Appendix B.

**Theorem 1** (Amplification on Failure Probability). *Let $A : \mathbb{X} \to \mathcal{P}(\mathbb{Y})$ be a classical mechanism satisfying $(\varepsilon, \delta)$-DP where $A = f_{par} \circ f_{cdp}$, and let $Q^{(\eta)} : \mathbb{Y} \to \mathcal{P}(\mathbb{Z})$ be a quantum mechanism in a $d$-dimensional Hilbert space defined as $Q^{(\eta)} = f_{mea} \circ f_{dep}^{(\eta)} \circ f_{enc}$ where $0 \leq \eta \leq 1$ is the depolarizing noise factor. Then, the composed mechanism $Q^{(\eta)} \circ A$ satisfies $(\varepsilon', \delta')$-DP, where*

$$\varepsilon' = \varepsilon, \quad \delta' = \left[ \frac{\eta(1 - e^{\varepsilon})}{d} + (1 - \eta)\delta \right]_+$$

From the final bound, it follows that for $\varepsilon \in [0, 1]$, we have $\delta' \leq \delta$. Therefore, the failure probability is strictly reduced, resulting in a privacy amplification effect, as formally stated in Corollary 1.

**Corollary 1.** *The composed mechanism $Q^{(\eta)} \circ A$ satisfies $(\varepsilon, \delta')$-DP with $\delta' < \delta$, thus strictly amplifying the overall failure probability.*

Based on (Lecuyer et al., 2019), we derive an explicit condition for certifiable adversarial robustness of the composed mechanism $Q^{(\eta)} \circ A$ in Corollary 2. This condition defines a robustness threshold that the model's expected confidence scores must exceed. Notably, due to the privacy amplification effect formalized in Corollary 1, the robustness threshold under the composed mechanism (parameterized by $\delta'$) is strictly lower than that of the original classical mechanism (parameterized by $\delta$). As a result, quantum post-processing provably enlarges the set of inputs for which adversarial robustness can be guaranteed. For further details on adversarial robustness, we refer readers to Appendix A.

**Corollary 2.** *The composed mechanism $Q^{(\eta)} \circ A$ is certifiably robust against adversarial perturbations for an input $x \in \mathbb{X}$ if the following condition holds for the correct class $k$:*

$$\mathbb{E}[[(Q^{(\eta)} \circ A)(x)]_k] > e^{2\varepsilon} \max_{i \neq k} \mathbb{E}[[(Q^{(\eta)} \circ A)(x)]_i] + (1 + e^{\varepsilon})\delta'$$

### 4.2.2 SECOND ANALYSIS — AMPLIFYING THE PRIVACY LOSS

We investigate how the composed mechanism $Q^{(\eta)} \circ A$ can simultaneously amplify both the privacy loss $\varepsilon$ and the failure probability $\delta$. The result is formalized in Theorem 2 which provides new $(\varepsilon', \delta')$ bound. The proof (detailed in Appendix B) relies on the *Advanced Joint Convexity* theory, originally introduced in (Balle et al., 2018). The key insight is that the depolarizing channel transforms the final output distribution into a convex combination of the original (noiseless) distribution and the distribution of a maximally mixed state. This explicit mixture structure allows the joint convexity theorem to be applied, yielding a new DP bound on both privacy loss and failure probability.

Theorem 2 reveals that the amplified failure probability $\delta'$ depends on the choice of POVMs. In particular, $\delta'$ becomes tighter as $\varphi = \min_k \left( \frac{\text{Tr}(E_k)}{d} \right)$ increases. This insight leads to Corollary 3, highlighting that $\delta'$ is minimized when all POVM elements $E_k$ have equal trace (i.e., $\text{Tr}(E_k) = \frac{1}{K}$).

Contrarily, $\varepsilon' \leq \varepsilon$ for all $\eta \in [0, 1]$, the privacy loss in terms of $\varepsilon$ is always reduced. However, the bound on $\delta$ is only improved (i.e., $\delta' \leq \delta$) when the noise level $\eta$ exceeds the threshold given in Corollary 4. This condition highlights that a sufficient level of quantum noise is required to achieve strict amplification of the privacy guarantee in both parameters.

**Theorem 2** (Amplification on Privacy Loss). *Let $A = f_{par} \circ f_{cdp}$ be $(\varepsilon, \delta)$-DP, and $Q^{(\eta)} = f_{mea} \circ f_{dep}^{(\eta)} \circ f_{enc}$ be a quantum mechanism in a $d$-dimensional Hilbert space where $0 \leq \eta \leq 1$ is the depolarizing noise factor. Then, the composition $Q^{(\eta)} \circ A$ is $(\varepsilon', \delta')$-DP where $\varepsilon' = \log\big(1 + (1 - \eta)(e^\varepsilon - 1)\big)$ and*
$$\delta' = (1 - \eta)\big(1 - e^{\varepsilon' - \varepsilon}(1 - \delta) - (e^\varepsilon - e^{\varepsilon'})\varphi\big) \text{ with } \varphi = \min_k \left( \frac{\text{Tr}(E_k)}{d} \right).$$

**Corollary 3.** *Let $\{E_k\}_{k=1}^K$ be the POVM used in $f_{mea}$. Then, the amplified failure probability $\delta'$ in Theorem 2 is minimized when all POVM elements have equal trace (i.e., $\text{Tr}[E_k] = \frac{d}{K}$ for all $k \in \{1, \ldots, K\}$).*

**Corollary 4.** *Given an optimal measurement such that $\text{Tr}[E_k] = \frac{d}{K} \forall k$, the composed mechanism $Q^{(\eta)} \circ A$ strictly improves the privacy guarantee (i.e., $\varepsilon' \leq \varepsilon$ and $\delta' \leq \delta$) if*
$$\eta \geq 1 - \frac{\delta}{(1 - \delta)(1 - e^{-\varepsilon}) - (e^\varepsilon - 1)/K}$$

### 4.2.3 Third Analysis — Generalization to Other Noise Channels

While our first analysis focuses on depolarizing noise, the underlying mechanism responsible for privacy amplification extends naturally to a broader class of quantum channels. The central insight is whenever a quantum noise channel induces a non-trivial contraction of the quantum hockey-stick divergence, it will inherently lead to privacy amplification. In this subsection, we show how this principle generalizes our analysis to two widely studied asymmetric noise models: Generalized Amplitude Damping (GAD) and Generalized Dephasing (GD).

**Amplification Under Generalized Amplitude Damping.** GAD channel is inherently asymmetric and non-unital. Despite this, we show that it contracts trace distance by a factor of at most $(2\sqrt{\eta} - \eta)$, where $\eta$ is the damping strength. Substituting this contraction into the proof framework for Theorem 1 yields the following amplification bound.

**Theorem 1.1** (Amplification Under Generalized Amplitude Damping Noise). *Let $A : \mathbb{X} \to \mathcal{P}(\mathbb{Y})$ be a classical mechanism satisfying $(\varepsilon, \delta)$-DP where $A = f_{par} \circ f_{cdp}$, and let $Q^{(p,\eta)} : \mathbb{Y} \to \mathcal{P}(\mathbb{Z})$ be a quantum mechanism in $d$-dimensional Hilbert space defined as $Q^{(p,\eta)} = f_{mea} \circ f_{GAD}^{(p,\eta)} \circ f_{enc}$. Then, the composed mechanism $Q^{(p,\eta)} \circ A$ satisfies $(\varepsilon', \delta')$-DP, where*
$$\varepsilon' = \varepsilon, \quad \delta' = (2\sqrt{\eta} - \eta)\delta.$$

**Generalized Dephasing Under Equatorial Encoding.** Dephasing noise preserves classical populations but suppresses quantum coherences. Although its worst-case contraction coefficient is 1, we show that for many QML encoding schemes, including angle-based encoders, the encoded states lie in the equatorial plane of the Bloch sphere. Under this structure, all distinguishability is encoded in coherence terms directly affected by GD noise, enabling nontrivial contraction.

**Assumption 1** (Product Equatorial Encoding on All Qubits). *For each input $y \in \mathbb{Y}$, the encoder prepares a product state*
$$\rho_y = f_{enc}(y) = \bigotimes_{j=1}^n \rho_y^{(j)},$$
*where each single-qubit factor $\rho_y^{(j)}$ is an equatorial state on the Bloch sphere, i.e.,*
$$\rho_y^{(j)} = \frac{1}{2}\Big(I + \cos\phi_y^{(j)}\, X + \sin\phi_y^{(j)}\, Y\Big),$$

*for some angle $\phi_y^{(j)} \in \mathbb{R}$ and with no Z-component.*

Under this assumption, the GD channel contracts all relevant coherence terms by a factor of $|1 - 2\eta|$, leading to the following privacy guarantee.

**Theorem 1.2.** *Let $A : \mathbb{X} \to \mathcal{P}(\mathbb{Y})$ be a classical mechanism satisfying $(\varepsilon, \delta)$-DP, and let*

$$Q^{(\eta)} := f_{\mathrm{mea}} \circ f_{\mathrm{GD}}^{(\eta)} \circ f_{\mathrm{enc}}$$

*be an $n$-qubit quantum mechanism where $f_{\mathrm{GD}}^{(\eta)}$ is the $n$-qubit GD channel defined above and $f_{\mathrm{enc}}$ satisfies Assumption 1. Then the composed mechanism $Q^{(\eta)} \circ A$ satisfies $(\varepsilon', \delta')$-DP with*

$$\varepsilon' = \varepsilon, \qquad \delta' = |1 - 2\eta| \cdot \delta.$$

The full proofs and derivations of Theorem 1.1 and 1.2 are provided in Appendix B.4.

### 4.3 UTILITY BOUND

We finally establish a rigorous framework to study the utility loss, defined as the absolute error between the noisy and noise-free versions of our mechanism. The final output of the mechanism is stochastic, due to the sampling-based measurement process. Thus, we analyze the difference between the expected values of their output. The expected value represents the average behavior of a mechanism and provides a deterministic quantity that we can use to measure utility loss.

Formally, we define the expectation measurement function $f_{\mathrm{exp}} : \mathcal{H} \to \mathbb{R}$ as:

$$f_{\mathrm{exp}}(\rho) = \sum_k k \operatorname{Tr}[E_k \rho] = \operatorname{Tr}\left[\left(\sum_k k E_k\right)\rho\right] = \operatorname{Tr}[E_{\mathrm{exp}}\rho]$$

where $E_{\mathrm{exp}} = \sum_k k E_k$ is the expectation value observable.

Using this function, we define our deterministic expectation mechanisms. The **full mechanism**, including classical and quantum noise, is $\mathcal{M}_{\mathrm{full}}(x) = (f_{\mathrm{exp}} \circ f_{\mathrm{dep}}^{(\eta)} \circ f_{\mathrm{enc}} \circ f_{\mathrm{par}} \circ f_{\mathrm{cdp}})(x)$. On the other hand, the **noise-free mechanism (clean)** is $\mathcal{M}_{\mathrm{clean}}(x) = (f_{\mathrm{exp}} \circ f_{\mathrm{enc}} \circ f_{\mathrm{par}})(x)$. The total utility loss is the worst-case absolute error between their expected outputs:

$$\text{Error} = \sup_{x \in \mathbb{X}} |\mathcal{M}_{\mathrm{full}}(x) - \mathcal{M}_{\mathrm{clean}}(x)|$$

**Theorem 3** (Utility Bound)**.** *Let the classical noise be $\kappa \sim \mathcal{N}(0, \sigma^2 I)$ acting on an input space $\mathbb{X}$ of dimension $d_X = \dim(\mathbb{X})$. For any desired failure probability $p > 0$, the utility loss is bounded probabilistically as:*

$$\Pr\left(\text{Error} \le L_\infty \cdot \sigma \sqrt{2 \ln \frac{2d_X}{p}} + 2\eta \|E_{\exp}\|_{op}\right) \ge 1 - p$$

*where $L_\infty = 2(1 - \eta) \|E_{\exp}\|_{op} \|W\|_\infty \left(\sum_j \|H_j\|_{op}\right)$.*

Theorem 3 provides a utility bound that quantifies the trade-off between privacy and performance. The proof (detailed in Appendix B) utilizes an **intermediate mechanism (half)** that includes only quantum noise as $\mathcal{M}_{\mathrm{half}}(x) = (f_{\mathrm{exp}} \circ f_{\mathrm{dep}}^{(\eta)} \circ f_{\mathrm{enc}} \circ f_{\mathrm{par}})(x)$. Specifically, first, we bound the error introduced by the quantum noise ($|\mathcal{M}_{\mathrm{half}} - \mathcal{M}_{\mathrm{clean}}|$), which is shown to be proportional to the quantum noise level $\eta$. Second, we bound the error from the classical noise by establishing a Lipschitz constant $L_\infty$ for the quantum-only mechanism. As the classical noise is unbounded, the final guarantee is a high-probability statement relating the utility loss to the classical ($\sigma$) and quantum ($\eta$) noise levels.

## 5 EXPERIMENTAL EVALUATION

We empirically evaluate HYPER-Q, focusing on adversarial robustness, a direct outcome of the Differential Privacy (DP) guarantees in Corollary 2. Specifically, we aim to show that for a fixed

privacy budget $(\varepsilon', \delta')$, the hybrid noise strategy of `HYPER-Q` yields higher model utility than the purely classical mechanisms including Basic Gaussian, Analytic Gaussian (Balle & Wang, 2018) and DP-SGD Abadi et al. (2016); Watkins et al. (2023) (more details can be found in Appendix A). We note that the first two mechanisms apply noise at the input level, whereas DP-SGD performs noise injection at the gradient level. We first evaluate `HYPER-Q` across various quantum noise settings and compare its performance to that of the classical mechanisms on a quantum machine learning (QML) model. We then benchmark the performance of the `HYPER-Q`-equipped QML model against various classical learning models protected by the Analytic Gaussian mechanism. Each experiment reports the averaged accuracy over 10 runs.

**Implementation Details.** We implement a QML model designed to incorporate `HYPER-Q`. The model architecture follows the mechanism proposed and analyzed in Section 4. The implementation uses the PennyLane library (Bergholm et al., 2022), with quantum circuits executed on simulators, which is a standard practice for prototyping and evaluating quantum applications (Cicero et al., 2025). To ensure DP, Gaussian noise is added directly to the input and depolarizing noise is applied as a layer in the quantum circuit. Specifically, given a target privacy budget $(\varepsilon', \delta')$, the depolarizing noise level $\eta$ is fixed, while the Gaussian noise level $\sigma^2$ is computed according to Theorem 1. Additional details are provided in Appendix C.

**Datasets & Benchmark Models.** We evaluate our approach on three standard image classification datasets: MNIST (Lecun et al., 1998), FashionMNIST (Xiao et al., 2017), and USPS (Hull, 2002). To assess the practical viability of `HYPER-Q`, we compare its robustness against three standard deep learning architectures: a Multi-Layer Perceptron (MLP), a ResNet-9-based Convolutional Neural Network (CNN) (He et al., 2016), and a Vision Transformer (ViT) (Dosovitskiy et al., 2021). Each of these classical models is protected by the Analytic Gaussian mechanism with identical privacy budgets. Specific descriptions of each dataset and benchmark are provided in Appendix D.

**Adversarial Robustness Settings.** We use a certified defense framework (Lecuyer et al., 2019) that trains models with noise layers calibrated by a DP budget $(\varepsilon', \delta')$ and a construction attack bound $L_{\text{cons}}$. We then evaluate robustness by measuring the model's accuracy against FGSM (Goodfellow et al., 2015) and PGD (Madry et al., 2018) attacks, whose strength is defined by the empirical attack bound $L_{\text{attk}}$. More details are provided in Appendix E.

## 5.1 ROBUSTNESS ANALYSIS IN QML

In this experiment, we illustrate that under the same privacy budget, `HYPER-Q` preserves adversarial robustness more efficiently than classical mechanisms in QML. We evaluate the adversarial robustness of `HYPER-Q` under two quantum noise settings, $\eta \in \{0.1, 0.3\}$. We compare its performance with Basic Gaussian, Analytic Gaussian and DP-SGD mechanisms. For fair comparisons, we ensure that all methods are evaluated under the same privacy budget and applied to the same QML model.

Figure 2 presents the average accuracy on the MNIST, FashionMNIST, and USPS datasets under both FGSM and PGD attacks for four distinct privacy budgets $\varepsilon' \in \{0.25, 0.5, 0.75, 1\}$. We observe that `HYPER-Q` with $\eta = 0.1$ consistently outperforms all baseline methods, both in the absence of attack ($L_{\text{attk}} = 0$) and under attack ($L_{\text{attk}} > 0$). As the $\varepsilon'$ increases, the performance gap becomes more pronounced. Specifically, `HYPER-Q` surpasses the second-best method, Analytic Gaussian, by an average of 16.54%, 5.37%, 6.44%, and 5.20% in accuracy across the four respective $\varepsilon'$ values. This demonstrates that replacing a reasonable amount of classical noise with quantum noise can significantly enhance adversarial accuracy. In addition, we observe that while `HYPER-Q` with $\eta = 0.3$ performs better than classical mechanisms at $\varepsilon' = 0.25$, its relative efficiency decreases at higher settings of $\varepsilon'$ where the amount of classical noise added diminishes. This suggests that when quantum noise outweighs classical noise, the overall performance degrades. Therefore, selecting an appropriate value of $\eta$ is crucial. For a detailed analysis of $\eta$, we refer readers to Appendix F.5.

## 5.2 COMPARATIVE BENCHMARK WITH CLASSICAL MODELS

`HYPER-Q` is intrinsically designed for QML models. This raises a critical question of practical viability: *Can a QML model protected by `HYPER-Q` compare to or outperform classical models that are protected by their own conventional privacy mechanisms?* Figure 3 illustrates the performance comparison of a QML model protected by `HYPER-Q` (with its empirically best quantum noise

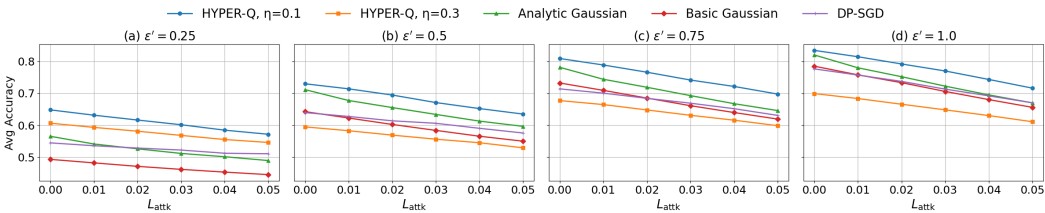

Figure 2: Average accuracy of noise-added mechanisms under FGSM and PGD attacks on MNIST, FashionMNIST, and USPS. Accuracy is averaged over all $L_{\text{cons}}$ settings for each $(L_{\text{attk}}, \varepsilon')$. HYPER-Q is evaluated with $\eta \in [0.1, 0.3]$ and $\delta' = 1 \times 10^{-5}$.

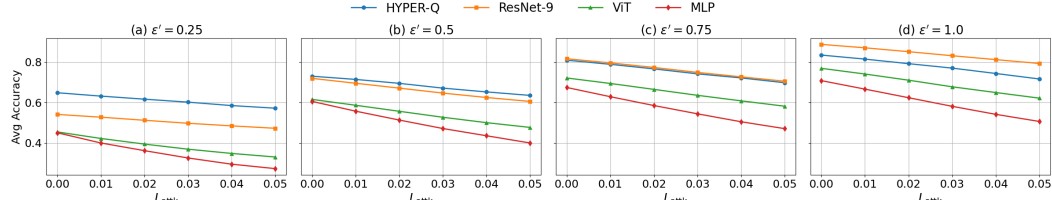

Figure 3: Average accuracy of the QML model with HYPER-Q protection versus three classical baselines (ResNet-9, ViT, and MLP) with Analytic Gaussian protection, averaged over FGSM and PGD attacks and across MNIST, FashionMNIST, and USPS. The HYPER-Q model is evaluated with its empirically best quantum noise setting ($\eta = 0.1$). For each $(L_{\text{attk}}, \varepsilon')$ pair, the reported accuracy is averaged over all $L_{\text{cons}}$ settings. $\delta' = 1 \times 10^{-5}$ for all settings.

setting, $\eta = 0.1$) against three classical baselines protected by Analytic Gaussian noise. We observe that for smaller privacy parameters, $\varepsilon' \in \{0.25, 0.5\}$, HYPER-Q outperforms the best classical baseline (ResNet-9) by $20.44\%$ and $3.41\%$ in average accuracy, respectively. This indicates that a large amount of Gaussian noise can significantly degrade model performance, and in such cases, substituting classical noise with quantum noise can result in better utility. However, for larger $\varepsilon'$ values, HYPER-Q performs comparably (at $\varepsilon' = 0.75$) and worse (at $\varepsilon' = 1$) than ResNet-9. This suggests that when only a small amount of classical noise is needed to preserve the utility of a classical model, QML may not yet offer a performance advantage due to current limitations in quantum systems compared to their classical counterparts.

For a complete performance evaluation, including results on each dataset (MNIST, FashionMNIST, and USPS) and robustness against each attack (FGSM and PGD), we refer the reader to Appendix F. In Appendix F, we also provide analysis of dimensional scalability, verification of utility bound tightness, sensitivity analysis of $\eta$ and analysis of performance on CIFAR-10.

## 6 CONCLUSION

In this work we have presented HYPER-Q as a hybrid privacy-preserving mechanism for quantum systems. Through extensive experimental analyses across three real-world datasets subjected to the FGSM and PGD attacks, we demonstrate that the combination of quantum and classical noise is both robust and scalable, while yielding significant improvements in privacy preservation and model utility. Classical components ensure stable training and feasibility in interpretation, while quantum noise introduces natural randomness that enhances privacy without heavily degrading model utility. As quantum hardware matures, we expect frameworks like HYPER-Q to be essential in shaping the future of privacy-preserving ML. An important direction for future work is to investigate the behavior of hybrid DP mechanisms on larger variational circuits deployed on actual quantum hardware.

## REPRODUCIBILITY STATEMENT

All datasets used in this work are publicly available for download. We include the model architecture of the proposed method, HYPER-Q, in Appendix C along with resources used to implement our work. Furthermore, we include descriptions of the benchmarks along with their respective citations for reproducibility in Appendix D. We also describe our specific hyperparameters to replicate our results. A repository to our code will be made publicly available upon acceptance.

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

APPENDIX

# A   ADDITIONAL BACKGROUND

## A.1   QUANTUM NEURAL NETWORKS

Quantum neural networks (QNNs) are a class of quantum machine learning models that employ parameterized quantum circuits to learn from classical or quantum data. In this work, we focus on QNNs designed for classical input. In a supervised learning context, a QNN aims to approximate an unknown function $K : \mathbb{X} \to \mathbb{Y}$ by training on a dataset $S = \{(x_i, y_i)\}_{i=1}^N$, where each $x_i \in \mathbb{R}^d$ is an input data vector and $y_i$ is the associated label.

QNN models use parameterized quantum circuits to process data. The workflow for a typical QNN involves:

- **Data Encoding:** Classical data is mapped into the quantum state of qubits using a parameterized "encoder" circuit. This step is crucial, as it can be trained to find powerful data representations and can introduce quantum features like entanglement to increase the model's capacity.

- **Model Circuit:** A sequence of parameterized quantum gates, analogous to the layers of a classical neural network, processes the encoded quantum state.

- **Measurement:** A measurement is performed on the final state to extract a classical output, which serves as the model's prediction.

Training a QNN is a hybrid quantum-classical process. The quantum computer executes the circuit and performs the measurement. A classical computer then calculates a loss function (e.g., Mean Squared Error) to quantify the error between the prediction and the true label. Given a differentiable loss function $f(\cdot)$, the objective is to minimize:

$$\mathcal{L}(\theta) = \sum_{i=1}^N f(\ell_i(\theta; y_i), y_i).$$

Finally, the classical computer uses gradient-based optimization to update the circuit's parameters, $\theta$. This process is repeated iteratively until the model converges. The goal is to find the optimal parameters $\theta^*$ that minimize the loss:

$$\theta^* = \arg\min_\theta L(\theta).$$

## A.2   ADVERSARIAL ROBUSTNESS

A model is considered *adversarially robust* if it can consistently make correct predictions even when its inputs are slightly altered by malicious perturbations. These altered inputs are known as *adversarial samples*. Formally, we define a model $f : \mathbb{X} \to \mathbb{Y}$, which maps an input in the space $\mathbb{X}$ to an output distribution over labels $y = \{y_1, y_2, \ldots, y_k\} \in \mathbb{Y}$. The model $f$ is considered adversarially robust if its prediction for an input $x$ is unchanged when a small perturbation $\alpha$ is added to $x$. This can be stated as:
$$\max_{i \in [1,k]} [f(x)]_i = \max_{i \in [1,k]} [f(x + \alpha)]_i, \quad \forall \alpha \in B_p(L),$$

where $B_p(L_{\text{cons}})$ represents the $p$-norm ball of radius $L_{\text{cons}}$, that restricts the perturbation size to $\|\alpha\|_p \leq L_{\text{cons}}$. We also call $L_{\text{cons}}$ as the construction bound.

Recently, Differential Privacy (DP) has emerged as a promising approach to enhance model robustness. Originally developed to protect individual data in statistical databases, DP ensures that the output of an algorithm does not significantly change when a single individual's data is added or removed. This is typically achieved by injecting carefully calibrated randomness into the algorithm's computation. This property of prediction stability forms the foundation of the connection between DP and adversarial robustness, as explored in (Lecuyer et al., 2019). By design, models trained with DP noise are

inherently less sensitive to small input perturbations, thereby improving their resistance to adversarial attacks.

Formally, given a model $f$ which is $(\varepsilon, \delta)$ differentially privated under a $p$-norm metric, it is guaranteed to be robust against adversarial perturbations $\alpha$ of size $\|\alpha\|_p \leq 1$ if the following condition holds Lecuyer et al. (2019):

$$E([f(x)]_k) > e^{2\varepsilon} \max_{i:i\neq k} E([f(x)]_i) + (1 + e^{\varepsilon})\delta, \exists k \in K,$$

where $E([f(x)]_k)$ is the expected confidence score for the correct label $k$, and $E([f(x)]_i)$ is the expected confidence score for other labels.

This condition certifies that any input satisfying the inequality is immune to adversarial attacks within the defined perturbation size. A stronger DP guarantee, meaning smaller values for $\varepsilon$ and $\delta$, expands the set of inputs for which this robustness holds. In this work, our goal is to explore how quantum noise can amplify the DP guarantee, thereby significantly enhancing the model's overall robustness.

### A.3 NOISE MECHANISMS

Noise injection is a simple, yet, useful technique that can achieve DP guarantees by perturbing inputs, gradients, or outputs. In this work, we focus on input-perturbation mechanisms that satisfy $(\varepsilon, \delta)$-DP. For adversarial robustness, the amount of noise added is determined by three factors: the desired privacy budget $(\varepsilon, \delta)$, the sensitivity $\Delta$ of the function, and the construction bound $L_{\text{cons}}$. Because we add noise directly into the input, we have the trivial sensitivity $\Delta = 1$ (Lecuyer et al., 2019). Thus, we can omit it in the following analysis. Below, we summarize three common noise-added mechanisms:

**Basic Gaussian.** The Gaussian mechanism is a standard approach for providing $(\varepsilon, \delta)$-DP. The Gaussian mechanism introduces noise from a normal distribution with zero mean and a variance calibrated to predefined privacy parameters (Dong et al., 2022). It's well-suited for functions whose sensitivity is measured using the $\ell_2$ norm. Given a function $f$ with a construction bound $L_{\text{cons}}$ measured in $\ell_2$ norm, the mechanism achieves $(\varepsilon, \delta)$-DP by adding noise $\mathcal{N}(0, \sigma^2 I)$ with $\sigma$ is computed as:

$$\sigma = \sqrt{2 \ln\left(\frac{1.25}{\delta}\right)} L_{\text{cons}}/\varepsilon$$

**Analytic Gaussian.** The analytic Gaussian mechanism improves on the basic Gaussian approach by exploiting tighter bounds derived from the privacy loss distribution (Balle & Wang, 2018). Specifically, we can implicitly characterize the privacy loss as (Cullen et al., 2024):

$$\delta(\varepsilon) = \Phi\left(-\frac{L_{\text{cons}}}{2\sigma} + \frac{\varepsilon\sigma}{L_{\text{cons}}}\right) - e^{\varepsilon} \cdot \Phi\left(-\frac{L_{\text{cons}}}{2\sigma} - \frac{\varepsilon\sigma}{L_{\text{cons}}}\right)$$

where $\Phi(\cdot)$ is the cumulative distribution function of the standard Gaussian distribution. This formulation allows us to numerically solve for the minimum $\sigma$ required to satisfy a target $(\varepsilon, \delta)$.

**Laplacian.** The Laplace mechanism introduces noise based on the Laplace distribution and centered at zero with scale proportional to defined sensitivity (Dwork et al., 2006). It is typically used in settings that call for $\epsilon$-DP. The noise introduced is proportional to the sensitivity of the function being analyzed, ensuring that small adjustments to input data produce statistically similar outputs. In this work, we focus on flexible mechanisms which are able to achieve $(\varepsilon, \delta)$-DP, so we do not consider Laplacian for our comparison.

**DP-SGD.** The Differentially Private Stochastic Gradient Descent (DP-SGD) algorithm achieves privacy by clipping per-sample gradients and adding calibrated Gaussian noise during each optimization step Abadi et al. (2016). As a gradient-perturbation mechanism, DP-SGD is designed to provide $(\varepsilon, \delta)$-DP while maintaining compatibility with large-scale deep learning. The privacy guarantee arises from controlling the sensitivity of gradient updates and injecting noise proportional

to this sensitivity. Although DP-SGD is a well-established baseline in classical machine learning, its behavior in QML settings remains largely unexplored. Beyond a direct application presented in Watkins et al. (2023), the interplay between DP-SGD's gradient noise and quantum gradient estimation has not been systematically examined. For completeness, we include DP-SGD in our comparison as a representative gradient-level classical mechanism.

## B THEORETICAL DERIVATIONS AND PROOFS

### B.1 DERIVATION OF THEOREM 1

We investigate how the failure probability is amplified under quantum post-processing, assuming a fixed privacy loss parameter $\varepsilon$. Specifically, we aim to upper bound the quantity:

$$\sup_{x,x'} D_{e^\varepsilon} \left( Q^{(\eta)} \circ A(x) \,\|\, Q^{(\eta)} \circ A(x') \right) \tag{1}$$

**Lemma 1.** *Let $\mu$ and $\nu$ be probability distributions such that $D_{e^\varepsilon}(\mu\|\nu) \leq \delta$, and define $\theta = D_{e^\varepsilon}(\mu\|\nu)$. Then, there exist distributions $\mu'$, $\nu'$, and $\omega$, along with a parameter $\tilde{\varepsilon} := \log\left(1 + \frac{e^\varepsilon - 1}{\theta}\right)$ such that:*

$$\mu = (1-\theta)\omega + \theta\mu', \quad \nu = \frac{1-\theta}{e^\varepsilon}\omega + \left(1 - \frac{1-\theta}{e^\varepsilon}\right)\nu',$$

*with disjoint distributions: $\mu' \perp \nu'$. Then, the following bound holds:*

$$D_{e^\varepsilon}(\mu\|\nu) = \theta \cdot D_{e^{\tilde{\varepsilon}}}(\mu'\|\nu')$$

Let the output distributions of $A$ be denoted by $\mu = A(x)$ and $\nu = A(x')$, where $\mu, \nu \in \mathcal{P}(\mathbb{Y})$. Lemma 1, originally studied in (Balle et al., 2019a), establishes a decomposition of two distributions $\mu$ and $\nu$ based on their divergence $\theta = D_{e^\varepsilon}(\mu\|\nu)$. Specifically, $\mu$ is decomposed into a mixture of an overlapping component $\omega$ and a residual component $\mu'$, while $\nu$ is similarly decomposed into $\omega$ and a residual $\nu'$. The shared component $\omega$ is defined via the density $p_\omega = \frac{\min(p_\mu, e^\varepsilon p_\nu)}{1-\theta}$ The remaining distributions $\mu'$ and $\nu'$ correspond to the non-overlapping parts of $\mu$ and $\nu$, and it is shown that they have disjoint support (i.e., $\mu' \perp \nu'$). Lemma 1 also yields a transformation of the divergence between $\mu$ and $\nu$ in terms of the divergence between their respective components $\mu'$ and $\nu'$, specifically $D_{e^\varepsilon}(\mu\|\nu) = \theta \cdot D_{e^{\tilde{\varepsilon}}}(\mu'\|\nu')$. Because the quantum process $Q$ is a linear map (Nielsen & Chuang, 2010), it preserves convex combinations of input distributions. Consequently, we obtain

$$D_{e^\varepsilon}(\mu Q\|\nu Q) = \theta \cdot D_{e^{\tilde{\varepsilon}}}(\mu' Q\|\nu' Q) \tag{2}$$

In addition, the orthogonality of $\mu'$ and $\nu'$ plays a crucial role in analyzing the contraction behavior of post-processing mechanisms, as will be demonstrated in the following lemma.

**Lemma 2.** *Given a post-process mechanism $Q$, we have:*

$$\sup_{\mu \perp \nu} D_\varepsilon(\mu Q\|\nu Q) \leq \sup_{y \neq y'} D_\varepsilon(Q(y)\|Q(y'))$$

Lemma 2 establishes an upper bound on the divergence between two orthogonal distributions after applying a post-processing mechanism. Let $\tau_y$ denote the point mass distribution at $y$, i.e., $\tau_y(\tilde{y}) = 1$ if $\tilde{y} = y$ and $\tau_y(\tilde{y}) = 0$ otherwise. Then, $\mu = \sum_{y \in \text{supp}(\mu)} \mu(y)\tau_y$ and similarly for $\nu$. By convexity of $D_{e^\varepsilon}$ and linearity of $Q$, we have:

$$D_{e^\varepsilon}(\mu Q\|\nu Q) \leq \sup_{y \neq y'} D_{e^\varepsilon}(\tau_y Q\|\tau_{y'} Q) \leq \sup_{y \neq y'} D_{e^\varepsilon}(Q(y)\|Q(y'))$$

Together, Lemmas 1 and 2 clarify how the divergence between the outputs of $A$ transforms under post-processing. It remains to analyze the divergence induced solely by $Q^{(\eta)}$, allowing us to focus on bounding:

$$\sup_{y \neq y'} \mathrm{D}_{e^\varepsilon}(Q^{(\eta)}(y) \| Q^{(\eta)}(y')) \tag{3}$$

**Lemma 3.** *Given a measurement $E = \{E_i\}$ with $\sum_i E_i = I$, and two quantum states $\rho$ and $\rho'$, the classical hockey-stick divergence of the resulting probability distributions is less than or equal to the quantum hockey-stick divergence between the states.*

$$\mathrm{D}_\alpha(P \| P') \leq D_\alpha^{(q)}(\rho \| \rho')$$

Lemma 3 establishes the dependence between classical and quantum hockey-stick divergences under a fixed measurement. As a consequence, we can eliminate the explicit measurement map $f_{\text{mea}}$ from the post-processing pipeline. Specifically, we have:

$$\mathrm{D}_\alpha\big(Q^{(\eta)}(y) \| Q^{(\eta)}(y')\big) \leq \mathrm{D}_\alpha^{(q)}\big(f_{\text{dep}}^{(\eta)} \circ f_{\text{enc}}(y) \| f_{\text{dep}}^{(\eta)} \circ f_{\text{enc}}(y')\big) \tag{4}$$

**Lemma 4.** *Given a depolarizing channel $f_{dep}^{(\eta)}(\rho) = \eta\frac{I}{d} + (1 - \eta)\rho$, for $\eta \in [0, 1]$ and $\alpha \geq 1$, we have:*

$$D_\alpha^{(q)}(f_{dep}^{(\eta)}(\rho) \| f_{dep}^{(\eta)}(\rho')) \leq \max\left\{0, (1 - \alpha)\frac{\eta}{d} + (1 - \eta)D_\alpha^{(q)}(\rho \| \rho')\right\}$$

Lemma 4 establishes that the quantum hockey-stick divergence contracts under a depolarizing channel by a factor of $(1 - \eta)$, with an additive term depending on $\alpha$ and the dimension $d$. Applying this result with $\rho = f_{\text{enc}}(y)$ and $\rho' = f_{\text{enc}}(y')$ yields an upper bound on the right-hand side of Equation 4, which in turn provides a bound for Equation 3.

**Theorem 1** (Amplification on Failure Probability). *Let $A : \mathbb{X} \to \mathcal{P}(\mathbb{Y})$ be a classical mechanism satisfying $(\varepsilon, \delta)$-DP where $A = f_{par} \circ f_{cdp}$, and let $Q^{(\eta)} : \mathbb{Y} \to \mathcal{P}(\mathbb{Z})$ be a quantum mechanism in a $d$-dimensional Hilbert space defined as $Q^{(\eta)} = f_{mea} \circ f_{dep}^{(\eta)} \circ f_{enc}$ where $0 \leq \eta \leq 1$ is the depolarizing noise factor. Then, the composed mechanism $Q^{(\eta)} \circ A$ satisfies $(\varepsilon', \delta')$-DP, where*

$$\varepsilon' = \varepsilon, \quad \delta' = \left[\frac{\eta(1 - e^\varepsilon)}{d} + (1 - \eta)\delta\right]_+$$

Theorem 1 establishes a bound on the failure probability $\delta'$ of the composed mechanism $Q^{(\eta)} \circ A$, while keeping the privacy loss fixed at $\varepsilon' = \varepsilon$. This result is derived by sequentially applying Lemmas 2, 3, and 4 to Equation 2. From the final bound, it follows that for $\varepsilon \in [0, 1]$, we have $\delta' \leq \delta$. Therefore, the failure probability is strictly reduced, resulting in a privacy amplification effect, as formally stated in Corollary 1.

Based on (Lecuyer et al., 2019), we derive an explicit condition for certifiable adversarial robustness of the composed mechanism $Q^{(\eta)} \circ A$ in Corollary 2. This condition defines a robustness threshold that the model's expected confidence scores must exceed. Notably, due to the privacy amplification effect formalized in Corollary 1, the robustness threshold under the composed mechanism (parameterized by $\delta'$) is strictly lower than that of the original classical mechanism (parameterized by $\delta$). As a result, quantum post-processing provably enlarges the set of inputs for which adversarial robustness can be guaranteed. For further details on adversarial robustness, we refer readers to Appendix A.

**Corollary 1.** *The composed mechanism $Q^{(\eta)} \circ A$ satisfies $(\varepsilon, \delta')$-DP with $\delta' < \delta$, thus strictly amplifying the overall failure probability.*

**Corollary 2.** *The composed mechanism $Q^{(\eta)} \circ A$ is certifiably robust against adversarial perturbations for an input $x \in \mathbb{X}$ if the following condition holds for the correct class $k$:*

$$\mathbb{E}[[(Q^{(\eta)} \circ A)(x)]_k] > e^{2\varepsilon} \max_{i \neq k} \mathbb{E}[[(Q^{(\eta)} \circ A)(x)]_i] + (1 + e^\varepsilon)\delta'$$

## B.2 Derivation of Theorem 2

We investigate how the composed mechanism $Q^{(\eta)} \circ A$ can simultaneously amplify both the privacy loss $\varepsilon$ and the failure probability $\delta$. Our approach relies on the *Advanced Joint Convexity* theory, originally introduced in (Balle et al., 2018). We restate the theory below as Lemma 5.

**Lemma 5** (Advanced Joint Convexity). *Let $\mu, \mu'$ be probability distributions such that*

$$\mu = (1 - \sigma)\mu_0 + \sigma\mu_1, \quad \mu' = (1 - \sigma)\mu_0 + \sigma\mu_1',$$

*for some $\sigma \in [0, 1]$, and distributions $\mu_0, \mu_1, \mu_1'$. Given $\alpha \geq 1$, define $\alpha' = 1 + \sigma(\alpha - 1), \quad \beta = \frac{\alpha'}{\alpha}$. Then the following inequality holds:*

$$D_{\alpha'}(\mu\|\mu') \leq (1 - \beta)\sigma D_\alpha(\mu_1\|\mu_0) + \beta\sigma D_\alpha(\mu_1\|\mu_1')$$

Lemma 5 provides an upper bound on the divergence $D_{\alpha'}(\mu\|\mu')$ in terms of $D_\alpha$ divergences between the component distributions $\mu_0$, $\mu_1$, and $\mu_1'$. The bound becomes tighter as the contribution of the shared (overlapping) distribution $\mu_0$, controlled by the mixing parameter $\sigma$, increases. Returning to our analysis, given $\mu = A(x)$ and $\nu = A(x')$ for adjacent inputs $x \sim x'$, and a tighter privacy loss $\varepsilon' = \log[1 + \sigma(e^\varepsilon - 1)]$ we are able to bound $D_{e^{\varepsilon'}}(\mu Q^{(\eta)}\|\nu Q^{(\eta)})$ by identifying the shared component between the distributions $\mu Q^{(\eta)}$ and $\nu Q^{(\eta)}$ as illustrated in Lemma 6.

**Lemma 6.** *Let $\rho$ be a density matrix on a $d$-dimensional Hilbert space, and let*

$$\rho' = f_{dep}(\rho) = \eta\frac{I}{d} + (1 - \eta)\rho$$

*be its depolarized version, where $0 \leq \eta \leq 1$. Let $\{E_k\}_{k=1}^K$ be a POVM satisfying $\sum_k E_k = I$. Then, the measurement probabilities satisfy:*

$$\zeta'(k) = \frac{\eta}{d}\operatorname{Tr}(E_k) + (1 - \eta)\zeta(k),$$

*where $\zeta' = f_{mea}(\rho')$ and $\zeta = f_{mea}(\rho)$ with $\zeta', \zeta \in \mathcal{P}(\mathbb{Z})$.*

Lemma 6 establishes how depolarizing noise in the quantum system $\mathcal{H}$ affects the resulting classical output distribution over $\mathbb{Z}$. Specifically, it shows that the measurement distribution under depolarization becomes a convex combination of the original (noiseless) distribution and that of a maximally mixed state, with the noise strength $\eta$ controlling the mixing ratio.

Based on Lemma 6, we can decompose the output distributions $\mu Q^{(\eta)}$ and $\nu Q^{(\eta)}$ accordingly. By definition, the quantum mechanism $Q^{(\eta)}$ can be expressed as a convex combination of two mechanisms: $Q^{(0)}$ (applies no noise) and $Q^{(1)}$ (applies full depolarizing noise). The mechanism $Q^{(1)}$ is constant, as it always outputs the measurement distribution of a maximally mixed state. That is, for all $y \in \mathbb{Y}$, we have $Q^{(1)}(y)(k) = \frac{\operatorname{Tr}(E_k)}{d}$, where $E_k$ is the $k$-th POVM element and $d = \dim(\mathcal{H})$. We denote this constant output distribution as $\zeta_{\mathrm{mix}}$. On the other hand, $Q^{(0)}(y)$ corresponds to the noiseless distribution $\zeta$, and $Q^{(\eta)}(y)$ corresponds to the distribution $\zeta'$ defined in Lemma 6. Using the decomposition given by the lemma, we have

$$Q^{(\eta)}(y) = \eta Q^{(1)}(y) + (1 - \eta)Q^{(0)}(y), \quad \forall y \in \mathbb{Y}$$

Using the linearity of $Q^{(\eta)}$ and the representations $\mu = \sum_{y\in\mathrm{supp}(\mu)} \mu(y)\tau_y$ and $\nu = \sum_{y\in\mathrm{supp}(\nu)} \nu(y)\tau_y$, we obtain $\mu Q^{(\eta)} = \eta\zeta_{\mathrm{mix}} + (1 - \eta)\mu Q^{(0)}$, and $\nu Q^{(\eta)} = \eta\zeta_{\mathrm{mix}} + (1 - \eta)\nu Q^{(0)}$.

By applying the *Advanced Joint Convexity* theory (Lemma 5) on $\mu Q^{(\eta)}$ and $\nu Q^{(\eta)}$ with $\varepsilon' = \log(1 + (1 - \eta)(e^\varepsilon - 1))$ and $\beta = e^{\varepsilon'-\varepsilon}$, we have:

$$D_{e^{\varepsilon'}}(\mu Q^{(\eta)}\|\nu Q^{(\eta)}) \leq (1 - \eta)\Big((1 - \beta)D_{e^\varepsilon}(\mu Q^{(0)}\|\zeta_{\mathrm{mix}}) + \beta D_{e^\varepsilon}(\mu Q^{(0)}\|\nu Q^{(0)})\Big) \tag{5}$$

**Lemma 7.** *Given the measurement distribution of a maximally mixed state $\zeta_{mix}$ and an arbitrary distribution $z \in \mathcal{P}(\mathbb{Z})$, we have:*

$$D_\alpha(z\|\zeta_{mix}) \leq 1 - \alpha\min_k\Big(\frac{\operatorname{Tr}(E_k)}{d}\Big)$$

Based on Lemma 7, we can derive an upper bound on $\mathrm{D}_{e^\varepsilon}(\mu Q^{(0)}\|\zeta_{\mathrm{mix}})$ in terms of the trace values of the POVM elements. Additionally, from the data-processing inequality for the hockey-stick divergence, we have $\mathrm{D}_{e^\varepsilon}(\mu Q^{(0)}\|\nu Q^{(0)}) \leq \mathrm{D}_{e^\varepsilon}(\mu\|\nu) \leq \delta$. Combining these results, we obtain an improved bound for Equation 5:

$$\mathrm{D}_{e^{\varepsilon'}}\left(\mu Q^{(\eta)}\|\nu Q^{(\eta)}\right) \leq (1-\eta)\left(1 - e^{\varepsilon'-\varepsilon}(1-\delta) - (e^\varepsilon - e^{\varepsilon'})\varphi\right)$$

, where $\varphi = \min_k\left(\frac{\mathrm{Tr}(E_k)}{d}\right)$. This result is formalized in Theorem 2, which characterizes how depolarizing noise amplifies the privacy guarantees of the composed mechanism $Q^{(\eta)} \circ A$. Specifically, the mechanism satisfies $(\varepsilon', \delta')$-DP, where $\varepsilon' = \log\left(1 + (1-\eta)(e^\varepsilon - 1)\right)$ and $\delta' = (1-\eta)\left[1 - e^{\varepsilon'-\varepsilon}(1-\delta) - (e^\varepsilon - e^{\varepsilon'})\varphi\right]$.

Theorem 2 reveals that the amplified failure probability $\delta'$ depends on the choice of POVMs. In particular, $\delta'$ becomes tighter as $\varphi = \min_k\left(\frac{\mathrm{Tr}(E_k)}{d}\right)$ increases. This insight leads to Corollary 3, highlighting that $\delta'$ is minimized when all POVM elements $E_k$ have equal trace (i.e., $\mathrm{Tr}(E_k) = \frac{1}{K}$).

Contrarily, $\varepsilon' \leq \varepsilon$ for all $\eta \in [0,1]$, the privacy loss in terms of $\varepsilon$ is always reduced. However, the bound on $\delta$ is only improved (i.e., $\delta' \leq \delta$) when the noise level $\eta$ exceeds the threshold given in Corollary 4. This condition highlights that a sufficient level of quantum noise is required to achieve strict amplification of the privacy guarantee in both parameters.

**Theorem 2** (Amplification on Privacy Loss). *Let $A = f_{par} \circ f_{cdp}$ be $(\varepsilon, \delta)$-DP, and $Q^{(\eta)} = f_{mea} \circ f_{dep}^{(\eta)} \circ f_{enc}$ be a quantum mechanism in a $d$-dimensional Hilbert space where $0 \leq \eta \leq 1$ is the depolarizing noise factor. Then, the composition $Q^{(\eta)} \circ A$ is $(\varepsilon', \delta')$-DP where $\varepsilon' = \log\left(1 + (1-\eta)(e^\varepsilon - 1)\right)$ and $\delta' = (1-\eta)\left(1 - e^{\varepsilon'-\varepsilon}(1-\delta) - (e^\varepsilon - e^{\varepsilon'})\varphi\right)$ with $\varphi = \min_k\left(\frac{\mathrm{Tr}(E_k)}{d}\right)$.*

**Corollary 3.** *Let $\{E_k\}_{k=1}^K$ be the POVM used in $f_{mea}$. Then, the amplified failure probability $\delta'$ in Theorem 2 is minimized when all POVM elements have equal trace (i.e., $\mathrm{Tr}[E_k] = \frac{d}{K}$ for all $k \in \{1, \ldots, K\}$).*

**Corollary 4.** *Given an optimal measurement such that $\mathrm{Tr}[E_k] = \frac{d}{K} \forall k$, the composed mechanism $Q^{(\eta)} \circ A$ strictly improves the privacy guarantee (i.e., $\varepsilon' \leq \varepsilon$ and $\delta' \leq \delta$) if*

$$\eta \geq 1 - \frac{\delta}{(1-\delta)(1 - e^{-\varepsilon}) - (e^\varepsilon - 1)/K}$$

## B.3 Derivation of Theorem 3

Here, we establish a rigorous framework to study the utility loss, defined as the absolute error between the noisy and noise-free versions of our mechanism. The final output of the mechanism is stochastic due to the sampling-based measurement process. Thus, we analyze the difference between the expected values of their output. The expected value represents the average behavior of a mechanism and provides a deterministic quantity that we can use to measure utility loss.

Formally, we define the expectation measurement function $f_{\exp} : \mathcal{H} \to \mathbb{R}$ as:

$$f_{\exp}(\rho) = \sum_k k\,\mathrm{Tr}[E_k \rho] = \mathrm{Tr}\left[\left(\sum_k k E_k\right)\rho\right] = \mathrm{Tr}[E_{\exp}\rho]$$

where $E_{\exp} = \sum_k k E_k$ is the expectation value observable.

Using this function, we define our deterministic expectation mechanisms. The **full mechanism**, including classical and quantum noise, is $\mathcal{M}_{\mathrm{full}}(x) = (f_{\exp} \circ f_{dep}^{(\eta)} \circ f_{enc} \circ f_{par} \circ f_{cdp})(x)$. On the other hand, the **noise-free mechanism (clean)** is $\mathcal{M}_{\mathrm{clean}}(x) = (f_{\exp} \circ f_{enc} \circ f_{par})(x)$. The total utility loss is the worst-case absolute error between their expected outputs:

$$\mathrm{Error} = \sup_{x \in \mathbb{X}} |\mathcal{M}_{\mathrm{full}}(x) - \mathcal{M}_{\mathrm{clean}}(x)|$$

To analyze this error, we introduce an **intermediate mechanism (half)** that includes only quantum noise as $\mathcal{M}_{\mathrm{half}}(x) = (f_{\exp} \circ f_{dep}^{(\eta)} \circ f_{enc} \circ f_{par})(x)$.

**Lemma 8.** *The intermediate mechanism $\mathcal{M}_{half}$ is $L_\infty$-Lipschitz with respect to the input perturbation $\kappa$, satisfying $|\mathcal{M}_{half}(x + \kappa) - \mathcal{M}_{half}(x)| \leq L_\infty \|\kappa\|_\infty$. $L_\infty$ is given by:*

$$L_\infty = 2(1 - \eta)\|E_{exp}\|_{op}\|W\|_\infty \left( \sum_j \|H_j\|_{op} \right)$$

Lemma 8 establishes a bound on the sensitivity of $\mathcal{M}_{half}$ with respect to perturbations in its classical input. We use $\|\cdot\|_p$ to denote the $p$-norm, and $\|\cdot\|_{op}$ to denote the operator norm. The proof leverages the chain rule for Lipschitz continuity, where the overall Lipschitz constant $L_\infty$ is given by the product of the individual constants associated with each component function in the composition, namely, $f_{\exp}$, $f_{\text{dep}}$, $f_{\text{enc}}$, and $f_{\text{par}}$. In addition, we observe that if $\kappa \sim \mathcal{N}(0, \sigma^2 I)$, then $\mathcal{M}_{\text{half}}(x + \kappa)$ is equivalent in distribution to $\mathcal{M}_{\text{full}}(x)$. Thus, this lemma results in a bound on the difference between these two mechanisms.

**Lemma 9.** *The absolute difference between the expected outputs of the intermediate and noise-free mechanisms is uniformly bounded by:*

$$|\mathcal{M}_{half}(x) - \mathcal{M}_{clean}(x)| \leq 2\eta\|E_{exp}\|_{op}$$

Lemma 9 directly bounds the difference between $\mathcal{M}_{\text{half}}$ and $\mathcal{M}_{\text{clean}}$. The proof leverages the Lipschitz property of the function $f_{\exp}$ and the fundamental property that the trace norm difference between any two density matrices is at most 2. Along with the result in Lemma 8, we can establish the bound on the absolute error.

**Theorem 3** (Utility Bound). *Let the classical noise be $\kappa \sim \mathcal{N}(0, \sigma^2 I)$ acting on an input space $\mathbb{X}$ of dimension $d_X = \dim(\mathbb{X})$. For any desired failure probability $p > 0$, the utility loss is bounded probabilistically as:*

$$\Pr\left( Error \leq L_\infty \cdot \sigma \sqrt{2 \ln \frac{2d_X}{p}} + 2\eta\|E_{exp}\|_{op} \right) \geq 1 - p$$

*where $L_\infty = 2(1 - \eta)\|E_{exp}\|_{op}\|W\|_\infty \left( \sum_j \|H_j\|_{op} \right)$.*

Theorem 3 combines the previous results to provide a single utility guarantee. The proof exploits the triangle inequality to additively combine the bounds from Lemmas 8 and 9. As the classical noise is unbounded, the final guarantee is a high-probability statement showing the trade-off between utility loss and the classical ($\sigma$) and quantum ($\eta$) noise level.

### B.4 GENERALIZATION TO OTHER QUANTUM NOISE CHANNELS

In this section, we show how the privacy amplification result of Theorem 1 can be extended to a broad class of quantum noise channels beyond depolarizing noise. First, we identify the essential mechanism responsible for privacy amplification. Then, we illustrate the generalization by analyzing two asymmetric and physically relevant noise processes: the Generalized Amplitude Damping (GAD) channel and the Generalized Dephasing (GD) channel.

#### B.4.1 KEY INSIGHT BEHIND THE GENERALIZATION

Here, first we review the proof trajectory of Theorem 1 presented in Appendix B.1. The analysis begins by decomposing the output distributions of the mechanism on neighboring inputs using Lemma 1 (Lemma 1). It then reduces the divergence analysis to the worst-case pair of orthogonal inputs via Lemma 2 (Lemma 2). Crucially, Lemma 3 (Lemma 3) establishes that the classical hockey-stick divergence of the measurement outcomes is upper-bounded by the quantum hockey-stick divergence of the evolved quantum states.

We can see that in Theorem 1, the privacy amplification is derived from Lemma 4, which establishes the contraction of the quantum hockey-stick divergence $D_\alpha^{(q)}$ under the depolarizing channel. While Theorem 1 utilizes the specific form of $D_{e^\epsilon}^{(q)}$, we discuss that even if we relax the bound to the

standard trace distance $D_1^{(q)}$, the privacy guarantee still holds. Specifically, for any privacy parameter $\alpha \geq 1$ (where $\alpha = e^{\tilde{\varepsilon}}$ in our context), the quantum hockey-stick divergence is upper-bounded by the trace distance divergence:

$$D_\alpha^{(q)}(\rho\|\sigma) = \mathrm{Tr}[(\rho - \alpha\sigma)_+] \leq \mathrm{Tr}[(\rho - \sigma)_+] = D_1^{(q)}(\rho\|\sigma).$$

This inequality holds because subtracting a larger multiple of $\sigma$ (since $\alpha \geq 1$) reduces the positive part of the operator difference.

Then, the insight is that to generalize Theorem 1 to an arbitrary noise channel $\mathcal{E}$, we need to identify its contraction coefficient under the trace distance (or $D_1$ divergence). If a channel $\mathcal{E}$ satisfies a contraction bound $\kappa(\mathcal{E})$ such that:

$$\sup_{\rho \neq \sigma} \frac{D_1(\mathcal{E}(\rho)\|\mathcal{E}(\sigma))}{D_1(\rho\|\sigma)} \leq \kappa(\mathcal{E}),$$

then the composed mechanism naturally satisfies a privacy amplification where the failure probability $\delta$ is scaled by $\kappa(\mathcal{E})$. In the following subsections, we apply this insight to two asymmetric noise channels.

### B.4.2 GENERALIZED AMPLITUDE DAMPING CHANNEL

Generalized Amplitude Damping (GAD) channel is a noise process describing energy exchange between a qubit and its thermal environment. Unlike depolarizing, GAD is inherently asymmetric because it drives the qubit toward a temperature-dependent equilibrium state while simultaneously suppressing quantum coherence. The channel is parameterized by a damping strength $\eta \in [0, 1]$ and an excitation probability $p \in [0, 1]$, where $p = 0$ corresponds to relaxation toward $|0\rangle$, $p = 1$ toward $|1\rangle$, and intermediate values represent nonzero-temperature behavior. This asymmetry makes GAD a realistic noise model for superconducting and trapped-ion devices. To formalize this, we consider the $n$-qubit channel acting on a single designated qubit:

$$f_{\mathrm{GAD}}^{(p,\eta)} = I_{2^{n-1}} \otimes A_{\mathrm{GAD}}^{(p,\eta)}.$$

Despite its non-unital nature, the GAD channel contracts distinguishability between quantum states. Differences in excitation probabilities shrink because all states relax toward the same thermal fixed point, while differences in coherence decay due to energy dissipation. In Lemma 4.1, we construct the contraction coefficient of $f_{\mathrm{GAD}}^{(p,\eta)}$.

**Lemma 4.1.** *Let $A_{GAD}^{(p,\eta)}$ be the generalized amplitude damping (GAD) channel on a single qubit, with damping parameter $\eta \in [0, 1]$ and excitation parameter $p \in [0, 1]$. Define the $n$-qubit channel*

$$f_{\mathrm{GAD}}^{(p,\eta)} := I_{2^{n-1}} \otimes A_{GAD}^{(p,\eta)},$$

*Then the contraction coefficient of $f_{\mathrm{GAD}}^{(p,\eta)}$ satisfies*

$$\kappa(f_{\mathrm{GAD}}^{(p,\eta)}) := \sup_{\rho \neq \sigma} \frac{D_1\left(f_{\mathrm{GAD}}^{(p,\eta)}(\rho) \,\Big\|\, f_{\mathrm{GAD}}^{(p,\eta)}(\sigma)\right)}{D_1(\rho\|\sigma)} \leq 2\sqrt{\eta} - \eta$$

*Proof.* By definition $D_1(\rho\|\sigma) = \frac{1}{2}\|\rho - \sigma\|_1$, so $\kappa(f_{\mathrm{GAD}}^{(p,\eta)})$ is the trace-distance contraction coefficient of the channel $f_{\mathrm{GAD}}^{(p,\eta)} = I_{2^{n-1}} \otimes A_{\mathrm{GAD}}^{(p,\eta)}$:

$$\kappa\left(f_{\mathrm{GAD}}^{(p,\eta)}\right) = \sup_{\rho \neq \sigma} \frac{\|f_{\mathrm{GAD}}^{(p,\eta)}(\rho) - f_{\mathrm{GAD}}^{(p,\eta)}(\sigma)\|_1}{\|\rho - \sigma\|_1}.$$

The supremum is over all $n$-qubit states $\rho, \sigma$, which may be entangled across the ancilla system and the noisy qubit.

Based on Hirche (2024), this is upper-bounded by the complete trace-distance contraction coefficient of the single-qubit channel $A_{\mathrm{GAD}}^{(p,\eta)}$, defined as

$$\eta_{Tr}^c(A_{\mathrm{GAD}}^{(p,\eta)}) := \sup_{k \geq 1} \sup_{\rho \neq \sigma} \frac{\|(I_k \otimes A_{\mathrm{GAD}}^{(p,\eta)})(\rho) - (I_k \otimes A_{\mathrm{GAD}}^{(p,\eta)})(\sigma)\|_1}{\|\rho - \sigma\|_1}.$$

Since $f_{\mathrm{GAD}}^{(p,\eta)}$ is exactly $\mathrm{I}_{2^{n-1}} \otimes A_{\mathrm{GAD}}^{(p,\eta)}$ for one particular ancilla dimension, we have

$$\kappa\big(f_{\mathrm{GAD}}^{(p,\eta)}\big) \leq \eta_{Tr}^c(A_{\mathrm{GAD}}^{(p,\eta)}).$$

Based on the Lemma 9 and Proposition 28 in Hirche (2024), we have:

$$\begin{aligned}
\kappa(f_{\mathrm{GAD}}^{(p,\eta)}) &= \eta_{Tr}^c(A_{\mathrm{GAD}}^{(p,\eta)}) \\
&\leq 1 - \alpha(A_{\mathrm{GAD}}^{(p,\eta)}) \\
&= 1 - (1 - \sqrt{\eta})^2 \\
&= 1 - (1 - 2\sqrt{\eta} + \eta) \\
&= 2\sqrt{\eta} - \eta.
\end{aligned}$$

$\square$

Based on Lemma 4.1, we now derive a privacy amplification result for the GAD channel in Theorem 1.1.

**Theorem 1.1** (Amplification Under Generalized Amplitude Damping Noise). *Let $A : \mathbb{X} \to \mathcal{P}(\mathbb{Y})$ be a classical mechanism satisfying $(\varepsilon, \delta)$-DP where $A = f_{par} \circ f_{cdp}$, and let $Q^{(p,\eta)} : \mathbb{Y} \to \mathcal{P}(\mathbb{Z})$ be a quantum mechanism in $d$-dimensional Hilbert space defined as $Q^{(p,\eta)} = f_{mea} \circ f_{\mathrm{GAD}}^{(p,\eta)} \circ f_{enc}$. Then, the composed mechanism $Q^{(p,\eta)} \circ A$ satisfies $(\varepsilon', \delta')$-DP, where*

$$\varepsilon' = \varepsilon, \quad \delta' = (2\sqrt{\eta} - \eta)\delta.$$

*Proof.* Let $\mu = A(x)$ and $\nu = A(x')$ be the output distributions of the mechanism $A$ on neighboring inputs $x$ and $x'$. We aim to bound the hockey-stick divergence

$$\mathrm{D}_{e^\varepsilon}(\mu Q^{(p,\eta)} \| \nu Q^{(p,\eta)}).$$

By Lemma 1, we can decompose $\mu$ and $\nu$ using a parameter $\theta = \mathrm{D}_{e^\varepsilon}(\mu\|\nu)$ and define auxiliary distributions $\mu'$, $\nu'$, and $\omega$ with $\mu' \perp \nu'$ such that

$$\mu = (1 - \theta)\omega + \theta\mu', \quad \nu = \frac{1 - \theta}{e^\varepsilon}\omega + \left(1 - \frac{1 - \theta}{e^\varepsilon}\right)\nu'.$$

Additionally, define $\tilde{\varepsilon} = \log\left(1 + \frac{e^\varepsilon - 1}{\theta}\right)$.
We now consider the post-processed outputs:

$$\begin{aligned}
& \mathrm{D}_{e^\varepsilon}(\mu Q^{(p,\eta)} \| \nu Q^{(p,\eta)}) \\
& \leq \theta \cdot \mathrm{D}_{e^{\tilde{\varepsilon}}}(\mu' Q^{(p,\eta)} \| \nu' Q^{(p,\eta)}) && \text{(Lemma 1)} \\
& \leq \theta \cdot \sup_{y \neq y'} \mathrm{D}_{e^{\tilde{\varepsilon}}}(Q^{(p,\eta)}(y) \| Q^{(p,\eta)}(y')) && \text{(Lemma 2)} \\
& \leq \theta \cdot \sup_{y \neq y'} \mathrm{D}_{e^{\tilde{\varepsilon}}}^{(q)}\left(f_{\mathrm{GAD}}^{(p,\eta)} \circ f_{enc}(y) \| f_{\mathrm{GAD}}^{(p,\eta)} \circ f_{enc}(y')\right) && \text{(Lemma 3)} \\
& = \theta \cdot \sup_{\rho, \rho'} \mathrm{D}_{e^{\tilde{\varepsilon}}}^{(q)}\left(f_{\mathrm{GAD}}^{(p,\eta)}(\rho) \| f_{\mathrm{GAD}}^{(p,\eta)}(\rho')\right) && \text{(where } \rho, \rho' \text{ are pure)} \\
& \leq \theta \cdot \sup_{\rho, \rho'} \mathrm{D}_1^{(q)}\left(f_{\mathrm{GAD}}^{(p,\eta)}(\rho) \| f_{\mathrm{GAD}}^{(p,\eta)}(\rho')\right) && (e^{\tilde{\varepsilon}} \geq 1) \\
& \leq \theta \cdot \sup_{\rho, \rho'}\left((2\sqrt{\eta} - \eta) \cdot \mathrm{D}_1^{(q)}(\rho\|\rho')\right) && \text{(Lemma 4.1)} \\
& \leq \theta \cdot (2\sqrt{\eta} - \eta) \cdot 1 && \text{(Because } \mathrm{D}_{e^{\tilde{\varepsilon}}}^{(q)}(\rho\|\rho') \leq 1)
\end{aligned}$$

Since the original mechanism $A$ is $(\varepsilon, \delta)$-DP, we have $\theta = \mathrm{D}_{e^\varepsilon}(\mu\|\nu) \leq \delta$. We substitute this into the final bound:

$$\mathrm{D}_{e^\varepsilon}(\mu Q^{(p,\eta)} \| \nu Q^{(p,\eta)}) \leq (2\sqrt{\eta} - \eta) \cdot \delta.$$

This yields the advertised DP parameters. $\square$

### B.4.3 GENERALIZED DEPHASING CHANNEL

Generalized Dephasing (GD) channel is one of the most fundamental and widely studied noise processes in quantum information. It suppresses quantum coherence while leaving classical populations unchanged. Specifically, this channel is formulated as:

$$A_{\text{GD}}^{(\eta)}(\rho) = (1 - \eta)\rho + \eta Z \rho Z$$

where $\eta \in [0, 1]$ is the dephasing parameter and $Z$ is the Pauli-Z operator. From Proposition 33 in Hirche (2024), the complete trace-distance contraction coefficient of a single-qubit GD channel is exactly 1 (i.e., $\eta_{Tr}^c(A_{\text{GD}}^{(\eta)}) = 1$). This implies that no worst-case privacy amplification can be guaranteed under dephasing noise. In other words, for some input states, the noise does not reduce distinguishability at all. However, this worst case is only attained for states distinguished solely through diagonal differences. In many common QML architectures such as those using angle encoding, the encoded data occupies families of states where all information is carried in the off-diagonal components (coherences). In this setting, GD noise does provide nontrivial contraction, and thus we obtain privacy amplification. One instance of this setting is formally presented in Assumption 1.

**Assumption 1** (Product Equatorial Encoding on All Qubits). *For each input $y \in \mathbb{Y}$, the encoder prepares a product state*

$$\rho_y = f_{\text{enc}}(y) = \bigotimes_{j=1}^{n} \rho_y^{(j)},$$

*where each single-qubit factor $\rho_y^{(j)}$ is an equatorial state on the Bloch sphere, i.e.,*

$$\rho_y^{(j)} = \frac{1}{2}\left(I + \cos\phi_y^{(j)} X + \sin\phi_y^{(j)} Y\right),$$

*for some angle $\phi_y^{(j)} \in \mathbb{R}$ and with no Z-component.*

This equatorial–state assumption is satisfied by common QML encoders where data are mapped into phases and superpositions via single-qubit rotations and Hadamard-type preparation such as circuits of the form $H \to R_Z(\phi_y^{(j)})$ on each qubit Schuld & Killoran (2019); Pérez-Salinas et al. (2020); Hatakeyama-Sato et al. (2023). Beyond QML, equatorial states also play a central role in quantum communication and quantum key distribution (QKD), where they are used for phase encoding and coherence-based information transfer Fisher et al. (2014); Xiao et al. (2014). Thus, analyzing privacy amplification of GD channel under this assumption is both realistic and practically meaningful.

We consider the $n$-qubit GD channel acting independently on every qubit as follow:

$$f_{\text{GD}}^{(\eta)} = \bigotimes_{j=1}^{n} A_{\text{GD}}^{(\eta)},$$

We now establish a contraction bound for $f_{\text{GD}}^{(\eta)}$ under Assumption 1 in Lemma 4.2.

**Lemma 4.2.** *Let $f_{\text{GD}}^{(\eta)}$ be the $n$-qubit GD channel defined above, and assume the encoder $f_{\text{enc}}$ satisfies Assumption 1. Then the trace-distance contraction coefficient of $f_{\text{GD}}^{(\eta)}$ over the encoder family $\{\rho_y\}_{y \in \mathbb{Y}}$ satisfies*

$$\kappa\big(f_{\text{GD}}^{(\eta)}\big) := \sup_{y \neq y'} \frac{D_1\big(f_{\text{GD}}^{(\eta)}(\rho_y) \,\big\|\, f_{\text{GD}}^{(\eta)}(\rho_{y'})\big)}{D_1(\rho_y \| \rho_{y'})} \leq |1 - 2\eta|.$$

*Proof.* For each $y$, we have

$$\rho_y = \bigotimes_{j=1}^{n} \rho_y^{(j)}, \qquad \rho_y^{(j)} = \tfrac{1}{2}\big(I + \cos\phi_y^{(j)} X + \sin\phi_y^{(j)} Y\big).$$

Let $\Delta = \rho_y - \rho_{y'}$ for two distinct inputs $y \neq y'$. Expanding $\Delta$ in the $n$-qubit Pauli basis, we have:

$$\Delta = \sum_{P \in \mathcal{P}_n} c_P P,$$

where $\mathcal{P}_n = \{I, X, Y, Z\}^{\otimes n}$ is the $n$-qubit Pauli group, and $c_P \in \mathbb{R}$ since $\Delta$ is Hermitian. Because each single-qubit factor $\rho_y^{(j)}$ contains only $I$, $X$, and $Y$ components and no $Z$ component, any product state $\rho_y = \bigotimes_j \rho_y^{(j)}$ expands only in Pauli strings whose single-qubit factors are in $\{I, X, Y\}$. The same holds for $\rho_{y'}$, and therefore their difference $\Delta = \rho_y - \rho_{y'}$ has no support on any string consisting solely of $I$'s and $Z$'s. In particular,

$$c_P = 0 \quad \text{for all } P \in \mathcal{P}_n \text{ such that } P \in \{I, Z\}^{\otimes n}.$$

Equivalently, every nonzero coefficient $c_P$ corresponds to a Pauli string $P$ that contains at least one factor $X$ or $Y$.

The $n$-qubit GD channel acts diagonally in the Pauli basis:

$$f_{\text{GD}}^{(\eta)}(P) = \lambda_P P,$$

where

$$\lambda_P = \prod_{j=1}^n \lambda_{P_j}, \quad \lambda_I = \lambda_Z = 1, \quad \lambda_X = \lambda_Y = 1 - 2\eta,$$

and $P = P_1 \otimes \cdots \otimes P_n$ with $P_j \in \{I, X, Y, Z\}$. Thus, for any Pauli string $P$ that contains at least one $X$ or $Y$, we have

$$\lambda_P = (1 - 2\eta)^k$$

for $k \geq 1$. As a result, we have:

$$|\lambda_P| \leq |1 - 2\eta|.$$

It implies that:

$$f_{\text{GD}}^{(\eta)}(\Delta) = \sum_{P \in \mathcal{P}_n} c_P \lambda_P P,$$

with each nonzero coefficient satisfying $|\lambda_P| \leq |1 - 2\eta|$. As a linear map on the subspace spanned by Pauli strings with at least one $X$ or $Y$, $f_{\text{GD}}^{(\eta)}$ is diagonal in an orthonormal operator basis with eigenvalues bounded in modulus by $|1 - 2\eta|$. Thus, its operator norm on any unitarily invariant norm, in particular the trace norm, is at most $|1 - 2\eta|$ on this subspace. Concretely,

$$\|f_{\text{GD}}^{(\eta)}(\Delta)\|_1 \leq |1 - 2\eta| \, \|\Delta\|_1.$$

Since $D_1(\rho\|\sigma) = \frac{1}{2}\|\rho - \sigma\|_1$, we conclude that for all $y \neq y'$,

$$\frac{D_1\big(f_{\text{GD}}^{(\eta)}(\rho_y) \,\big\|\, f_{\text{GD}}^{(\eta)}(\rho_{y'})\big)}{D_1(\rho_y\|\rho_{y'})} = \frac{\frac{1}{2}\|f_{\text{GD}}^{(\eta)}(\Delta)\|_1}{\frac{1}{2}\|\Delta\|_1} \leq |1 - 2\eta|.$$

Taking the supremum over all $y \neq y'$ gives the desired bound. $\qquad\square$

Finally, similar to the amplification analysis for depolarizing noise and GAD noise, we now derive a privacy amplification theorem for the GD channel acting on all qubits. The result follows immediately by combining the contraction bound in Lemma 4.2 with the classical post-processing and distribution-decomposition tools used earlier.

**Theorem 1.2.** *Let $A : \mathbb{X} \to \mathcal{P}(\mathbb{Y})$ be a classical mechanism satisfying $(\varepsilon, \delta)$-DP, and let*

$$Q^{(\eta)} := f_{\text{mea}} \circ f_{\text{GD}}^{(\eta)} \circ f_{\text{enc}}$$

*be an $n$-qubit quantum mechanism where $f_{\text{GD}}^{(\eta)}$ is the $n$-qubit GD channel defined above and $f_{\text{enc}}$ satisfies Assumption 1. Then the composed mechanism $Q^{(\eta)} \circ A$ satisfies $(\varepsilon', \delta')$-DP with*

$$\varepsilon' = \varepsilon, \qquad \delta' = |1 - 2\eta| \cdot \delta.$$

*Proof.* Let $\mu = A(x)$ and $\nu = A(x')$ be the output distributions of $A$ on neighboring inputs $x, x'$. As in Theorem 1.1, we apply Lemma 1 to decompose $\mu, \nu$ with parameter $\theta = D_{e^\varepsilon}(\mu\|\nu) \leq \delta$ and reduce the analysis to the worst-case pair of orthogonal inputs. Using Lemma 2 and Lemma 3, we can bound

$$D_{e^\varepsilon}(\mu Q^{(\eta)}\|\nu Q^{(\eta)}) \leq \theta \cdot \sup_{\rho \neq \rho'} D_1\big(f_{\text{GD}}^{(\eta)}(\rho) \,\big\|\, f_{\text{GD}}^{(\eta)}(\rho')\big),$$

where the supremum is taken over encoded states $\rho$, $\rho'$ in the image of $f_{\text{enc}}$.

By Lemma 4.2,

$$D_1\big(f_{\text{GD}}^{(\eta)}(\rho) \,\|\, f_{\text{GD}}^{(\eta)}(\rho')\big) \leq |1 - 2\eta| \, D_1(\rho\|\rho') \leq |1 - 2\eta|.$$

Therefore,

$$D_{e^\varepsilon}(\mu Q^{(\eta)}\|\nu Q^{(\eta)}) \leq \theta\,|1 - 2\eta| \leq |1 - 2\eta|\,\delta,$$

which yields the claimed privacy parameters $\varepsilon' = \varepsilon$ and $\delta' = |1 - 2\eta|\delta$. $\qquad\square$

## B.5 PROOFS

**Lemma 1.** *Let $\mu$ and $\nu$ be probability distributions such that $D_{e^\varepsilon}(\mu\|\nu) \leq \delta$, and define $\theta = D_{e^\varepsilon}(\mu\|\nu)$. Then, there exist distributions $\mu'$, $\nu'$, and $\omega$, along with a parameter $\tilde{\varepsilon} := \log\big(1 + \frac{e^\varepsilon - 1}{\theta}\big)$ such that:*

$$\mu = (1 - \theta)\omega + \theta\mu', \quad \nu = \frac{1 - \theta}{e^\varepsilon}\omega + \left(1 - \frac{1 - \theta}{e^\varepsilon}\right)\nu',$$

*with disjoint distributions: $\mu' \perp \nu'$. Then, the following bound holds:*

$$D_{e^\varepsilon}(\mu\|\nu) = \theta \cdot D_{e^{\tilde{\varepsilon}}}(\mu'\|\nu').$$

*Proof.* Studied in (Balle et al., 2019a) $\qquad\square$

**Lemma 2.** *Given a post-process mechanism $Q$, we have:*

$$\sup_{\mu\perp\nu} D_\varepsilon(\mu Q\|\nu Q) \leq \sup_{y\neq y'} D_\varepsilon(Q(y)\|Q(y')).$$

*Proof.* Studied in (Balle et al., 2019a) $\qquad\square$

**Lemma 3.** *Given a measurement $E = \{E_i\}$ with $\sum_i E_i = I$, and two quantum states $\rho$ and $\rho'$, the classical hockey-stick divergence of the resulting probability distributions is less than or equal to the quantum hockey-stick divergence between the states.*

$$D_\alpha(P \,\|\, P') \leq D_\alpha^{(q)}(\rho \,\|\, \rho')$$

*Proof.* The quantum hockey-stick divergence is defined as:

$$D_\alpha^{(q)}(\rho \,\|\, \rho') = \text{Tr}\big[(\rho - \alpha\rho')_+\big],$$

where $A_+$ denotes the positive part of a Hermitian operator $A$. Let us define the operator $A = \rho - \alpha\rho'$.

Applying measurement $E$ to $\rho$ and $\rho'$ yields probability distributions with elements:

$$P(i) = \text{Tr}(E_i\rho), \quad P'(i) = \text{Tr}(E_i\rho').$$

The classical hockey-stick divergence is defined as:

$$D_\alpha(P \,\|\, P') = \sum_i [P(i) - \alpha P'(i)]_+,$$

where $[x]_+ = \max(x, 0)$.

We begin the proof from the definition of the classical divergence:

$$D_\alpha(P \parallel P') = \sum_i \max\left(0, \mathrm{Tr}(E_i\rho) - \alpha\mathrm{Tr}(E_i\rho')\right)$$

$$= \sum_i \max\left(0, \mathrm{Tr}\left(E_i(\rho - \alpha\rho')\right)\right)$$

$$= \sum_i \max\left(0, \mathrm{Tr}(E_iA)\right)$$

For any positive semi-definite operator $E_i$ and any Hermitian operator $A$, it holds that $\mathrm{Tr}(E_iA) \leq \mathrm{Tr}(E_iA_+)$. Since $A_+$ is a positive semi-definite operator, $\mathrm{Tr}(E_iA_+)$ is non-negative. Therefore, we can conclude that $\max(0, \mathrm{Tr}(E_iA)) \leq \mathrm{Tr}(E_iA_+)$.

Applying this inequality to our expression, we get:

$$D_\alpha(P \parallel P') \leq \sum_i \mathrm{Tr}(E_iA_+)$$

$$= \mathrm{Tr}\left(\sum_i E_iA_+\right)$$

$$= \mathrm{Tr}\left(\left(\sum_i E_i\right)A_+\right)$$

$$= \mathrm{Tr}(I \cdot A_+)$$

$$= D_\alpha^{(q)}(\rho \parallel \rho').$$

$\square$

**Lemma 4.** *Given a depolarizing channel $f_{dep}^{(\eta)}(\rho) = \eta\frac{I}{d} + (1-\eta)\rho$, for $\eta \in [0,1]$ and $\alpha \geq 1$, we have:*

$$D_\alpha^{(q)}(f_{dep}^{(\eta)}(\rho) \parallel f_{dep}^{(\eta)}(\rho'))$$

$$\leq \max\left\{0, (1-\alpha)\frac{\eta}{d} + (1-\eta)D_\alpha^{(q)}(\rho \parallel \rho')\right\}$$

*Proof.* Define the operator:

$$U = f_{dep}^{(\eta)}(\rho) - \alpha f_{dep}^{(\eta)}(\rho) = (1-\eta)(\rho - \alpha\rho') + \eta(1-\alpha)\frac{I}{d}.$$

Then:

$$\mathrm{D}_\alpha^{(q)}(f_{dep}^{(\eta)}(\rho) \parallel f_{dep}^{(\eta)}(\rho')) = \mathrm{Tr}[U_+],$$

where $U_+$ denotes the positive part of $U$.

Let $P_+$ be the projector onto the positive eigenspace of $U$. Since $\mathrm{D}_\alpha^{(q)}(f_{dep}^{(\eta)}(\rho) \parallel f_{dep}^{(\eta)}(\rho') > 0$, we have $\mathrm{Tr}[P_+] \geq 1$. Then:

$$\mathrm{Tr}[U_+] = \mathrm{Tr}[P_+U]$$

$$= (1-\eta)\mathrm{Tr}[P_+(\rho - \alpha\rho')] + (1-\alpha)\frac{\eta}{d}\mathrm{Tr}[P_+]$$

$$\leq (1-\eta)\mathrm{D}_\alpha^{(q)}(\rho\|\rho') + (1-\alpha)\frac{\eta}{d},$$

since $\mathrm{Tr}[P_+] \geq 1$ and $1 - \alpha \leq 0$.

$\square$

**Theorem 1** (Amplification on Failure Probability). *Let $A : \mathbb{X} \to \mathcal{P}(\mathbb{Y})$ be a classical mechanism satisfying $(\varepsilon, \delta)$-DP where $A = f_{par} \circ f_{cdp}$, and let $Q^{(\eta)} : \mathbb{Y} \to \mathcal{P}(\mathbb{Z})$ be a quantum mechanism in $d$-dimensional Hilbert space defined as $Q^{(\eta)} = f_{mea} \circ f_{dep}^{(\eta)} \circ f_{enc}$ where $0 \leq \eta \leq 1$ is the depolarizing noise factor. Then, the composed mechanism $Q^{(\eta)} \circ A$ satisfies $(\varepsilon', \delta')$-DP, where*

$$\varepsilon' = \varepsilon, \quad \delta' = \left[\frac{\eta(1 - e^\varepsilon)}{d} + (1-\eta)\delta\right]_+.$$

*Proof.* Let $\mu = A(x)$ and $\nu = A(x')$ be the output distributions of the mechanism $A$ on neighboring inputs $x$ and $x'$. We aim to bound the hockey-stick divergence

$$D_{e^\varepsilon}(\mu Q^{(\eta)} \| \nu Q^{(\eta)}).$$

By Lemma 1, we can decompose $\mu$ and $\nu$ using a parameter $\theta = D_{e^\varepsilon}(\mu\|\nu)$ and define auxiliary distributions $\mu'$, $\nu'$, and $\omega$ with $\mu' \perp \nu'$ such that

$$\mu = (1-\theta)\omega + \theta\mu', \quad \nu = \frac{1-\theta}{e^\varepsilon}\omega + \left(1 - \frac{1-\theta}{e^\varepsilon}\right)\nu'.$$

Additionally, define $\tilde{\varepsilon} = \log\left(1 + \frac{e^\varepsilon - 1}{\theta}\right)$. By Lemma 1, it follows that

$$D_{e^\varepsilon}(\mu\|\nu) \le \theta \cdot D_{e^{\tilde{\varepsilon}}}(\mu'\|\nu').$$

We now consider the post-processed outputs:

$$D_{e^\varepsilon}(\mu Q^{(\eta)} \| \nu Q^{(\eta)})$$
$$\le \theta \cdot D_{e^{\tilde{\varepsilon}}}(\mu' Q^{(\eta)} \| \nu' Q^{(\eta)})$$
$$\le \theta \cdot \sup_{y \ne y'} D_{e^{\tilde{\varepsilon}}}(Q^{(\eta)}(y) \| Q^{(\eta)}(y')) \quad \text{(Lemma 2)}$$
$$\le \theta \cdot \sup_{y \ne y'} D_{e^{\tilde{\varepsilon}}}^{(q)}\left(f_{\text{dep}}^{(\eta)} \circ f_{\text{enc}}(y) \,\|\, f_{\text{dep}}^{(\eta)} \circ f_{\text{enc}}(y')\right) \quad \text{(Lemma 3)}$$
$$= \theta \cdot \sup_{\rho, \rho'} D_{e^{\tilde{\varepsilon}}}^{(q)}\left(f_{\text{dep}}^{(\eta)}(\rho) \,\|\, f_{\text{dep}}^{(\eta)}(\rho')\right)$$
$$\le \theta \cdot \max\left\{0, \frac{\eta(1-e^{\tilde{\varepsilon}})}{d} + (1-\eta) \cdot D_{e^{\tilde{\varepsilon}}}^{(q)}(\rho\|\rho')\right\} \quad \text{(Lemma 4)}$$
$$\le \max\left\{0, \frac{\theta\eta(1-e^{\tilde{\varepsilon}})}{d} + \theta(1-\eta)\right\} \quad \text{(Because } D_{e^{\tilde{\varepsilon}}}^{(q)}(\rho\|\rho') \le 1\text{)}$$

Recall that $e^{\tilde{\varepsilon}} = 1 + \frac{e^\varepsilon - 1}{\theta}$, we substitute this into the expression:

$$\frac{\theta\eta(1-e^{\tilde{\varepsilon}})}{d} = \frac{\theta\eta}{d}\left(1 - \left(1 + \frac{e^\varepsilon - 1}{\theta}\right)\right)$$
$$= \frac{\theta\eta}{d}\left(-\frac{e^\varepsilon - 1}{\theta}\right)$$
$$= \frac{\eta(1-e^\varepsilon)}{d}$$

Additionally, since the original mechanism $A$ is $(\varepsilon, \delta)$-DP, we have $\theta = D_{e^\varepsilon}(\mu\|\nu) \le \delta$. Because $1 - \eta \ge 0$, we have the final result:

$$D_{e^\varepsilon}(\mu Q^{(\eta)} \| \nu Q^{(\eta)}) \le \left[\frac{\eta(1-e^\varepsilon)}{d} + (1-\eta)\delta\right]_+$$

$\square$

**Corollary 1.** *The composed mechanism $Q^{(\eta)} \circ A$ satisfies $(\varepsilon, \delta')$-DP with $\delta' < \delta$, thus strictly amplifying the failure probability.*

*Proof.* The goal is to show that $\delta' < \delta$ for any non-trivial case where quantum post-processing is active ($\eta > 0$). From Theorem 1, we have:

$$\delta' = \left[\frac{\eta(1-e^\varepsilon)}{d} + (1-\eta)\delta\right]_+$$

Let the first term be $C = \frac{\eta(1-e^\varepsilon)}{d}$. Since $\eta > 0$, $d \ge 2$, and $\varepsilon > 0$, we have $C \le 0$. Since $C$ is strictly negative, $C + (1-\eta)\delta < (1-\eta)\delta \le \delta$. Thus, $\delta' < \delta$.

$\square$

**Corollary 2.** *The composed mechanism $Q^{(\eta)} \circ A$ is certifiably robust against adversarial perturbations for an input $x \in \mathbb{X}$ if the following condition holds for the correct class $k$:*

$$\mathbb{E}[[(Q^{(\eta)} \circ A)(x)]_k] > e^{2\varepsilon} \max_{i \neq k} \mathbb{E}[[(Q^{(\eta)} \circ A)(x)]_i] + (1 + e^{\varepsilon})\delta'$$

*Proof.* Studied in (Lecuyer et al., 2019). □

**Lemma 5** (Advanced Joint Convexity). *Let $\mu, \mu'$ be probability distributions such that*

$$\mu = (1 - \sigma)\mu_0 + \sigma\mu_1, \quad \mu' = (1 - \sigma)\mu_0 + \sigma\mu_1',$$

*for some $\sigma \in [0, 1]$, and distributions $\mu_0, \mu_1, \mu_1'$. Given $\alpha \geq 1$, define $\alpha' = 1 + \sigma(\alpha - 1), \quad \beta = \frac{\alpha'}{\alpha}$. Then the following inequality holds:*

$$D_{\alpha'}(\mu\|\mu') \leq (1 - \beta)\sigma D_\alpha(\mu_1\|\mu_0) + \beta\sigma D_\alpha(\mu_1\|\mu_1').$$

*Proof.* Studied in (Balle et al., 2018) □

**Lemma 6.** *Let $\rho$ be a density matrix on a $D$-dimensional Hilbert space, and let*

$$\rho' = f_{dep}(\rho) = \eta\frac{I}{d} + (1 - \eta)\rho$$

*be its depolarized version, where $0 \leq \eta \leq 1$. Let $\{E_k\}_{k=1}^K$ be a POVM satisfying $\sum_k E_k = I$. Then, the measurement probabilities satisfy:*

$$\zeta'(k) = \frac{\eta}{d}\text{Tr}(E_k) + (1 - \eta)\zeta(k),$$

*where $\zeta' = f_{mea}(\rho')$ and $\zeta = f_{mea}(\rho)$ with $\zeta', \zeta \in \mathcal{P}(\mathbb{Z})$.*

*Proof.* By linearity of the trace operator,

$$\zeta'(k) = \text{Tr}(E_k\rho')$$
$$= \text{Tr}\left(E_k\left(\eta\frac{I}{d} + (1 - \eta)\rho\right)\right)$$
$$= \eta\,\text{Tr}\left(E_k\frac{I}{d}\right) + (1 - \eta)\text{Tr}(E_k\rho)$$
$$= \frac{\eta}{d}\text{Tr}(E_k) + (1 - \eta)\zeta_k.$$

□

**Lemma 7.** *Given the measurement distribution of a maximally mixed state $\zeta_{mix}$ and an arbitrary distribution $z \in \mathcal{P}(\mathbb{Z})$, we have:*

$$D_\alpha(z\|\zeta_{mix}) \leq 1 - \alpha\min_k(\frac{\text{Tr}(E_k)}{d})$$

*Proof.* Recall the definition of the hockey-stick divergence:

$$\text{D}_\alpha(z\|\zeta_{\text{mix}}) = \sum_k [z(k) - \alpha\zeta_{\text{mix}}(k)]_+,$$

where $[x]_+ = \max\{x, 0\}$. Since $\zeta_{\text{mix}}(k) = \frac{\text{Tr}(E_k)}{d} \geq \varphi = \min_k\left(\frac{\text{Tr}(E_k)}{d}\right)$, we have

$$[z(k) - \alpha\zeta_{\text{mix}}(k)]_+ \leq [z(k) - \alpha\varphi]_+.$$

Summing over $k$ yields

$$\mathrm{D}_\alpha(z\|\zeta_{\mathrm{mix}}) \leq \sum_k [z(k) - \alpha\varphi]_+.$$

Since $\sum_k z(k) = 1$, it follows that

$$\sum_k [z(k) - \alpha\varphi]_+ \leq 1 - \alpha\varphi.$$

Therefore,

$$\mathrm{D}_\alpha(z\|\zeta_{\mathrm{mix}}) \leq 1 - \alpha\min_k\left(\frac{\mathrm{Tr}(E_k)}{d}\right).$$

$\square$

**Theorem 2** (Amplification on Privacy Loss). *Let $A = f_{par}\circ f_{cdp}$ be $(\varepsilon, \delta)$-DP, and $Q^{(\eta)} = f_{mea}\circ f_{dep}^{(\eta)}\circ f_{enc}$ be a quantum mechanism in $d$-dimensional Hilbert space where $0 \leq \eta \leq 1$ is the depolarizing noise factor. Then, the composition $Q^{(\eta)}\circ A$ is $(\varepsilon', \delta')$-DP where $\varepsilon' = \log\big(1 + (1-\eta)(e^\varepsilon - 1)\big)$ and $\delta' = (1-\eta)\big(1 - e^{\varepsilon'-\varepsilon}(1-\delta) - (e^\varepsilon - e^{\varepsilon'})\varphi\big)$ with $\varphi = \min_k\left(\frac{\mathrm{Tr}(E_k)}{d}\right)$.*

*Proof.* Let $x, x' \in \mathbb{X}$ be neighboring inputs, i.e., $x \simeq x'$. Let $\mu = A(x)$ and $\nu = A(x')$ denote the output distributions of $A$. From the definition, we have $Q^{(0)}$ and $Q^{(1)}$ which are the mechanisms without noise and with full noise. We can see that $Q^{(1)}$ is a constant mechanism because the output of $Q^{(1)}$ is always the measurement of a maximally mixed state, i.e., $Q^{(1)}(y)(k) = \frac{\mathrm{Tr}(E_k)}{d}$ with $\forall y \in \mathbb{Y}$. Based on Lemma 6, we have:

$$Q^{(\eta)}(y) = \eta Q^{(1)}(y) + (1-\eta)Q^{(0)}(y), \forall y \in \mathbb{Y}$$

. Thus, we can write $Q^{(\eta)}$ as a mixture of $Q^{(0)}$ and $Q^{(1)}$ where $Q^{(\eta)} = \eta Q^{(1)} + (1-\eta)Q^{(0)}$.

Let the constant output of $Q^{(1)}$ be $\zeta_{\mathrm{mix}}$. Based on the advanced joint convexity theorem in (Balle et al., 2018), given $\varepsilon' = \log\big(1 + (1-\eta)(e^\varepsilon - 1)\big)$, we have:

$$\begin{aligned}
&\mathrm{D}_{e^{\varepsilon'}}\big(\mu Q^{(\eta)}\|\nu Q^{(\eta)}\big) \\
&= \mathrm{D}_{e^{\varepsilon'}}\big(\eta\mu Q^{(1)} + (1-\eta)\mu Q^{(0)}\|\eta\nu Q^{(1)} + (1-\eta)\nu Q^{(0)}\big) \\
&= \mathrm{D}_{e^{\varepsilon'}}\big(\eta\zeta_{\mathrm{mix}} + (1-\eta)\mu Q^{(0)}\|\eta\zeta_{\mathrm{mix}} + (1-\eta)\nu Q^{(0)}\big) \\
&= (1-\eta)\mathrm{D}_{e^\varepsilon}\big(\mu Q^{(0)}\|(1-\beta)\zeta_{\mathrm{mix}} + \beta\nu Q^{(0)}\big) \\
&\quad \text{(Based on the advanced joint convexity theorem, } \beta = e^{\varepsilon'-\varepsilon}) \\
&\leq (1-\eta)\bigg((1-\beta)\mathrm{D}_{e^\varepsilon}(\mu Q^{(0)}\|\zeta_{\mathrm{mix}}) + \beta\mathrm{D}_{e^\varepsilon}(\mu Q^{(0)}\|\nu Q^{(0)})\bigg)
\end{aligned}$$

We have $\mathrm{D}_{e^\varepsilon}(\mu Q^{(0)}\|\zeta_{\mathrm{mix}}) \leq 1 - e^\varepsilon\min_k\left(\frac{\mathrm{Tr}(E_k)}{d}\right) = 1 - e^\varepsilon\varphi$ and $\mathrm{D}_{e^\varepsilon}(\mu Q^{(0)}\|\nu Q^{(0)}) \leq \mathrm{D}_{e^\varepsilon}(\mu\|\nu) \leq \delta$. Thus, we can conclude:

$$\mathrm{D}_{e^{\varepsilon'}}\big(\mu Q^{(\eta)}\|\nu Q^{(\eta)}\big) \leq (1-\eta)\big(1 - e^{\varepsilon'-\varepsilon}(1-\delta) - (e^\varepsilon - e^{\varepsilon'})\varphi\big)$$

$\square$

**Corollary 3.** *Let $\{E_k\}_{k=1}^K$ be the POVM used in $f_{mea}$. Then, the amplified failure probability $\delta'$ in Theorem 2 is minimized when all POVM elements have equal trace, i.e., $\mathrm{Tr}(E_k) = \frac{d}{K}$ for all $k \in \{1, \ldots, K\}$.*

*Proof.* The goal is to minimize the amplified failure probability $\delta'$ with respect to the choice of the POVM $\{E_k\}_{k=1}^K$. From Theorem 2, the expression for $\delta'$ is:

$$\delta' = (1-\eta)\big(1 - e^{\varepsilon'-\varepsilon}(1-\delta) - (e^\varepsilon - e^{\varepsilon'})\varphi\big)$$

All terms in this expression are independent of the specific measurement choice except for $\varphi = \min_k\left(\frac{\mathrm{Tr}(E_k)}{d}\right)$.

To analyze how $\delta'$ depends on $\varphi$, we examine the sign of $-(1-\eta)(e^\varepsilon - e^{\varepsilon'})$. Since $\eta \in [0,1]$ and $\varepsilon' \leq \varepsilon$, this coefficient is non-positive. Thus, $\delta'$ is a monotonically decreasing function of $\varphi$.

Therefore, to minimize $\delta'$, we must maximize $\varphi$. This is equivalent to maximizing $\min_k(\text{Tr}(E_k))$ subject to the POVM completeness constraint $\sum_{k=1}^{K} E_k = I$. Taking the trace of the completeness relation gives:

$$\sum_{k=1}^{K} \text{Tr}(E_k) = \text{Tr}(I) = d$$

The function $\min_k(\text{Tr}[E_k])$ is maximized when all $\text{Tr}[E_k]$ are equal. Thus, the optimal choice is to have $\text{Tr}[E_k] = d/K$ for all $k$. $\qquad\square$

**Corollary 4.** *Given an optimal measurement such that* $\text{Tr}[E_k] = \frac{d}{K} \forall k$, *the composed mechanism* $Q^{(\eta)} \circ A$ *strictly improves the privacy guarantee—i.e.,* $\varepsilon' \leq \varepsilon$ *and* $\delta' \leq \delta$—*if*

$$\eta \geq 1 - \frac{\delta}{(1-\delta)(1-e^{-\varepsilon}) - (e^\varepsilon - 1)/K}$$

*Proof.* We find the condition on $\eta$ that ensures $\delta' \leq \delta$ under the assumption of an optimal measurement, where, from Corollary 3, we have $\varphi = 1/K$. The guarantee $\varepsilon' \leq \varepsilon$ holds for all $\eta \in [0,1]$.

We start with the inequality $\delta' \leq \delta$ using the expression from Theorem 2:

$$(1-\eta)\big(1 - e^{\varepsilon'-\varepsilon}(1-\delta) - (e^\varepsilon - e^{\varepsilon'})\varphi\big) \leq \delta$$

Substitute the identities $e^{\varepsilon'-\varepsilon} = 1 - \eta + \eta e^{-\varepsilon}$, $e^\varepsilon - e^{\varepsilon'} = \eta(e^\varepsilon - 1)$, and $\varphi = 1/K$, we have:

$$(1-\eta)\left(1 - (1-\eta+\eta e^{-\varepsilon})(1-\delta) - \frac{\eta(e^\varepsilon - 1)}{K}\right) \leq \delta$$

The expression inside the main brackets simplifies to $\delta + \eta(1-\delta)(1-e^{-\varepsilon}) - \frac{\eta(e^\varepsilon - 1)}{K}$. Substituting this back, expanding, and simplifying for $\eta > 0$, we have:

$$(1-\delta)(1-e^{-\varepsilon}) - \frac{e^\varepsilon - 1}{K} - \delta \leq \eta\left((1-\delta)(1-e^{-\varepsilon}) - \frac{e^\varepsilon - 1}{K}\right)$$

Solving for $\eta$ gives the threshold:

$$\eta \geq \frac{(1-\delta)(1-e^{-\varepsilon}) - (e^\varepsilon - 1)/K - \delta}{(1-\delta)(1-e^{-\varepsilon}) - (e^\varepsilon - 1)/K}$$

$$= 1 - \frac{\delta}{(1-\delta)(1-e^{-\varepsilon}) - (e^\varepsilon - 1)/K}$$

$\qquad\square$

**Lemma 8.** *The intermediate mechanism* $\mathcal{M}_{half}$ *is* $L_\infty$-*Lipschitz with respect to the input perturbation* $\kappa$, *satisfying* $|\mathcal{M}_{half}(x+\kappa) - \mathcal{M}_{half}(x)| \leq L_\infty \|\kappa\|_\infty$. *The constant is given by:*

$$L_\infty = 2(1-\eta)\|E_{exp}\|_{op}\|W\|_\infty \left(\sum_j \|H_j\|_{op}\right)$$

*where* $E_{exp} = \sum_k k E_k$.

*Proof.* First, we prove that a Lipschitz bound for a composition of functions can be obtained as the product of their individual Lipschitz constants. Specifically, suppose that $f$ can be written as

$$f = f_1 \circ f_2 \circ \cdots \circ f_h,$$

where $\circ$ denotes function composition, and each $f_i$ admits a Lipschitz constant $L_i$ for $i = 1, \ldots, h$. Then, for any inputs $x$ and a small deviation $\kappa$, it holds that

$$
\begin{aligned}
&\|f(x + \kappa) - f(x)\| \\
&\leq L_1 \|f_2 \circ \cdots \circ f_h(x + \kappa) - f_2 \circ \cdots \circ f_h(x)\| \\
&\leq L_1 L_2 \|f_3 \circ \cdots \circ f_h(x + \kappa) - f_3 \circ \cdots \circ f_h(x)\| \\
&\quad\vdots \\
&\leq \left( \prod_{i=1}^{h} L_i \right) \|\kappa\|.
\end{aligned}
$$

Since the mechanism $\mathcal{M}_{\text{half}}$ is expressed as a composition of $f_{\exp}$, $f_{\text{dep}}^{\eta}$, $f_{\text{enc}}$, and $f_{\text{par}}$, our goal is to determine the Lipschitz bound for each individual function.

**Lipschitz bound of $f_{\text{par}}$:**

The function $f_{\text{par}} : \mathbb{X} \to \mathbb{Y}$ is defined as

$$
f_{\text{par}}(x) = Wx + b,
$$

'Since $b$ is a constant shift (which does not affect Lipschitz continuity), we have:

$$
\|f_{\text{par}}(x + \kappa) - f_{\text{par}}(x)\| = \|W\kappa\| \leq \|W\|_\infty \|\kappa\|_\infty,
$$

Thus, $f_{\text{par}}$ is $\|W\|_\infty$-Lipschitz.

**Lipschitz bound of $f_{\text{enc}}$:**

The function $f_{\text{enc}} : \mathbb{Y} \to \mathcal{H}$ encodes a classical vector $y$ into a density matrix $f_{\text{enc}}(y) = U_{\text{enc}}(y)|0\rangle\langle 0|U_{\text{enc}}(y)^\dagger$ where $U_{\text{enc}}(y) = \prod_{j=1}^{N} e^{-i(\mathbf{w}_j \cdot y_j + b_j)H_j}$. We need to bound the trace norm distance $f_{\text{enc}}(y + \kappa) - f_{\text{enc}}(y)$ in terms of $\|\kappa\|_\infty$.

$$
\begin{aligned}
&\|f_{\text{enc}}(y + \kappa) - f_{\text{enc}}(y)\| \\
&= \|U_{\text{enc}}(y + \kappa)\rho_0 U_{\text{enc}}(y + \kappa)^\dagger - U_{\text{enc}}(y)\rho_0 U_{\text{enc}}(y)^\dagger\| \\
&\leq 2\|U_{\text{enc}}(y + \kappa) - U_{\text{enc}}(y)\|
\end{aligned}
$$

where $\rho_0 = |0\rangle\langle 0|$ and we used the triangle inequality and properties of the trace norm. The difference between the unitary operators is bounded by:

$$
\begin{aligned}
&\|U_{\text{enc}}(y + \kappa) - U_{\text{enc}}(y)\| \\
&\leq \sum_{j=1}^{N} \|e^{-i(y_j + \kappa_j)H_j} - e^{-iy_j H_j}\| \\
&\leq \sum_{j=1}^{N} |\kappa_j| \|H_j\| \leq \sum_{j=1}^{N} \|H_j\| \|\kappa\|_\infty
\end{aligned}
$$

(Based on (Berberich et al., 2024))

Thus, $f_{\text{enc}}$ is $2\left(\sum_{j=1}^{n} \|H_j\|\right)$-Lipschitz.

**Lipschitz bound of $f_{\text{dep}}^{(\eta)}$:**

The function $f_{\text{dep}}^{(\eta)} : \mathcal{H} \to \mathcal{H}$ models the depolarizing noise:

$$
f_{\text{dep}}^{(\eta)}(\rho) = (1 - \eta)\rho + \eta\frac{I}{d},
$$

where $I$ is the identity matrix and $d$ is the dimension of the Hilbert space.

Since the term $\eta\frac{I}{d}$ is constant, the difference between two outputs is:

$$\|f_{\text{dep}}^{(\eta)}(\rho) - f_{\text{dep}}^{(\eta)}(\sigma)\| = (1-\eta)\|\rho - \sigma\|.$$

Thus, $f_{\text{dep}}^{(\eta)}$ is $(1-\eta)$-Lipschitz.

**Lipschitz bound of $f_{\text{exp}}$:**

The measurement function $f_{\text{mea}} : \mathcal{H} \to \mathbb{R}^K$, defined by a set of POVMs $\{E_k\}$, maps a quantum state $\rho$ to a probability vector:

$$f_{\text{exp}}(\rho) = \sum_k k\,\text{Tr}(E_k\rho).$$

Given $E_{\text{exp}} = \sum_k kE_k$, by trace duality and Hölder's inequality, we have:

$$\|f_{\text{exp}}(\rho) - f_{\text{exp}}(\rho')\| = \left|\text{Tr}(E_{\text{exp}}(\rho - \rho'))\right| \leq \|E_{\text{exp}}\|_{op}\|\rho - \rho'\|.$$

Therefore, $f_{\text{exp}}$ is $\|E_{\text{exp}}\|_{op}$-Lipschitz.

As a result, the mechanism $\mathcal{M}_{\text{half}}$ is $L_\infty$-Lipschitz where $L_\infty = 2(1-\eta)\|E_{\text{exp}}\|_{op}\|W\|_\infty\left(\sum_j \|H_j\|_{op}\right)$.

$\square$

**Lemma 9.** *The absolute difference between the expected outputs of the intermediate and clean mechanisms is uniformly bounded by:*

$$\sup_{x\in\mathbb{X}} |\mathcal{M}_{half}(x) - \mathcal{M}_{clean}(x)| \leq 2\eta\|E_{exp}\|$$

*Proof.* Let $\rho(x) = (f_{\text{enc}} \circ f_{\text{par}})(x)$ be the clean quantum state.

$$|\mathcal{M}_{\text{half}}(x) - \mathcal{M}_{\text{clean}}(x)|$$
$$= \left|\text{Tr}[f_{\text{exp}} \cdot f_{\text{dep}}^\eta(\rho(x))) - \text{Tr}(f_{\text{exp}} \cdot \rho(x))\right|$$
$$\leq \|E_{\text{exp}}\|_{op} \cdot \|f_{\text{dep}}^\eta(\rho(x)) - \rho(x)\| \quad \text{(Lipschitz property of } f_{\text{exp}})$$

The trace distance term is bounded as:

$$\|f_{\text{dep}}^\eta(\rho) - \rho\| = \|((1-\eta)\rho + \eta\frac{I}{d}) - \rho\| = \eta\left\|\frac{I}{d} - \rho\right\|$$

Since $\rho$ and $I/d$ are both valid density matrices, the trace distance between them is at most 2. Thus, $\|\frac{I}{d} - \rho\| \leq 2$. Substituting this back gives the final bound of $2\eta\|E_{\text{exp}}\|_{op}$. $\square$

**Theorem 3** (Utility bound). *Let the classical noise be $\kappa \sim \mathcal{N}(0, \sigma^2 I)$ acting on an input space $\mathbb{X}$ of dimension $d_X = \dim(\mathbb{X})$. For any desired failure probability $p > 0$, the utility loss is bounded probabilistically as:*

$$\Pr\left(Error \leq L_\infty \cdot \sigma\sqrt{2\ln\frac{2d_X}{p}} + 2\eta\|E_{exp}\|_{op}\right) \geq 1 - p$$

*where $L_\infty = 2(1-\eta)\|E_{exp}\|_{op}\|W\|_\infty(\sum_j \|H_j\|_{op})$.*

*Proof.* We use the triangle inequality for the absolute error for a given $x$ and classical noise $\kappa$:

$$|\mathcal{M}_{\text{full}}(x) - \mathcal{M}_{\text{clean}}(x)|$$
$$= |\mathcal{M}_{\text{half}}(x + \kappa) - \mathcal{M}_{\text{clean}}(x)|$$
$$\leq |\mathcal{M}_{\text{half}}(x + \kappa) - \mathcal{M}_{\text{half}}(x)| + |\mathcal{M}_{\text{half}}(x) - \mathcal{M}_{\text{clean}}(x)|$$

Applying our two lemmas, the first term is bounded by $L_\infty \cdot \|\kappa\|_\infty$ and the second term is bounded by $2\eta\|E_{\text{exp}}\|_{op}$.

$$\text{Error} \leq L_\infty \cdot \|\kappa\|_\infty + 2\eta\|E_{\text{exp}}\|_{op}$$

The stochastic error depends on the magnitude of $\|\kappa\|_\infty = \max_i |\kappa_i|$, where each component $\kappa_i$ of the noise vector is an independent draw from a Gaussian distribution, $\kappa_i \sim \mathcal{N}(0, \sigma^2)$.

To obtain a high-probability bound on the maximum of $d$ independent Gaussian variables, we can apply a standard union bound on their tails. For any desired failure probability $p > 0$, with probability at least $1 - p$, the infinity norm of $\kappa$ is bounded by:

$$\|\kappa\|_\infty \leq \sigma\sqrt{2\ln(2d_X/p)}$$

By combining these bounds, we can state that for any $p > 0$, the total utility loss is bounded with probability at least $1 - p$:

$$\text{Error} \leq L_\infty \cdot \sigma\sqrt{2\ln\frac{2d_X}{p}} + 2\eta\|E_{\exp}\|_{op}$$

$\square$

## C   IMPLEMENTATION

We implement all experiments with Python 3.8. Each experiment is conducted on a single GPU-assisted compute node installed with a Linux 64-bit operating system. Our testbed resources include 72 CPU cores with 377 GB of RAM in total. Our allocated node is also provisioned with 2 GPUs with 40GB of VRAM per GPU.

**Implementation of HYPER-Q.** HYPER-Q was implemented using the PennyLane QML simulator (Bergholm et al., 2022). The detailed architecture implements the general mechanism proposed and analyzed in Section 4. Specifically, each input image first passes through two convolutional layers, each followed by batch normalization and max pooling to reduce spatial dimensions and extract salient features. The resulting feature maps are flattened and passed through two fully connected layers to produce a low-dimensional feature vector. This vector is then encoded into a 5-qubit quantum circuit comprising three alternating layers of single-qubit rotations (implemented via RX gates) and entangling layers. This corresponds to the encoding function $f_{\text{enc}}$, where Hermitian generators are given by RX gates. The entangling layers employ a circular arrangement of CNOT gates, such that each qubit $i$ is entangled with qubit $i + 1$, with the last qubit entangled with the first. A projective measurement is applied in the computational basis to extract the quantum outputs, which are then processed by a final fully connected layer to produce the prediction.

| Dataset | Image Dims. | Training | Testing | No. of Labels | Description |
|---|---|---|---|---|---|
| MNIST | 28×28 | 60,000 | 10,000 | 10 | Handwritten digits |
| USPS | 16×16 | ≈ 7,300 | ≈ 2,000 | 10 | Scanned U.S. postal envelopes |
| FashionMNIST | 28×28 | 60,000 | 10,000 | 10 | Clothing items |
| CIFAR-10 | 32×32× 3 | 50,000 | 10,000 | 10 | Natural objects items |

Table 1: Dataset descriptions.

Additionally, the classical and quantum noise levels are set as follows. Given a target differential privacy budget $(\varepsilon', \delta')$, we first fix the quantum depolarizing noise factor $\eta$, and then calibrate the classical Gaussian noise variance $\sigma^2$ to satisfy the budget based on Theorem 1. Specifically, $\sigma^2$ is chosen so that the classical mechanism $A$ achieves $(\varepsilon, \delta)$-DP with

$$\varepsilon = \varepsilon', \qquad \delta = \frac{\delta' - \frac{\eta(1-e^\varepsilon)}{d}}{1 - \eta}.$$

The variance $\sigma^2$ is then computed using the Analytic Gaussian mechanism (Balle & Wang, 2018), ensuring that the classical mechanism $A$ satisfies $(\varepsilon, \delta)$-DP and the composed mechanism $Q^{(\eta)} \circ A$ satisfies the target $(\varepsilon', \delta')$-DP.

# D    DESCRIPTION OF DATASETS & BENCHMARKS

**Datasets:** We evaluate our approach on three image classification datasets: MNIST (Lecun et al., 1998), FashionMNIST (Xiao et al., 2017), and USPS (Hull, 2002). Table 1 briefly describes each of them.

**Benchmarks:** We compare our approach on QML with three classical ML models: Multi-Layer Perceptron (MLP), ResNet-9, and Vision Transformer (ViT). We describe the implementations of those benchmarks below:

- **MLP:** We implement an MLP with a feedforward network composed of fully connected layers and ReLU activations. It consists of one hidden layer with 100 units and a final linear output layer corresponding to the number of classes. It is identical to the default MLP from the `Sci-Kit Learn` library [1]. We implemented it without the library as it is not tailored for GPU usage out of the box. Our *from scratch* version is parallelizable on GPUs.

- **ResNet-9:** We implement a ResNet-9 model inspired by the original in (He et al., 2016). It is comprised of a series of convolutional layers and two residual blocks that include skip connections. It processes inputs through increasing feature dimensions: [32, 64, 128]. We employ batch normalization and ReLU activations throughout the model following by MaxPooling layers. The model ends with a fully-connected layer for classification.

- **ViT:** We implement a ViT model inspired by (Dosovitskiy et al., 2021). It splits input images into non-overlapping patches and linearly embeds them before adding positional encodings and a class token. Multiple self-attention layers processes each sequence before classifying via a fully connected head applied to the class token.

# E    ADVERSARIAL TRAINING AND TESTING

We evaluate the adversarial robustness of `HYPER-Q` via an adversarial training and testing framework inspired by the PixelDP mechanism (Lecuyer et al., 2019). Similar to PixelDP, during training, we define a *construction attack bound* $L_{cons}$ to represent the theoretical robustness guarantee in terms of $\ell_2$ norm. Specifically, this bound establishes the maximum allowable adversarial perturbation under which the model is certified to preserve its prediction capabilities. In our experiments, we vary this value where $L_{cons} = \{0.1, 0.2, 0.3, 0.4\}$. In both `HYPER-Q` and classical baseline models, $\ell_2$-based noise is injected directly into the input. This setup permits a fair comparison of robustness guarantees between quantum and classical models despite their underlying architectural differences.

To evaluate empirical robustness beyond certified guarantees, we assess each model against adversarial perturbations constrained by the $\ell_\infty$ norm. Specifically, for every $L_{cons}$ value, we experiment against empirical attack bounds $L_{attk}$. In our experiments, we vary this value where $L_{attk} = \{0, 0.01, 0.02, 0.03, 0.04, 0.05\}$ while implementing two adversarial attacks: Fast Gradient Sign Method (FGSM) and Projected Gradient Descent (PGD). With this, we are able to observe model performance under realistic threats that may not satisfy the constraints of our certified threat model. In addition, we adopt the randomized smoothing technique proposed by (Cohen et al., 2019) to provide certified predictions against adversarial examples.

# F    ADDITIONAL EXPERIMENTS

## F.1    ROBUSTNESS ANALYSIS IN QML

As in Section 5.1, we evaluate the adversarial robustness of `HYPER-Q` under two quantum noise levels, $\eta \in \{0.1, 0.3\}$. We compare its performance with Basic Gaussian, Analytic Gaussian and DP-SGD mechanisms, ensuring that all methods are evaluated under the same privacy budget and applied to the same QML model. Figures 4, 5 and 6 present the results of the FGSM attack on the FashionMNIST and USPS datasets, respectively with $\varepsilon' \in \{0.25, 0.5, 0.75, 1\}$. In all cases, with the exception of $\varepsilon' = 1$ on USPS, `HYPER-Q` clearly outperforms all baseline methods. On the USPS dataset when $\varepsilon' = 1$, the Analytic Gaussian mechanism outperforms `HYPER-Q` at lower values of

---

[1] https://scikit-learn.org/stable/

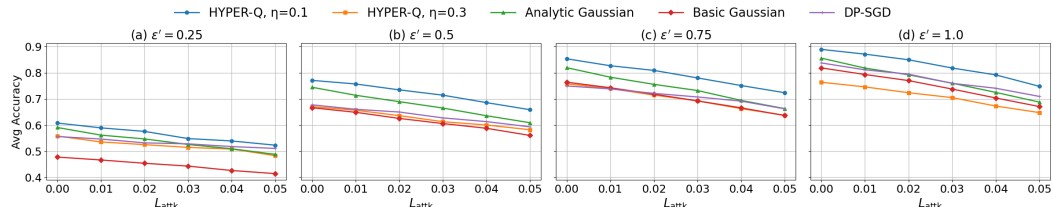

Figure 4: Accuracy of various noise-added mechanisms under the FGSM attack on the MNIST dataset with different $\varepsilon'$ values and $\delta' = 1 \times 10^{-5}$. For each pair of $(L_{\text{attk}}, \varepsilon')$, the reported accuracy is averaged over all $L_{\text{cons}}$ settings. HYPER-Q is examined with $\eta \in [0.1, 0.3]$.

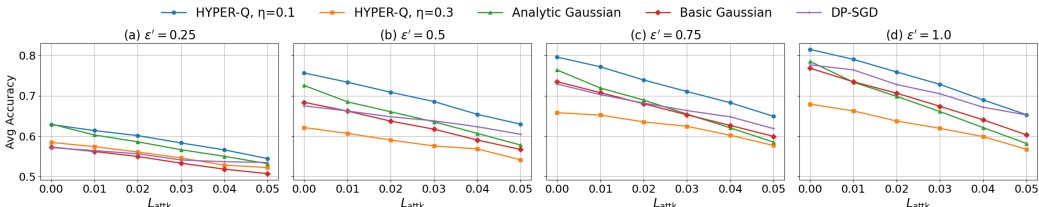

Figure 5: Accuracy of various noise-added mechanisms under the FGSM attack on the FashionMNIST dataset with different $\varepsilon'$ values and $\delta' = 1 \times 10^{-5}$. For each pair of $(L_{\text{attk}}, \varepsilon')$, the reported accuracy is averaged over all $L_{\text{cons}}$ settings. HYPER-Q is examined with $\eta \in [0.1, 0.3]$.

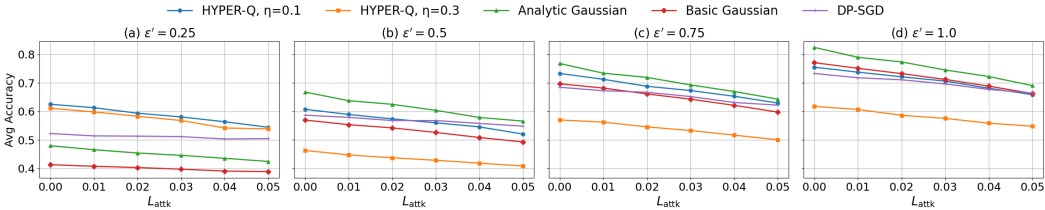

Figure 6: Accuracy of various noise-added mechanisms under the FGSM attack on the USPS dataset with different $\varepsilon'$ values and $\delta' = 1 \times 10^{-5}$. For each pair of $(L_{\text{attk}}, \varepsilon')$, the reported accuracy is averaged over all $L_{\text{cons}}$ settings. HYPER-Q is examined with $\eta \in [0.1, 0.3]$.

$L_{attk}$ ($L_{attk} \in \{0, 0.01\}$), eventually degrading to comparable performance ($L_{attk} \in \{0.02, 0.03\}$) before beginning to underperform at higher values of $L_{attk}$ ($L_{attk} \in \{0.04, 0.05\}$). Similar to the results in Section 5.1, we observe that HYPER-Q with $\eta = 0.3$ degrades very quickly like the Analytic Gaussian and Basic Gaussian mechanisms, even dropping below the two in most cases as the value of $\varepsilon'$ increases.

Figures 7, 8, and 9 present the results of the PGD attack on HYPER-Q and our baseline methods for MNIST, FashionMNIST, and USPS datasets, respectively. Even against the PGD attack, results are similar to the FGSM attack where HYPER-Q clearly outperforms all baselines on each dataset with the exception of $\varepsilon' = 1$ on USPS where the Analytic Gaussian mechanism varies performance as it outperforms HYPER-Q with smaller values of $L_{attk}$ ($L_{attk} = 0$) before becoming comparable ($L_{attk} \in \{0.01, 0.02\}$) and eventually underperforming at higher values of $L_{attk}$ ($L_{attk} \in \{0.03, 0.04, 0.05\}$).

## F.2 COMPARATIVE BENCHMARK WITH CLASSICAL MODELS

As in Section 5.2, we illustrate the performance comparison of a QML model protected by HYPER-Q (with its empirically optimal quantum noise setting, $\eta = 0.1$) against three classical baselines: ResNet-9, ViT, and MLP, each protected by Analytic Gaussian noise. Figures 10, 11 and 12 illustrate the performance comparison between all models on the FashionMNIST and USPS datasets, respectively, while under the FGSM attack. In Figure 11, we observe that the ResNet-9 model, across all values of

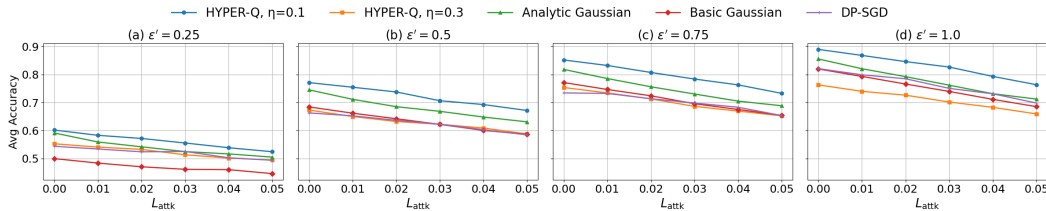

Figure 7: Accuracy of various noise-added mechanisms under the PGD attack on the MNIST dataset with different $\varepsilon'$ values and $\delta' = 1 \times 10^{-5}$. For each pair of $(L_{\text{attk}}, \varepsilon')$, the reported accuracy is averaged over all $L_{\text{cons}}$ settings. HYPER-Q is examined with $\eta \in [0.1, 0.3]$.

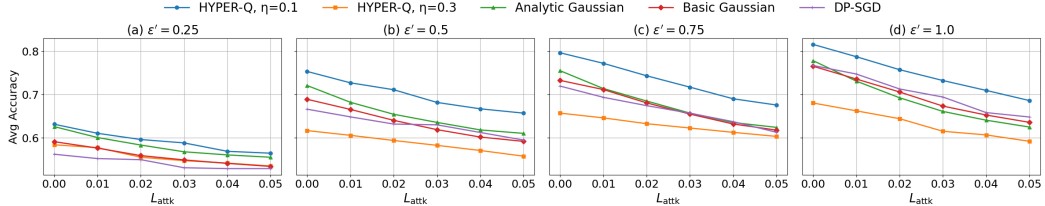

Figure 8: Accuracy of various noise-added mechanisms under the PGD attack on the FashionMNIST dataset with different $\varepsilon'$ values and $\delta' = 1 \times 10^{-5}$. For each pair of $(L_{\text{attk}}, \varepsilon')$, the reported accuracy is averaged over all $L_{\text{cons}}$ settings. HYPER-Q is examined with $\eta \in [0.1, 0.3]$.

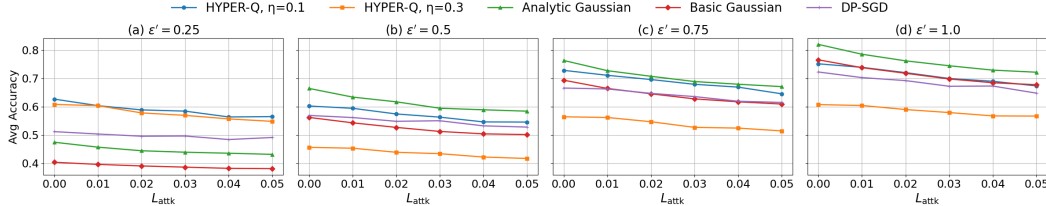

Figure 9: Accuracy of various noise-added mechanisms under the PGD attack on the USPS dataset with different $\varepsilon'$ values and $\delta' = 1 \times 10^{-5}$. For each pair of $(L_{\text{attk}}, \varepsilon')$, the reported accuracy is averaged over all $L_{\text{cons}}$ settings. HYPER-Q is examined with $\eta \in [0.1, 0.3]$.

$\varepsilon'$, outperforms HYPER-Q and the other baseline models. However, it is noted that HYPER-Q is very comparable to the ResNet-9 model with larger values of $\varepsilon'$. Only at higher values of $L_{attk}$ do we observe noticeable separation between the two models. Contrarily, for the USPS dataset, HYPER-Q dominates all other baseline models when $\varepsilon' \in \{0.25, 0.5\}$. Specifically, compared to the ResNet-9 model, HYPER-Q maintains an $\approx 30\%$ higher average accuracy when $\varepsilon' = 0.25$. This value drops to $\approx 2\%$ when $\varepsilon' = 0.5$. The ResNet-9 model becomes more competitive as $\varepsilon' \in \{0.75, 1.0\}$, where it is comparable to HYPER-Q and then outperforms it by $\approx 5\%$, respectively. An interesting observation is the subtle fluctuations of the MLP and quick degradation across all values of $\varepsilon'$. HYPER-Q and the other baselines are much more stable across all values. The results shown in Figures 13, 14, and 15 illustrate the comparative performance of HYPER-Q and our baseline models when subjected to the PGD attack and are virtually identical in nature to the results of the FGSM attack on all three datasets.

### F.3 EMPIRICAL ANALYSIS OF DIMENSIONAL SCALABILITY

To address the practical scalability of HYPER-Q, we empirically investigated the impact of the Hilbert space dimension $d = 2^n$ on the reduction of the required classical noise. While Theorem 1 introduces an additive term $\frac{\eta(1-e^\epsilon)}{d}$ that ostensibly shrinks as the system scales, our analysis reveals that the privacy amplification stabilizes rather than vanishes.

Figure 16 presents the average percentage reduction in classical noise variance ($\sigma^2$) as a function of the number of qubits $n$, ranging from 1 to 29, across various quantum noise levels $\eta \in [0.05, 0.4]$.

The results highlight two observations. For small-scale systems, we observe a massive reduction in the required classical noise, exceeding $90\%$ for $n < 3$. In this regime, the dimension-dependent term $\frac{1}{d}$ in Theorem 1 is dominant, providing a significant bonus to the privacy budget. On the other hand, as the number of qubits increases and the $1/d$ term vanishes, the noise reduction does not drop to zero. Instead, the curves flatten into a stable, non-zero level. This represents the scale-independent multiplicative amplification $(1 - \eta)$ derived in our theoretical framework. For instance, with $\eta = 0.4$, the mechanism maintains a consistent noise reduction of approximately $8\%$ even at $n = 29$ (where $d \approx 5 \times 10^8$).

### F.4 EMPIRICAL VERIFICATION OF UTILITY BOUND TIGHTNESS

To rigorously assess the tightness of the theoretical utility bound derived in Theorem 3, we conducted an empirical analysis comparing the observed worst-case error against the bound. Theorem 3 characterizes the stability of the mechanism by bounding the maximum deviation Error $= \sup_x |\mathcal{M}_{\text{full}}(x) - \mathcal{M}_{\text{clean}}(x)|$. The bound states that

$$\Pr\left(\text{Error} \leq L_\infty \cdot \sigma \sqrt{2 \ln \frac{2d_X}{p}} + 2\eta \|E_{\text{exp}}\|_{op}\right) \geq 1 - p$$

where $L_\infty = 2(1 - \eta)\|E_{\text{exp}}\|_{op}\|W\|_\infty (\sum_j \|H_j\|_{op})$.

We evaluated the tightness of our bound by measuring the ratio between the maximum empirical error observed in simulation and the theoretical bound $\text{Bound}(\sigma, \eta, p) = L_\infty \sigma \sqrt{2 \ln(2d_X/p)} + 2\eta\|E_{\text{exp}}\|_{op}$. In particular, given a sample set $S$, the ratio is calculated by:

$$\text{Ratio} = \frac{\max_{x \in S} |\mathcal{M}_{\text{full}}(x) - \mathcal{M}_{\text{clean}}(x)|}{\text{Bound}(\sigma, \eta, p)}$$

In this experiment, we set the failure probability to $p = 0.01$. That ensures the theoretical bound holds with a $99\%$ confidence level. We computed the Ratio for each $(\sigma, \eta)$ configuration using a sample size of $|S| = 10000$, where a value approaching 1 indicates a tight bound. Figure 17 presents the resulting ratios across the parameter grid. We observe that in low-noise settings, such as $(\sigma, \eta) = (0.5, 0)$, the ratio reaches significant magnitudes (e.g., $0.923$), confirming that the bound effectively captures the worst-case error. While the bound becomes looser as the total noise magnitude increases, it remains non-trivial. Notably, in the absence of classical noise ($\sigma = 0$), the ratios remain constant across all $\eta$. This behavior is attributed to the linearity of the depolarizing channel, where both the empirical error and the theoretical quantum term ($2\eta\|E_{\text{exp}}\|_{op}$) scale linearly with $\eta$.

### F.5 SENSITIVITY ANALYSIS OF THE DEPOLARIZING NOISE PARAMETER $\eta$

To characterize the impact of the quantum noise parameter on model utility, we evaluate model accuracy across varying levels of depolarizing noise $\eta \in \{0.05, 0.1, \dots, 0.4\}$. For each noise level $\eta$, we measured robustness against FGSM attacks with varying strengths $L_{\text{attk}} \in \{0, 0.01, \dots, 0.05\}$. We conducted these experiments across three datasets (MNIST, Fashion-MNIST, and USPS) under four distinct differential privacy guarantees $\varepsilon' \in \{0.25, 0.5, 0.75, 1\}$.

Figures 18,19 and 20 collectively show that the relationship between depolarizing noise $\eta$ and and model performance exhibits a remarkably consistent structure across all three datasets: MNIST, Fashion-MNIST and USPS. Despite differences in dataset complexity, the accuracy curves share the same unimodal shape. Specifically, performance initially increases as $\eta$ moves away from 0, reach a peak and, then declines once the quantum distortion dominates. This pattern is visible in every privacy budget $\varepsilon' \in \{0.25, 0.5, 0.75, 1\}$ and across all attack bounds $L_{\text{attk}}$.

When comparing the location of the performance peaks across datasets, we observe a highly aligned trend. Under stricter privacy budgets ($\varepsilon' \leq 0.5$), the best performance is usually achieved at $\eta = 0.1$ or $\eta = 0.5$. For example, MNIST and Fashion-MNIST peak at $\eta = 0.1$ and USPS similarly peaks at $\eta = 0.1 - 0.15$ for $\varepsilon' = 0.25$. As the privacy requirement becomes more relaxed ($\varepsilon' \geq 0.75$), all three datasets shift their peaks toward smaller noise levels, typically $\eta = 0.05$. We observe that the optimal $\eta$ consistently falls within the narrow interval 0.05-0.15. Although $\eta$ is often difficult to

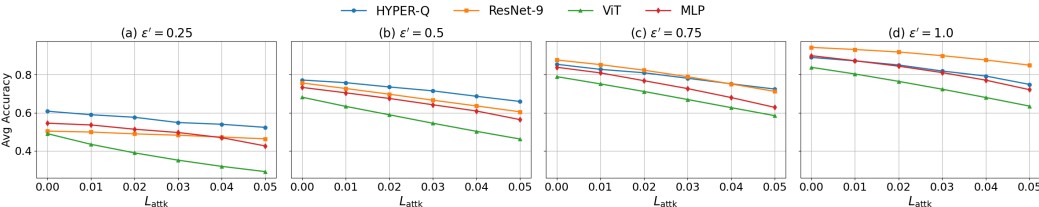

Figure 10: Accuracy comparison between the QML model protected by HYPER-Q and three classical baselines (ResNet-9, ViT, and MLP) protected by Analytic Gaussian noise under the FGSM attack on the MNIST dataset. The HYPER-Q model is evaluated with its empirically best quantum noise setting ($\eta = 0.1$). For each ($L_{\text{attk}}, \varepsilon'$) pair, the reported accuracy is averaged over all $L_{\text{cons}}$ settings. $\delta' = 1 \times 10^{-5}$ for all settings.

calibrate precisely in practice Hu et al. (2023), this stable peak range provides a robust guideline that users can reliably calibrate $\eta$ within this interval without requiring an exhaustive sweep.

### F.6 PERFORMANCE ANALYSIS ON CIFAR-10

To evaluate the robustness of HYPER-Q on more complex data, we extend our experiments to CIFAR-10, a significantly more challenging benchmark than MNIST and USPS. Unlike grayscale datasets, CIFAR-10 consists of RGB images with higher variability and richer feature structure, requiring a larger quantum feature map. For this setting, we employ a 10-qubit variational QML model and compare HYPER-Q with three classical baselines under a fixed privacy budget $\varepsilon' = 1$. Figure 21 shows the test accuracy as a function of attack strength $L_{\text{attk}} \in \{0, 0.01, \ldots, 0.05\}$. Across all models, accuracy decreases as the attack strength increases, but the rate of degradation varies significantly. HYPER-Q begins at 73.9% accuracy at $L_{\text{attk}} = 0$ and declines smoothly to 47.7% at $L_{\text{attk}} = 0.05$. This degradation profile is comparable to ResNet-9, which starts at a higher baseline of 86.0% but similarly drops to 48.7% at the highest attack bound. In contrast, ViT and MLP degrade much more rapidly, falling from 75.9% and 69.3% initially to only 14.2% and 10.8% at $L_{\text{attk}} = 0.05$, respectively.

These results highlight two insights. First, even for a high-dimensional image dataset requiring a deeper quantum representation, HYPER-Q remains competitive with classical baselines under moderate attack strengths. Second, while classical deep models exhibit higher clean accuracy, their robustness diminishes sharply under increasing perturbation, whereas HYPER-Q shows a more controlled and stable decline. This demonstrates that the hybrid-noise mechanism HYPER-Q continues to offer meaningful utility benefits in more complex, higher-qubit QML settings.

### F.7 GENERAL OBSERVATIONS

We note that HYPER-Q exhibits resilience to small $L_\infty$ perturbations attributing to the nonlinear separability and the enhanced representational capacity of quantum feature embeddings. However, we note that as the attack strength increases, sensitivity varies. Contrarily, the classical baselines show a much more pronounced and predictable degradation in robustness when increasing $L_\infty$ perturbations. However, even though the identical $\ell_2$ certification bounds are applied to each model, architectural differences lead to variations where quantum models may underutilize or overconservatively interpret certification bounds due to the non-Euclidean geometry of Hilbert spaces. This further highlights the distinct robustness characteristics of quantum-ehanced learning in adversarial settings.

## G USE OF LARGE LANGUAGE MODELS

Portions of this manuscript were refined using a large language model (LLM) to improve clarity, grammar, and readability. The use of the LLM was limited strictly to language polishing, and no content, analysis, or results were generated by the model.

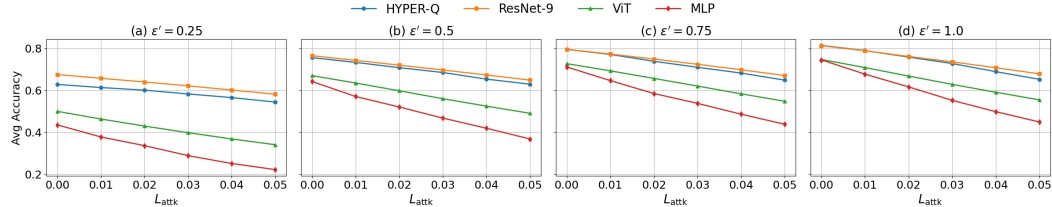

Figure 11: Accuracy comparison between the QML model protected by HYPER-Q and three classical baselines (ResNet-9, ViT, and MLP) protected by Analytic Gaussian noise under the FGSM attack on the FashionMNIST dataset. The HYPER-Q model is evaluated with its empirically best quantum noise setting ($\eta = 0.1$). For each $(L_{\text{attk}}, \varepsilon')$ pair, the reported accuracy is averaged over all $L_{\text{cons}}$ settings. $\delta' = 1 \times 10^{-5}$ for all settings.

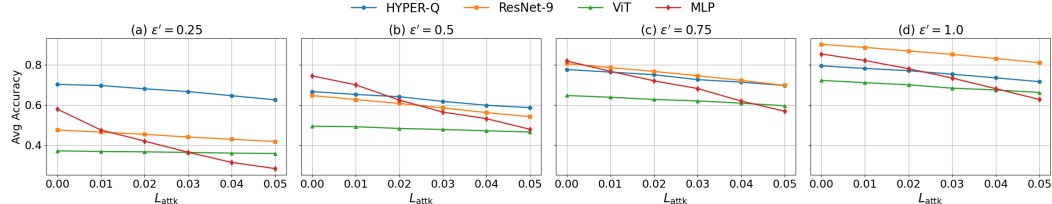

Figure 12: Accuracy comparison between the QML model protected by HYPER-Q and three classical baselines (ResNet-9, ViT, and MLP) protected by Analytic Gaussian noise under the FGSM attack on the USPS dataset. The HYPER-Q model is evaluated with its empirically best quantum noise setting ($\eta = 0.1$). For each $(L_{\text{attk}}, \varepsilon')$ pair, the reported accuracy is averaged over all $L_{\text{cons}}$ settings. $\delta' = 1 \times 10^{-5}$ for all settings.

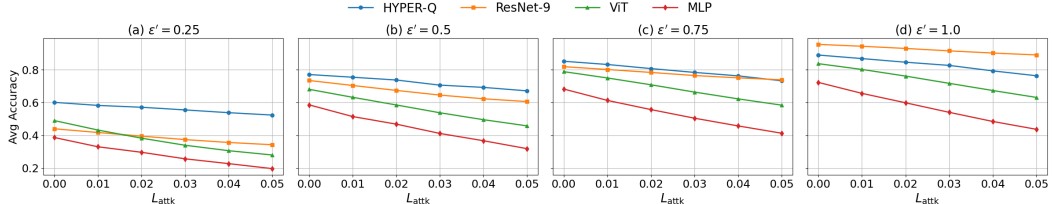

Figure 13: Accuracy comparison between the QML model protected by HYPER-Q and three classical baselines (ResNet-9, ViT, and MLP) protected by Analytic Gaussian noise under the PGD attack on the MNIST dataset. The HYPER-Q model is evaluated with its empirically best quantum noise setting ($\eta = 0.1$). For each $(L_{\text{attk}}, \varepsilon')$ pair, the reported accuracy is averaged over all $L_{\text{cons}}$ settings. $\delta' = 1 \times 10^{-5}$ for all settings.

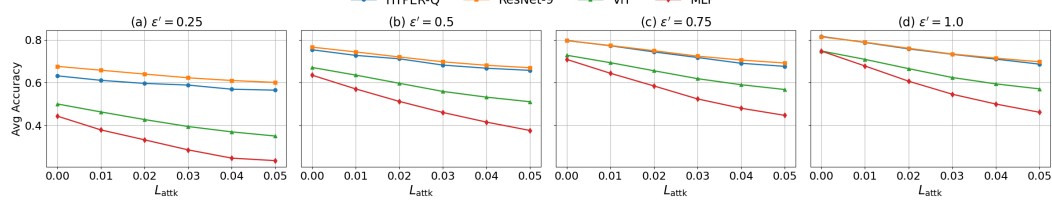

Figure 14: Accuracy comparison between the QML model protected by HYPER-Q and three classical baselines (ResNet-9, ViT, and MLP) protected by Analytic Gaussian noise under the PGD attack on the FashionMNIST dataset. The HYPER-Q model is evaluated with its empirically best quantum noise setting ($\eta = 0.1$). For each $(L_{\text{attk}}, \varepsilon')$ pair, the reported accuracy is averaged over all $L_{\text{cons}}$ settings. $\delta' = 1 \times 10^{-5}$ for all settings.

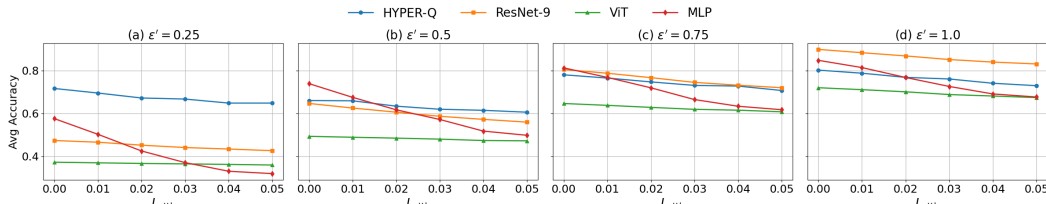

Figure 15: Accuracy comparison between the QML model protected by HYPER-Q and three classical baselines (ResNet-9, ViT, and MLP) protected by Analytic Gaussian noise under the PGD attack on the USPS dataset. The HYPER-Q model is evaluated with its empirically best quantum noise setting ($\eta = 0.1$). For each ($L_{\text{attk}}, \varepsilon'$) pair, the reported accuracy is averaged over all $L_{\text{cons}}$ settings. $\delta' = 1 \times 10^{-5}$ for all settings.

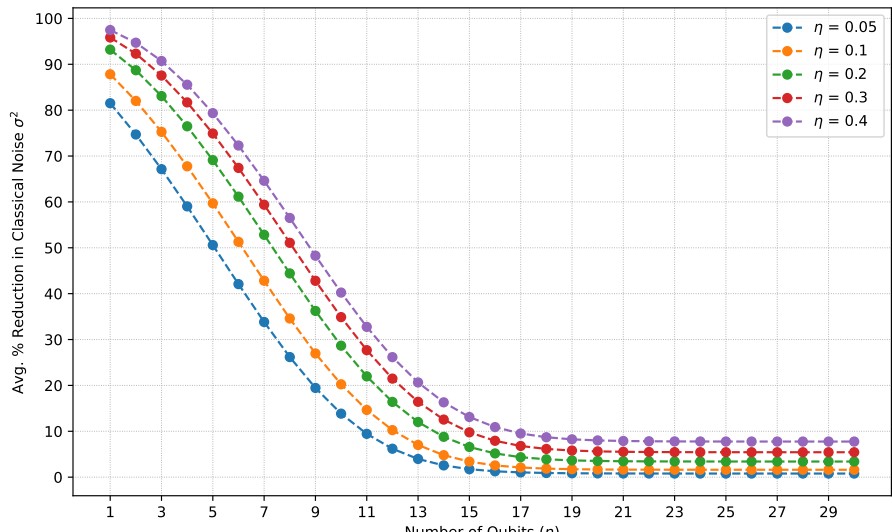

Figure 16: Effect of Quantum Noise Level on Classical-Noise Reduction Across Scaling Qubit Counts

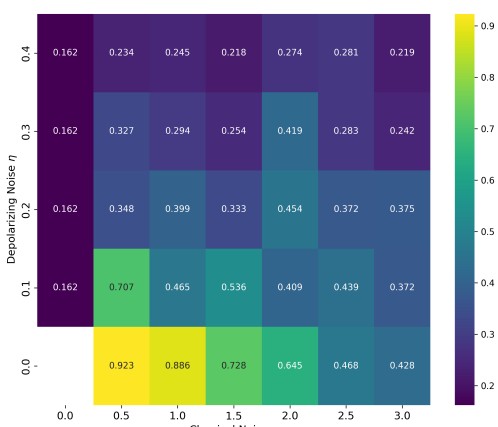

Figure 17: Heatmap of the ratio between the actual utility loss and the theoretical bound across $(\eta, \sigma)$ values.

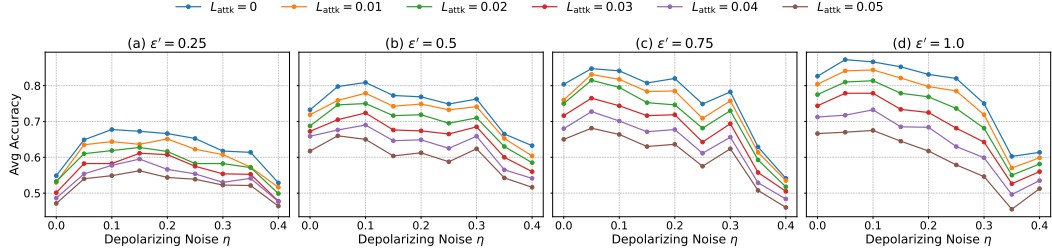

Figure 18: Impact of depolarizing noise $\eta$ on model utility for the MNIST dataset. Subplots (a)-(d) show performance under varying privacy budgets $\varepsilon' \in \{0.25, 0.5, 0.75, 1.0\}$. Each curve represents the average accuracy against FGSM attacks with varying strengths $L_{\text{attk}} \in \{0, \ldots, 0.05\}$.

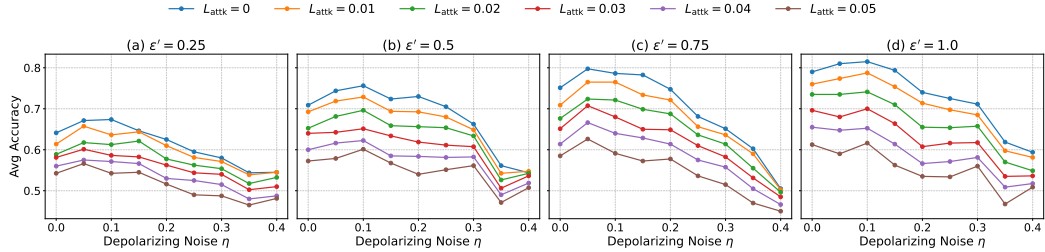

Figure 19: Impact of depolarizing noise $\eta$ on model utility for the Fashion-MNIST dataset. Subplots (a)-(d) show performance under varying privacy budgets $\varepsilon' \in \{0.25, 0.5, 0.75, 1.0\}$. Each curve represents the average accuracy against FGSM attacks with varying strengths $L_{\text{attk}} \in \{0, \ldots, 0.05\}$.

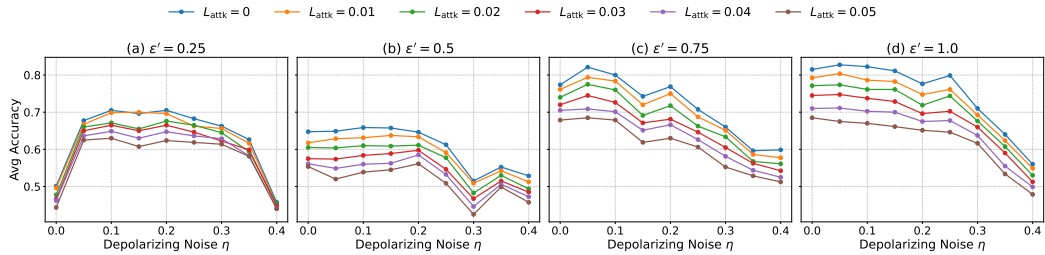

Figure 20: Impact of depolarizing noise $\eta$ on model utility for the USPS dataset. Subplots (a)-(d) show performance under varying privacy budgets $\varepsilon' \in \{0.25, 0.5, 0.75, 1.0\}$. Each curve represents the average accuracy against FGSM attacks with varying strengths $L_{\text{attk}} \in \{0, \dots, 0.05\}$.

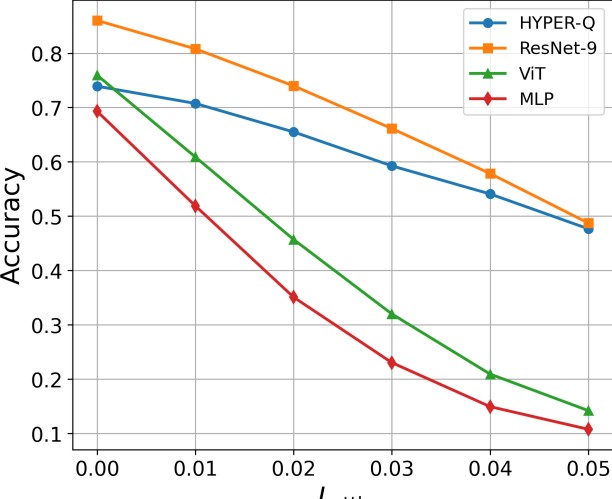

Figure 21: Performance of HYPER-Q and classical baselines (ResNet-9, ViT, and MLP) on CIFAR-10 under varying attack strengths $L_{\text{attk}}$ with a fixed privacy budget $\varepsilon' = 1$.

