# OpenReview forum: "Guaranteeing Privacy in Hybrid Quantum Learning through Theoretical Mechanisms"
_ICLR.cc/2026/Conference — Submitted to ICLR 2026_

### Official Review · Reviewer_BQou · 2025-10-28

**Soundness:** 2
**Presentation:** 3
**Contribution:** 2
**Rating:** 4
**Confidence:** 5

**Summary:**

The manuscript proposes a hybrid privacy mechanism that combines classical differentially private input noise with depolarizing noise (quantum) to amplify differential privacy in quantum machine learning. The core idea is to consider a classical privacy mechanism to provide an initial DP and then use the depolarizing noise of the quantum circuit as a formal privacy amplifier.

**Strengths:**

The key strength is that this framework is the first of its kind: a novel theoretical framework for analyzing privacy (DP) in hybrid quantum-classical models.
Second, it provides rigorous mathematical proofs (Theorems 1 and 2) showing that the quantum depolarizing noise acts as a privacy amplifier.
The authors identify POVM trace balancing as a design rule to further tighten delta. This provided a concrete measurement layer condition for optimal amplification.
The paper is well written (except for the abstract) and is easy to follow. The work is commendable, lighting a torch towards a new direction of quantum noise-based differential privacy.

**Weaknesses:**

The abstract is misleading, although the manuscript follows by introducing DP and QDP eventually, the abstract says “noise means privacy” which is incomplete and misleading to a non-DP expert.
The idea is very good. Leveraging the quantum noise as a DP mechanism is like realizing a foe can be a friend. But the formulation in the manuscript seems simpler than the actual noise scenario that is in quantum world.  It starts with the scope of the paper which considers only the depolarization noise. Before moving into the discussion of the theoretical formulation aspect, this umbrella assumption itself is flawed/incomplete.  The mathematical properties are not universal, although acknowledged, one of the most critical aspects is considered as future work.

For example, let’s just take some unavoidable temporal ones, like, amplitude damping noise (T1) or phase damping (T2). T1 is not a uniform contraction of the Bloch sphere. It asymmetrically pulls the state towards the ground state. It does not have the simple convex combination form. Another dangerous one is coherent noises. They contradict the whole concept of randomness; they are systematic.

It is highly plausible that these noises might create a new vulnerability by systematically shifting outputs in a predictable way.

If the authors aim to show a proof a concept that the quantum noise has the potential to show DP properties, then, as per my judgment, it has already been done rigorously in the literature. The adaptation of quantum noise as an amplifier to classical form is a novelty but not enough.

Experiments: Datasets selected are although suitable for a first proof of concept, they are considered as the toy-datsets by the community; Mnist, FashionMNIST (just an extension with same dimension 28x28, and features as grayscale pixel value), and USPS. More complex, at least something like Cifar-10 is expected.  This might steer in a new direction, say more qubits requirements. Adding qubits to a circuit generally reduces fidelity and forces η upward, quickly negating the dimensional advantage. Will privacy–utility curve changes shape? My intuition says it will.

Clarification required regarding the narrative “HYPER-Q improves robustness in QML models.”
The experiment compares HyperQ (0.1, 0.3) against classical-DP applied to the same QML model. While the HYPER-Q-protected QML model wins under very strict privacy budgets, it seems outperformed by a standard classical ResNet-9 when the privacy budget is relaxed. This suggests that the quantum advantage shown is not inherent to the QML model itself, but rather an artifact of the classical model being "over-penalized" by extreme noise. Is this happening? I would love a clarification.

**Questions:**

Is it quite impractical to assume that the quantum computers in today’s generation are unequipped with QEM techniques?
What kind of roles the famous QEMs will play in this formulation? For eg, Lets take ZNE.   What will be the effect on the variance? Will this deterministic extrapolation destroy the stochastic needed? Take PEC, for instance, it’s much more complicated.

---

> ### Author Response · Authors · 2025-11-22
>
> We thank the reviewer for the constructive feedback and the encouraging remarks. We appreciate the recognition that our work introduces the first theoretical framework for analyzing differential privacy in hybrid quantum–classical models, and that our mathematical development is rigorous. We are also encouraged by the reviewer’s view that this work opens a new direction in quantum noise–based differential privacy.
>
> Below, we address all raised weaknesses (`W`) and questions (`Q`) in detail. We have also submitted our revised manuscript, with all new material highlighted in yellow.

---

> ### Author Response · Authors · 2025-11-22
>
> `[W1]`. The phase “noise means privacy” in the abstract can be incomplete to a non-DP expert.
>
> `[A1]`. We thank the reviewer for this important comment regarding the precision of our abstract. To address this concern, we have revised the abstract to use more precise terminology. In the revised version, we explicitly state that intrinsic quantum noise provides a natural stochastic resource that, when rigorously analyzed within the differential privacy (DP) framework and composed with classical mechanisms, can satisfy formal $(\varepsilon, \delta)$-DP guarantees. This revision removes any potential ambiguity and avoids the unintended implication that "noise means privacy" outside of the DP setting.

---

> ### Author Response · Authors · 2025-11-22
>
> `[W2]`. The formulation in the manuscript seems simpler than the actual noise scenario that is in quantum world. More realistic noise processes, such as amplitude damping (T1) or phase damping (T2), behave very differently and may not satisfy the same mathematical properties.
>
> `[A2]`. We thank the reviewer for this critical insight regarding the complexity of real-world quantum noise. While our main theorems are stated for the depolarizing channel, the central insight of our framework is more general. **In particular, the central insight is whenever a quantum noise channel induces a non-trivial contraction of the quantum hockey-stick divergence, it will inherently lead to privacy amplification**. In the revised manuscript, we have added **two new theoretical results** that demonstrate how our framework applies to realistic hardware noise channels in Section 4.2.3 with the detailed proofs in Appendix B.4. Two new theoretical results can be summarized as follows:
>
> **(1) Generalized Amplitude Damping (GAD) Channel.** We introduce a **new lemma (Lemma 4.1)** showing that the generalized amplitude damping channel, which is one of the most widely used models for energy relaxation in superconducting and trapped-ion hardware, contracts the quantum hockey-stick divergence by an explicit state-independent factor. The contraction coefficient leads directly to improved $(\varepsilon',\delta')$ stated in Theorem 1.1 as follows:
>
> **Theorem 1.1 [Amplification Under Generalized Amplitude Damping Noise].**
> Let $A: \mathbb{X} \rightarrow \mathcal{P}(\mathbb{Y})$ be a classical mechanism satisfying $(\varepsilon, \delta)$-DP where $A = f_{\text{par}} \circ f_{\text{cdp}}$, and let $Q^{(p,\eta)}: \mathbb{Y} \rightarrow \mathcal{P}(\mathbb{Z})$ be a quantum mechanism in $d$-dimensional Hilbert space defined as $Q^{(p,\eta)} = f_{\text{mea}} \circ f_{\mathrm{GAD}}^{(p,\eta)} \circ f_{\text{enc}}$ where $f_{\mathrm{GAD}}^{(p,\eta)}$ is the GAD channel parameterized by a damping strength $\eta \in [0,1]$ and an excitation probability $p \in [0,1]$. Then, the composed mechanism $Q^{(p,\eta)} \circ A$ satisfies $(\varepsilon', \delta')$-DP, where $\varepsilon' = \varepsilon, \quad \delta' = (2\sqrt{\eta} - \eta) \delta.$
>
> **(2) Generalized Dephasing (GD) Channel.** We further provide a theorem establishing amplification under the generalized dephasing channel. Under a practical encoder assumption (e.g., equatorial encodings commonly used in QML), the channel yields a closed-form contraction bound for the quantum hockey-stick divergence (**new Lemma 4.2**). This again produces strictly improved privacy parameters, stated in Theorem 1.2.
>
> **Theorem 1.2 [Amplification Under Generalized Dephasing Noise].**
> Let $A: \mathbb{X}\to\mathcal{P}(\mathbb{Y})$ be a classical mechanism satisfying $(\varepsilon,\delta)$-DP, and let $Q^{(\eta)} := f_{\text{mea}} \circ f_{\mathrm{GD}}^{(\eta)} \circ f_{\text{enc}}$ be an $n$-qubit quantum mechanism where $f_{\mathrm{GD}}^{(\eta)}$ is the GD channel parameterized by a dephasing parameter $\eta \in [0,1]$, and $f_{\mathrm{enc}}$ satisfies Assumption 1 (equatorial–state assumption). Then the composed mechanism $Q^{(\eta)}\circ A$ satisfies $(\varepsilon',\delta')$-DP with
> $
>     \varepsilon' = \varepsilon,  \quad
>     \delta' = |1-2\eta|\cdot \delta.
> $
>
> These two results demonstrate that our privacy-amplification framework is not restricted to the depolarizing channel, but extends to two of the most dominant noise processes in practical quantum hardware.

---

> ### Author Response · Authors · 2025-11-22
>
> `[W3]`. If the authors aim to show a proof a concept that the quantum noise has the potential to show DP properties, then, as per my judgment, it has already been done rigorously in the literature. The adaptation of quantum noise as an amplifier to classical form is a novelty but not enough.
>
> `[A3]`. We would like to clarify that our work does not aim to simply show that quantum noise has the potential to show DP properties. Our work studies the hybrid privacy landscape that integrates both classical noise injection and intrinsic quantum noise. Existing literature has not investigated this problem.
>
> We also would like to discuss about the impact of our work. Hybrid classical–quantum computing is becoming an inevitable trend ([1], [2], [3], [4]) and understanding how privacy behaves in such systems is essential. In addition, although quantum noise has been rigorously analyzed as a source of DP, it is difficult to calibrate in practice ([5]), which limits its direct applicability in future hybrid architectures. Classical noise, on the other hand, is easily tunable and can serve as a flexible component that complements intrinsic quantum noise to achieve a desired privacy budget. Our work establishes the first DP foundation for hybrid classical–quantum systems and can serve as a starting point for future exploration.
>
> ---
> ## References
> [1]. Yao, G., \& Gupta, L. (2025). A Benchmarking Framework for Hybrid Quantum–Classical Edge-Cloud Computing Systems. Applied Sciences, 15(18), 10245.
>
> [2]. Long, C., Huang, M., Ye, X., Futamura, Y., \& Sakurai, T. (2025). Hybrid quantum-classical-quantum convolutional neural networks. Scientific Reports, 15(1), 31780.
>
> [3]. IBM, “IBM and AMD Join Forces to Build the Future of Computing,” IBM Newsroom, Aug. 26 2025. [Online]. Available: https://newsroom.ibm.com/2025-08-26-ibm-and-amd-join-forces-to-build-the-future-of-computing.
>
> [4]. NVIDIA Corporation, “NVIDIA Introduces NVQLink — Connecting quantum and GPU computing for 17 quantum builders and nine scientific labs,” NVIDIA Newsroom, Oct. 28 2025. [Online]. Available: https://nvidianews.nvidia.com/news/nvidia-nvqlink-quantum-gpu-computing.
>
> [5]. Hu, Z., Wolle, R., Tian, M., Guan, Q., Humble, T., \& Jiang, W. (2023). Toward Consistent High-Fidelity Quantum Learning on Unstable Devices via Efficient In-Situ Calibration. Proceedings - 2023 IEEE International Conference on Quantum Computing and Engineering, QCE 2023 (pp. 848-858).

---

> ### Author Response · Authors · 2025-11-22
>
> `[W4]`. Experiments: Although selected datasets (MNIST, Fashion-MNIST, USPS) are suitable for a first proof of concept, they are simple. More complex data like CIFAR-10 is expected. This might steer in a new direction, say more qubits requirements. Adding qubits to a circuit generally reduces fidelity and forces $\eta$ upward, quickly negating the dimensional advantage. Will privacy–utility curve changes shape?
>
> `[A4]`. Following the reviewer's recommendation, we have added **a new experiment** to include the CIFAR-10 dataset using a 10-qubit QML model. The results demonstrate that HYPER-Q maintains competitive utility against classical baselines even on this more complex input space. That suggests the mechanism generalizes beyond simple grayscale images. For **new experimental results**, we refer the reviewer to Figure 21, Appendix F.6 in our revised manuscript or this anonymous link: https://imgur.com/a/V37S1KQ.
>
> We acknowledge the reviewer's insight that on physical hardware, increasing the qubit count to accommodate complex data naturally forces the intrinsic $\eta$ upward. Studying how the privacy–utility curve may change its shape as qubit count grows requires access to real hardware, where both device-specific noise channels and coherence limitations are present. Due to our limited access to quantum simulators, we view this as an important direction for future work which studies the behavior of hybrid DP mechanisms on larger variational circuits deployed on actual quantum hardwares.

---

> ### Author Response · Authors · 2025-11-22
>
> `[W5]`. While the HYPER-Q-protected QML model wins under very strict privacy budgets, it seems outperformed by a standard classical ResNet-9 when the privacy budget is relaxed. This suggests that the quantum advantage shown is not inherent to the QML model itself, but rather an artifact of the classical model being "over-penalized" by extreme noise. Is this happening?
>
> `[A5]`. Before providing such discussion, we would like to clarify that the behavior observed in our experiments is consistent with the broader findings in the quantum machine learning literature. In non-DP settings, hybrid quantum models have not demonstrated an accuracy advantage over strong classical CNNs (such as ResNet-9) on image datasets like MNIST or Fashion-MNIST ([1], [2]). This limitation is well documented and generally attributed to current quantum challenges such as limited qubit counts, circuit depth constraints, and scalability issues ([3]).
>
> Under DP settings, classical CNNs are highly sensitive to noise, especially with small $\varepsilon'$. Naturally, when the privacy budget becomes more permissive ($\varepsilon' = 1$), the amount of classical noise is much smaller and ResNet-9 regains its usual advantage. That explains for the reviewer's observation on ResNet-9's performance.
>
> We also would like to clarify that our target is not desiging a new hyrbid quantum that outperforms classical CNNs in image classification tasks. Instead, our objective is to develop a principled framework that leverages quantum noise inside a hybrid model to reduce the amount of classical Gaussian noise required for DP. The comparison with classical CNNs demonstrates that in strict privacy regimes (small $\varepsilon'$), classical CNNs become highly unstable due to the large noise they must inject, whereas HYPER-Q remains substantially more stable.
>
> ---
> ## References
> [1]. Bowles, Joseph et al. “Better than classical? The subtle art of benchmarking quantum machine learning models.” ArXiv abs/2403.07059 (2024).
>
> [2]. Long, C., Huang, M., Ye, X., Futamura, Y., \& Sakurai, T. (2025). Hybrid quantum-classical-quantum convolutional neural networks. Scientific Reports, 15(1), 31780.
>
> [3]. McClean, Jarrod R. et al. Barren plateaus in quantum neural network training landscapes. Nature Communications 9 (2018).

---

> ### Author Response · Authors · 2025-11-22
>
> `[Q1]`. Is it quite impractical to assume that the quantum computers in today’s generation are unequipped with QEM techniques? What kind of roles the famous QEMs (i.e., ZNE) will play in this formulation? What will be the effect on the variance? Will this deterministic extrapolation destroy the stochastic needed?
>
> `[A6]`. This is a very interesting question regarding how Quantum Error Mitigation (QEM) interacts with differential privacy. First of all, we would like to clarify that the scope of our work does not focus on QEM techniques or their formal integration into the DP analysis. Nevertheless, we are happy to discuss the potential impact of Zero-Noise Extrapolation (ZNE), one of the most widely used QEM methods, in the context of HYPER-Q.
>
> In HYPER-Q, the privacy guarantee comes from two components: (i) classical Gaussian noise, which is unbiased and increases variance, and (ii) quantum depolarizing noise, which increases both variance and bias and contributes to privacy amplification through channel contraction. On the other hand, ZNE reduces bias in quantum circuits by deliberately evaluating the circuit at several amplified noise levels and extrapolating the expectation value back to the zero-noise limit. If ZNE is applied only to the quantum component of HYPER-Q, then it effectively reduces the bias contribution of the quantum depolarizing channel. At the same time, ZNE increases variance because the method relies on repeated noisy measurements. From our perspective, ZNE trades reduced bias for increased statistical variance. Thus, it does not destroy the stochasticity needed for DP.

---

### Official Review · Reviewer_2RVG · 2025-11-01

**Soundness:** 3
**Presentation:** 3
**Contribution:** 4
**Rating:** 6
**Confidence:** 2

**Summary:**

This paper proposes HYPER-Q, a new theoretical framework for ensuring differential privacy of hybrid quantum-classical machine learning (QML) models. The authors present the availability of quantum noise, previously considered only an error, as a useful resource to amplify privacy, and demonstrate the robustness of the proposed technique through experiments.

**Strengths:**

It addresses a very important and timely problem of ensuring privacy in practical hybrid quantum machine learning (QML) models.

It is considered a novel attempt to address how the classical DP mechanism and quantum noise can be theoretically combined to be exploited.

By mathematically proving the proposed technology using Theorem, theoretical completion is high, and it has been demonstrated through experiments.

**Weaknesses:**

Although the paper assumes that quantum noise is the perfect depolarizing channel, the noise in real NISQ devices takes a much more complex and asymmetric form. There is no analysis of whether the privacy "amplification" effect remains the same in non-defolarizing noise environments.

**Questions:**

In Figure 3, in the $\epsilon'=1.0$ interval, the classical model ResNet-9 performs better than HYPER-Q (QML). The benefits of HYPER-Q appear only under very strict privacy constraints ($\epsilon' \le 0.5$), which needs to be clarified why.

Theorem 1 (privacy amplification), the core of this paper, is proved to be the perfect depolarization channel for noise. Is privacy amplification always guaranteed even for general and asymmetric quantum channels?

---

> ### Author Response · Authors · 2025-11-22
>
> We thank the reviewer for the thoughtful and encouraging feedback. We appreciate the recognition that our work addresses an important and timely challenge. We are grateful for the reviewer’s acknowledgment of the novelty in combining classical DP mechanisms with intrinsic quantum noise and for highlighting the strength of our theoretical contributions.
>
> Below, we address all raised weaknesses (`W`) and questions (`Q`) in detail. We have also submitted our revised manuscript, with all new material highlighted in yellow.

---

> ### Author Response · Authors · 2025-11-22
>
> `[W1 + Q2]`. Theorem 1 (privacy amplification) is proved to be the perfect depolarization channel for noise. Is privacy amplification always guaranteed even for general and asymmetric quantum channels?
>
> `[A1]`. We thank the reviewer for highlighting this important point. While our main theorems are stated for the depolarizing channel, the central insight of our framework is more general. **In particular, the central insight is whenever a quantum noise channel induces a non-trivial contraction of the quantum hockey-stick divergence, it will inherently lead to privacy amplification**. In the revised manuscript, we have added **two new theoretical results** that demonstrate how our framework applies to realistic hardware noise channels in Section 4.2.3 with the detailed proofs in Appendix B.4. Two new theoretical results can be summarized as follows:
>
> **(1) Generalized Amplitude Damping (GAD) Channel.** We introduce a **new lemma (Lemma 4.1)** showing that the generalized amplitude damping channel, which is one of the most widely used models for energy relaxation in superconducting and trapped-ion hardware, contracts the quantum hockey-stick divergence by an explicit state-independent factor. The contraction coefficient leads directly to improved $(\varepsilon',\delta')$ stated in Theorem 1.1 as follows:
>
> **Theorem 1.1 [Amplification Under Generalized Amplitude Damping Noise].**
> Let $A: \mathbb{X} \rightarrow \mathcal{P}(\mathbb{Y})$ be a classical mechanism satisfying $(\varepsilon, \delta)$-DP where $A = f_{\text{par}} \circ f_{\text{cdp}}$, and let $Q^{(p,\eta)}: \mathbb{Y} \rightarrow \mathcal{P}(\mathbb{Z})$ be a quantum mechanism in $d$-dimensional Hilbert space defined as $Q^{(p,\eta)} = f_{\text{mea}} \circ f_{\mathrm{GAD}}^{(p,\eta)} \circ f_{\text{enc}}$ where $f_{\mathrm{GAD}}^{(p,\eta)}$ is the GAD channel parameterized by a damping strength $\eta \in [0,1]$ and an excitation probability $p \in [0,1]$. Then, the composed mechanism $Q^{(p,\eta)} \circ A$ satisfies $(\varepsilon', \delta')$-DP, where $\varepsilon' = \varepsilon, \quad \delta' = (2\sqrt{\eta} - \eta) \delta.$
>
> **(2) Generalized Dephasing (GD) Channel.** We further provide a theorem establishing amplification under the generalized dephasing channel. Under a practical encoder assumption (e.g., equatorial encodings commonly used in QML), the channel yields a closed-form contraction bound for the quantum hockey-stick divergence (**new Lemma 4.2**). This again produces strictly improved privacy parameters, stated in Theorem 1.2.
>
> **Theorem 1.2 [Amplification Under Generalized Dephasing Noise].**
> Let $A: \mathbb{X}\to\mathcal{P}(\mathbb{Y})$ be a classical mechanism satisfying $(\varepsilon,\delta)$-DP, and let $Q^{(\eta)} := f_{\text{mea}} \circ f_{\mathrm{GD}}^{(\eta)} \circ f_{\text{enc}}$ be an $n$-qubit quantum mechanism where $f_{\mathrm{GD}}^{(\eta)}$ is the GD channel parameterized by a dephasing parameter $\eta \in [0,1]$, and $f_{\mathrm{enc}}$ satisfies Assumption 1 (equatorial–state assumption). Then the composed mechanism $Q^{(\eta)}\circ A$ satisfies $(\varepsilon',\delta')$-DP with
> $
>     \varepsilon' = \varepsilon,  \quad
>     \delta' = |1-2\eta|\cdot \delta.
> $
>
> These two results demonstrate that our privacy-amplification framework is not restricted to the depolarizing channel, but extends to two of the most dominant noise processes in practical quantum hardware.

---

> ### Author Response · Authors · 2025-11-22
>
> `[Q1]`. In Figure 3, in the $\varepsilon' = 1$
>  interval, the classical model ResNet-9 performs better than HYPER-Q (QML). The benefits of HYPER-Q appear only under very strict privacy constraints $(\varepsilon' \leq 0.5)$, which needs to be clarified why.
>
> `[A2]`. Before providing such discussion, we would like to clarify that the behavior observed in our experiments aligns with well-established findings in the quantum machine learning literature. In non-DP settings, hybrid quantum models have not demonstrated an accuracy advantage over strong classical CNNs (such as ResNet-9) on image datasets such as MNIST or Fashion-MNIST ([1], [2]). This limitation is generally attributed to current quantum challenges including limited qubit counts, circuit depth constraints, and scalability issues ([3]).
>
> Under DP settings, classical CNNs are highly sensitive to noise, especially with small $\varepsilon'$. Naturally, when the privacy budget becomes more permissive ($\varepsilon' = 1$), the amount of classical noise is much smaller and ResNet-9 regains its usual advantage. That explains for the reviewer's observation on ResNet-9's performance.
>
> We also would like to clarify that our target is not designing a new hybrid quantum that outperforms classical CNNs in image classification tasks. Instead, our objective is to develop a principled framework that leverages quantum noise inside a hybrid model to reduce the amount of classical Gaussian noise required for DP. The comparison with classical CNNs demonstrates that in strict privacy regimes (small $\varepsilon'$), classical CNNs become highly unstable due to the large noise they must inject, whereas HYPER-Q remains substantially more stable.
>
> ---
> ## References
> [1]. Bowles, Joseph et al. “Better than classical? The subtle art of benchmarking quantum machine learning models.” ArXiv abs/2403.07059 (2024).
>
> [2]. Long, C., Huang, M., Ye, X., Futamura, Y., \& Sakurai, T. (2025). Hybrid quantum-classical-quantum convolutional neural networks. Scientific Reports, 15(1), 31780.
>
> [3]. McClean, Jarrod R. et al. Barren plateaus in quantum neural network training landscapes. Nature Communications 9 (2018).

---

### Official Review · Reviewer_X27r · 2025-11-01

**Soundness:** 2
**Presentation:** 3
**Contribution:** 2
**Rating:** 4
**Confidence:** 4

**Summary:**

This paper investigates privacy guarantees for hybrid quantum-classical machine learning models. The authors propose HYPER-Q, a mechanism that composes a classical (ϵ,δ)-DP mechanism (e.g., input perturbation) with the intrinsic depolarizing noise (η) of a quantum circuit, which acts as a post-processing step. The paper provides a theoretical analysis of this composition, arguing that the quantum noise can act as a privacy amplifier. Theorem 1 analyzes the reduction in the failure probability δ, while Theorem 2 explores conditions for amplifying both ϵ and δ. A utility bound is also derived (Theorem 3). Empirical results on 5-qubit simulations suggest that for a fixed privacy budget, this hybrid approach can improve adversarial robustness over classical-only noise.

**Strengths:**

1.	The paper addresses an interesting and relevant problem: how to formally account for privacy in hybrid quantum-classical models. This is a valid direction for the QML community.
2.	The formal analysis of composing classical DP with quantum post-processing (Theorems 1 & 2) is a good theoretical starting point for this line of inquiry.
3.	The core idea of leveraging intrinsic quantum noise as a privacy-enhancing feature, rather than just a bug, is novel and worth exploring.

**Weaknesses:**

1.	The central, significant weakness is that the entire theoretical framework (Theorems 1 & 2, proofs) is valid only for the depolarizing channel. This is a highly simplified noise model. Real quantum hardware is dominated by other, more complex noise channels (e.g., amplitude damping, phase-flip, crosstalk) for which these proofs do not hold. Therefore, the paper's claims about providing privacy guarantees for "intrinsic quantum noise" are not general and may not apply to any practical quantum device.
2.	All experiments are conducted on a 5-qubit simulator. This is a "toy model" scale. It is well-known that QML model performance and trainability are highly dependent on scale. There is no evidence provided that the utility improvements (e.g., in Figure 2) will hold for QML models with a practical number of qubits (e.g., 50-100+). The claims of "practical viability" are severely undermined by this lack of scaling analysis.
3.	The proposed mechanism requires setting the quantum noise level η. The experiments show this choice is critical: η=0.1 works well, but η=0.3 leads to poor performance. The paper provides no principled method, algorithm, or even a strong heuristic for selecting an optimal η. This is a major methodological gap. A user cannot apply HYPER-Q without an expensive, brute-force grid search, making the method impractical.
4.	The theoretical utility bound (Theorem 3) is difficult to reconcile with the empirical results. The bound includes terms that explicitly increase the error as η increases, suggesting quantum noise is always detrimental to utility. This is in direct contradiction to the main empirical claim, which is that adding η=0.1 improves utility (accuracy) over the η=0 case (pure classical noise). This discrepancy is not addressed, raising questions about the tightness or practical relevance of the utility proof.

**Questions:**

1.	Given that all results are on 5-qubit simulations, what justification do you have that these findings, particularly the utility gains over classical models like ResNet-9, will scale to QML models of practical size (e.g., >50 qubits)?
2.	How do you reconcile the utility bound in Theorem 3, which implies utility worsens with η, with your empirical claim that utility improves when setting η =0.1 (compared to the η=0 baseline)? Does this not suggest the bound is too loose to be practically informative?
3.	Since your entire analysis is specific to the depolarizing channel, can you comment on the validity of your framework for more realistic noise models like amplitude and phase damping? Is it not the case that the claims of "guaranteeing privacy" are limited to a theoretical, idealized noise model?
4.	The paper shows η=0.1 is good and η=0.3 is bad. How should a practitioner choose η for a new dataset or model without resorting to a full hyperparameter sweep, which would be computationally expensive? Is there a theoretical principle to guide this trade-off?

---

> ### Author Response · Authors · 2025-11-22
>
> We sincerely thank the reviewer for the thoughtful and constructive evaluation of our work. We appreciate the reviewer’s recognition of our key contributions: the importance of studying privacy guarantees in hybrid models, the value of our theoretical analysis in Theorems 1 and 2 for understanding classical–quantum DP composition, and the novelty of using intrinsic quantum noise as a privacy-enhancing resource rather than just a hardware limitation. The reviewer’s acknowledgment that the work addresses an important and timely QML problem is also encouraging to us.
>
> Below, we address all raised weaknesses (`W`) and questions (`Q`) in detail. We have also submitted our revised manuscript, with all new material highlighted in yellow.

---

> ### Author Response · Authors · 2025-11-22
>
> `[W1 + Q3]`. Since your analysis is specific to the depolarizing channel, can you comment on the validity of your framework for more realistic noise models like amplitude and phase damping? Is it not the case that the claims of "guaranteeing privacy" are limited to a theoretical, idealized noise model?
>
> `[A1]`. We thank the reviewer for highlighting this important point. While our main theorems are stated for the depolarizing channel, the central insight of our framework is more general. **In particular, the central insight is whenever a quantum noise channel induces a non-trivial contraction of the quantum hockey-stick divergence, it will inherently lead to privacy amplification**. In the revised manuscript, we have added **two new theoretical results** that demonstrate how our framework applies to realistic hardware noise channels in Section 4.2.3 with the detailed proofs in Appendix B.4. Two new theoretical results can be summarized as follows:
>
> **(1) Generalized Amplitude Damping (GAD) Channel.** We introduce a **new lemma (Lemma 4.1)** showing that the generalized amplitude damping channel, which is one of the most widely used models for energy relaxation in superconducting and trapped-ion hardware, contracts the quantum hockey-stick divergence by an explicit state-independent factor. The contraction coefficient leads directly to improved $(\varepsilon',\delta')$ stated in Theorem 1.1 as follows:
>
> **Theorem 1.1 [Amplification Under Generalized Amplitude Damping Noise].**
> Let $A: \mathbb{X} \rightarrow \mathcal{P}(\mathbb{Y})$ be a classical mechanism satisfying $(\varepsilon, \delta)$-DP where $A = f_{\text{par}} \circ f_{\text{cdp}}$, and let $Q^{(p,\eta)}: \mathbb{Y} \rightarrow \mathcal{P}(\mathbb{Z})$ be a quantum mechanism in $d$-dimensional Hilbert space defined as $Q^{(p,\eta)} = f_{\text{mea}} \circ f_{\mathrm{GAD}}^{(p,\eta)} \circ f_{\text{enc}}$ where $f_{\mathrm{GAD}}^{(p,\eta)}$ is the GAD channel parameterized by a damping strength $\eta \in [0,1]$ and an excitation probability $p \in [0,1]$. Then, the composed mechanism $Q^{(p,\eta)} \circ A$ satisfies $(\varepsilon', \delta')$-DP, where $\varepsilon' = \varepsilon, \quad \delta' = (2\sqrt{\eta} - \eta) \delta.$
>
> **(2) Generalized Dephasing (GD) Channel.** We further provide a theorem establishing amplification under the generalized dephasing channel. Under a practical encoder assumption (e.g., equatorial encodings commonly used in QML), the channel yields a closed-form contraction bound for the quantum hockey-stick divergence (**new Lemma 4.2**). This again produces strictly improved privacy parameters, stated in Theorem 1.2.
>
> **Theorem 1.2 [Amplification Under Generalized Dephasing Noise].**
> Let $A: \mathbb{X}\to\mathcal{P}(\mathbb{Y})$ be a classical mechanism satisfying $(\varepsilon,\delta)$-DP, and let $Q^{(\eta)} := f_{\text{mea}} \circ f_{\mathrm{GD}}^{(\eta)} \circ f_{\text{enc}}$ be an $n$-qubit quantum mechanism where $f_{\mathrm{GD}}^{(\eta)}$ is the GD channel parameterized by a dephasing parameter $\eta \in [0,1]$, and $f_{\mathrm{enc}}$ satisfies Assumption 1 (equatorial–state assumption). Then the composed mechanism $Q^{(\eta)}\circ A$ satisfies $(\varepsilon',\delta')$-DP with
> $
>     \varepsilon' = \varepsilon,  \quad
>     \delta' = |1-2\eta|\cdot \delta.
> $
>
> These two results demonstrate that our privacy-amplification framework is not restricted to the depolarizing channel, but extends to two of the most dominant noise processes in practical quantum hardware.

---

> ### Author Response · Authors · 2025-11-22
>
> `[W2 + Q1]`. All experiments are conducted on a 5-qubit simulator which is a "toy model" scale. Given that all results are on 5-qubit simulations, what justification do you have that these findings, particularly the utility gains over classical models like ResNet-9, will scale to QML models of practical size (e.g., $>$ 50 qubits)? The lack of scaling analysis makes it difficult to assess the "practical viability" of the approach.
>
> `[A2]`. We would like to clarify that the scalability challenge raised here is fundamentally a limitation of current QML training itself, rather than of our privacy mechanism. Large-scale (50–100 qubit) noisy quantum training remains an open problem across the field. Many state-of-the-art QML ([1,2,3,4]) and QDP ([5,6]) works benchmark models on 2–12 qubits due to simulator and hardware constraints.
>
> To strengthen our experimental assessment, we have added **a new experiment** using 10-qubit QML models on the more complex CIFAR-10 dataset which includes $32\times32\times3$ RGB images compared to $28\times28$ grayscale images in MNIST. The new results show that applying HYPER-Q to 10-qubit models maintains competitive utility against classical baselines even on this more complex input space. Thus, our use of 5-qubit and 10-qubit models is fully aligned with the experimental scales adopted in prior QML literature, and our additional results further demonstrate that the benefits of HYPER-Q extend beyond the smallest toy models. For **new experimental results**, we refer the reviewer to Figure 21, Appendix F.6 in our revised manuscript or this anonymous link: https://imgur.com/a/V37S1KQ.
>
> Regarding the claim on "practical viability", we clarify that our contribution does not depend on demonstrating QML superiority at 50+ qubits (which is an open challenge for the entire field), but on demonstrating that HYPER-Q itself scales. For better illustration, we have added **a new experimental result** showing that, when depolarizing noise is incorporated, the average percentage reduction in required classical noise does not vanish as the qubit count increases. It implies that the scalability of HYPER-Q naturally follows the scalability of QML models themselves. For **new experimental results**, we refer the reviewer to Figure 16, Appendix F.3 in our revised manuscript or this corresponding anonymous link: https://imgur.com/a/CMcltH7.
>
> ---
> ## References
>
> [1]. Havlíček, V., Córcoles, A. D., Temme, K., Harrow, A. W., Kandala, A., Chow, J. M., \& Gambetta, J. M. (2019). Supervised learning with quantum-enhanced feature spaces. Nature, 567(7747), 209–212.
>
> [2]. Melo, A., Earnest-Noble, N., \& Tacchino, F. (2023). Pulse-efficient quantum machine learning. Quantum, 7, 1130.
>
> [3]. Long, C., Huang, M., Ye, X., Futamura, Y., \& Sakurai, T. (2025). Hybrid quantum-classical-quantum convolutional neural networks. Scientific Reports, 15(1), 31780.
>
> [4]. Hur, T., Kim, L., \& Park, D. K. (2022). Quantum convolutional neural network for classical data classification. Quantum Machine Intelligence, 4(1), 3.
>
> [5]. Du, Y., Hsieh, M.-H., Liu, T., Tao, D., \& Liu, N. (2021). Quantum noise protects quantum classifiers against adversaries. Phys. Rev. Res., 3(2), 023153.
>
> [6]. Watkins, W. M., Chen, S. Y.-C., \& Yoo, S. (2023). Quantum machine learning with differential privacy. Scientific Reports, 13(1), 2453.

---

> ### Author Response · Authors · 2025-11-22
>
> `[W3 + Q4]`.  How should a practitioner choose $\eta$ for a new dataset or model without resorting to a full hyperparameter sweep, which would be computationally expensive? Is there a theoretical principle to guide this trade-off?
>
> `[A3]`. We have added **a new experiment** that explicitly study this dependency across multiple privacy budgets and attack bounds with $\eta \in$ {0.05, 0.1, ..., 0.4}. These results show clear and consistent patterns. That is for any fixed privacy budget $\varepsilon'$, the accuracy curve exhibits a smooth unimodal shape with a clear peak, and the location of this peak (the highest $\eta$) shifts with $\varepsilon'$. Specifically, stricter privacy regimes (smaller $\varepsilon'$) favor slightly larger $\eta$, while more permissive regimes (larger $\varepsilon'$) peak at smaller $\eta$. In addition, across all settings, the peak occurs within the narrow range from $0.05$ to $0.15$. These patterns can provide a simple and reliable strategy for choosing a good value range of $\eta$ without brute-forcing. For **new experimental results**, we refer the reviewer to Figures 18,19, and 20, Appendix F.5 in our revised manuscript or in this anonymous link: https://imgur.com/a/8LCj2dN.

---

> ### Author Response · Authors · 2025-11-22
>
> `[W4 + Q2]`. How do you reconcile the utility bound in Theorem 3, which implies utility worsens with $\eta$, with your empirical claim that utility improves when setting $\eta =0.1$ (compared to the $\eta=0$ baseline)? Does this not suggest the bound is too loose to be practically informative?
>
> `[A4]`. First, we'd like to clarify why the utility bound in Theorem 3 and the empirical observation that accuracy improves at $\eta =0.1$ (compared to $\eta =0$) are not contradictory. In our experiments, the final privacy budget $(\varepsilon',\delta')$ is fixed. Under this fixed budget, decreasing the depolarizing noise level $\eta$ requires increasing the amount of classical DP noise $\sigma$, and vice versa, as determined by Theorem 1. On the other hand, Theorem 3 provides a worst-case bound which isolates the effect of $\eta$ by keeping all other components fixed (including $\sigma$). Thus, the monotonic dependence on $\eta$ in Theorem 3 does not imply that empirical accuracy must decrease with $\eta$ in the experimental setting where $\eta$ and $\sigma$ vary jointly.
>
>  Second, we'd like to explain the practical relevance of the utility bound in Theorem 3 and provide a tightness analysis. Theorem 3 is a worst-case stability guarantee. In particular, it upper-bounds the maximum deviation between the clean and noisy mechanisms for any input and any measurement outcome. Similar to classical DP utility bounds (e.g., in DP-SGD [1]), it is not intended to predict typical accuracy but to certify that the mechanism remains stable under jointly applied classical and quantum noise. To assess tightness, we added **a new experiment** that measures the ratio between the actual utility error and the theoretical utility bound across a grid of $(\eta, \sigma)$ values. We observe that in low-noise settings, the ratios reach significant magnitudes (e.g., $0.923$), confirming that the bound effectively captures the worst-case error. Further details and explanations are provided in Appendix F.4. For **new experimental results**, we refer the reviewer to Figure 17, Appendix F.4 in our revised manuscript or in this anonymous link: https://imgur.com/a/1CSfXG1.
>
> ---
> ## Reference
> [1]. Abadi, M., Chu, A., Goodfellow, I., McMahan, H. B., Mironov, I., Talwar, K., & Zhang, L. (2016). Deep learning with differential privacy. Proceedings of the 2016 ACM SIGSAC Conference on Computer and Communications Security, 308–318.

---

> > ### Comment · Reviewer_X27r · 2025-11-26
> >
> > Thank you for taking my comments into account. For the moment, I am keeping my score.

---

### Official Review · Reviewer_eVCt · 2025-11-03

**Soundness:** 3
**Presentation:** 3
**Contribution:** 2
**Rating:** 6
**Confidence:** 3

**Summary:**

The paper proposes HYPER-Q, a hybrid privacy mechanism that composes classical input noise with intrinsic quantum noise in a hybrid QNN pipeline. and proves privacy amplification results (two theorems) and a utility bound (one theorem). It claims and demonstrates empirical improvements in certified and empirical adversarial robustness on MNIST / FashionMNIST / USPS under fixed end-to-end privacy budgets.

**Strengths:**

1. The original idea comes from the idea of using quantum post-processing to amplify the privacy.

2. The author proposed the detailed theoretical results. Theorems 1-3 show rigorous theoretical guarantees.

3. The experimental pipeline is clear. Uses standard datasets (MNIST/FashionMNIST/USPS) and compares to sensible baselines (Analytic Gaussian). Implementation details and compute resources are partially documented in the appendix.

**Weaknesses:**

While the proofs are provided in Appendix B, I want to flag spots where hidden assumptions could weaken results:

1. Corollaries 3 and 4 depend on POVM trace uniformity. Real measurements (projective or noisy POVMs) may violate these assumptions. This paper partially addresses an “optimal measurement” case, but it should discuss robustness to measurement mismatch.  Otherwise, the advantage the paper claimed is not so practical.

2. Many bounds in the theorems (such as Thm 1) seem to depend on the dimension $d$ of the encoded Hilbert space. In high-dimensional encodings, $d$ grows $2^n$. In Thm 1, the $\eta(1−e^\varepsilon)/d$ term shrinks. And obviously, to achieve the claimed result, there is a requirement for dimension $d$. It is worth discussing scaling and whether the amplification remains meaningful for small $\eta$ in practice.

3. The classical noise is an input perturbation only. In classical DP for ML, many strong results come from gradient- or parameter-level DP (e.g., DPSGD). The authors should justify (or experimentally compare) input perturbation vs. other DP placements and explain how HYPER-Q composes with gradient DP if used in training.

**Questions:**

As I mentioned before, in quantum machine learning, the dimension of the encoded Hilbert space is quite crucial.
1. Can authors discuss the influence of $d$ on the desired result?
2. It will be helpful to indicate how the parameters $d$ are configured in the experiment. I don't find the setting up of dimension $d$.

And there are some typos.

1. Comma missing. line 122 - 123. Line 339 - 342 and et.al.
2. Line 132: One column is missing the divergence expression.
3. Line 761-765: Where is the reference to the inequality?
4. Line 267-268: What is the statement of the post-processing theorem?

---

> ### Author Response · Authors · 2025-11-22
>
> We thank the reviewer for the thoughtful evaluation of our paper. We appreciate the reviewer’s recognition that HYPER-Q introduces a novel hybrid privacy mechanism that leverages quantum post-processing for privacy amplification; that our theoretical contributions (Theorems 1-3) provide rigorous guarantees; and that our empirical pipeline is clear and well-grounded, using standard datasets and sensible baselines. We also appreciate the positive assessments on soundness and presentation.
>
> Below, we address all raised weaknesses (`W`) and questions (`Q`) in detail. We have also submitted our revised manuscript, with all new material highlighted in yellow.

---

> ### Author Response · Authors · 2025-11-22
>
> `⁠ [W1]:⁠` Corollaries 3 and 4 depend on POVM trace uniformity. Real measurements (projective or noisy POVMs) may violate these assumptions. This paper partially addresses an “optimal measurement” case, but it should discuss robustness to measurement mismatch.
>
> `⁠ [A1]:⁠` We respectfully argue that our framework is robust to realistic measurement mismatch and that the “optimal” case in Corollaries 3 and 4 is not restrictive for practical QML.
>
> First of all, we clarify that the optimal measurement assumed in Corollaries 3 and 4 is the  **standard projective measurement** [1]. Each projective measurement outcome is represented by a rank-1 projector $E_k = \ket{k}\bra{k}$ which satisfies:
>     $$\operatorname{Tr}(E_k) = 1 \forall k$$
> Therefore, it has uniform trace across all POVM elements.
>
> Second, our analysis framework naturally extends to **noisy measurement**. Specifically, any noisy measurement can be expressed as a noise channel followed by a perfect projective measurement, i.e.,
>     $$E_k^{\mathrm{noisy}} = \Lambda^\dagger(\Pi_k)$$
> where $\Lambda$ is the readout-noise channel and $\Pi_k$ are the ideal projective POVM elements. In this form, the noisy measurement simply becomes an additional post-processing layer in the chain $f_{\mathrm{mea}} \circ \Lambda$, which cannot weaken differential privacy guarantees. Therefore, measurement mismatch does not affect the correctness or applicability of our privacy analysis.
>
> Finally, we emphasize that the optimal measurement condition in our corollaries 3 and 4 is **practical**. Computational-basis projective measurements, which satisfy the required equal-trace property, are the default readout method in foundational QML literature ([1], [2], [3]) and standard software frameworks like PennyLane ([4]).
>
> ---
> ## References
>
> [1]. Nielsen, M. A., \& Chuang, I. L. (2010). Quantum Computation and Quantum Information: 10th Anniversary Edition. Cambridge University Press.
>
> [2]. Havlíček, V., Córcoles, A. D., Temme, K., Harrow, A. W., Kandala, A., Chow, J. M., \& Gambetta, J. M. (2019). Supervised learning with quantum-enhanced feature spaces. Nature, 567(7747), 209–212.
>
> [3]. Du, Y., Hsieh, M.-H., Liu, T., Tao, D., \& Liu, N. (2021). Quantum noise protects quantum classifiers against adversaries. Phys. Rev. Res., 3(2), 023153.
>
> [4]. Bergholm, V., Izaac, J., Schuld, M., Gogolin, C., Ahmed, S., Ajith, V., Alam, M. S., Alonso-Linaje, G., AkashNarayanan, B., Asadi, A., Arrazola, J. M., Azad, U., Banning, S., Blank, C., Bromley, T. R., Cordier, B. A., Ceroni, J., Delgado, A., Matteo, O. D., … Killoran, N. (2022). PennyLane: Automatic differentiation of hybrid quantum-classical computations.

---

> ### Author Response · Authors · 2025-11-22
>
> `⁠[W2 + Q1]`. To achieve the claimed result in Thm 1, there is a requirement for dimension $d$ of the encoded Hilbert space. It is worth discussing scaling and whether the amplification remains meaningful for small $\eta$ in practice.
>
> `⁠[A2]`. Thank you for this insightful comment. In this rebuttal, we provide more discussion on scaling $d$ and empirical study of small-$\eta$ behavior.
>
> Before providing such discussion, we'd like clarify the theoretical relation between amplification and $d$. In Theorem 1, the amplification is expressed by two terms, $(1-\eta)\delta$ and $\frac{\eta(1-e^\varepsilon)}{d}$. The first term, $(1-\eta)\delta$, is independent of $d$, ensuring that $\delta' < \delta$ for any $\eta > 0$. The second term, $\frac{\eta(1-e^\varepsilon)}{d}$, naturally decreases as $d$ grows, which aligns with the behavior of the depolarizing channel in high dimensions. In the empirical study, we show that a meaningful amplification effect is present regardless of the dimension.
>
> **The empirical study of scaling with $d$**. To illustrate the practical impact, we include **a new experiment** showing how increasing the number of qubits (where $d = 2^n$) affects the reduction in required classical noise under Theorem 1. The experimental result shows that the average percentage reduction in required classical noise as a function of the number of qubits $n$, where the Hilbert-space dimension scales as $d = 2^n$. Each curve corresponds to a depolarizing-noise strength $\eta \in ${0.05, 0.1, 0.2, 0.3, 0.4}. For larger $n$, the curves gradually flatten, consistent with the theoretical $1/d$ dependence in Theorem 1. In addition, higher $\eta$ values lead to consistently stronger amplification across all qubit counts. For **new experimental results**, we refer the reviewer to Figure 16, Appendix F.3 in our revised manuscript or this corresponding anonymous link: https://imgur.com/a/CMcltH7.
>
> **The empirical study of small-$\eta$ behavior**. We further provide **a new set of experiments** examining performance across a range of small quantum-noise values $\eta$. The new experimental results report the average practical accuracy as a function of the depolarizing strength $\eta$ for four privacy budgets $\varepsilon \in$ {0.25, 0.5, 0.75, 1.0} under various attack strength. Across all settings, the curves show a clear non-monotonic pattern that the accuracy initially increases with $\eta$, reaches a peak, and then gradually decreases for larger noise levels. Importantly, the location of this peak depends on $\varepsilon$. Specifically, smaller privacy budgets (e.g., $\varepsilon=0.25$) achieve their highest accuracy at relatively larger $\eta$, whereas larger budgets (e.g., $\varepsilon=1.0$) peak at much smaller $\eta$. Overall, these results confirm that small $\eta$ values are not only sufficient but often optimal in practical regimes. For **new experimental results**, we refer the reviewer to Figures 18, 19, and 20, Appendix F.5 in our revised manuscript or in this anonymous link: https://imgur.com/a/8LCj2dN.

---

> ### Author Response · Authors · 2025-11-22
>
> `[W3]`. The authors should justify (or experimentally compare) input perturbation vs. other DP placements (e.g., DPSGD) and explain how HYPER-Q composes with gradient DP if used in training.
>
> `[A3]`. To address this concern directly, we add **a new set of experiments** to compare against DP-SGD in QML settings. The new experimental results show that across all privacy levels and attack bounds, HYPER-Q consistently outperforms DP-SGD, demonstrating that input-level perturbation provides better utility for QML models. For **new experimental results**, we refer reviewer to Figure 2, Section 5.2 (the overall result) and Figures 4,5,6,7,8,9, Appendix F.1 (the detailed breakdown across all attacks and datasets). Those results can also be found in this anonymous link: https://imgur.com/a/6N9lpKS.
>
> Regarding how HYPER-Q composes with gradient-based DP methods such as DP-SGD, we note that HYPER-Q and DP-SGD are two separate DP mechanisms applied to the same underlying dataset. Therefore, by the standard differential privacy composition theorem, applying HYPER-Q, which is $(\varepsilon_I,\delta_I)$-DP, and DP-SGD, which is $(\varepsilon_G,\delta_G)$-DP, during training yields an overall guarantee of $(\varepsilon_I+\varepsilon_H,\delta_I+\delta_G)$-DP with respect to the original data. Thus, HYPER-Q can be seamlessly combined with gradient-level DP if desired, with total privacy cost obtained via standard composition.

---

> ### Author Response · Authors · 2025-11-22
>
> `[Q2]`. It will be helpful to indicate how the parameters $d$ are configured in the experiment.
>
> `[A4]`. The Hilbert-space dimension is determined by the number of qubits used in the quantum model, i.e. $d= 2^n$. In our experiments on MNIST, Fashion-MNIST, and USPS, we use $n=5$ qubits. We have also added **a new experiment** on CIFAR-10, we use $n=10$.

---

> ### Author Response · Authors · 2025-11-22
>
> `Typos and Minor Issues`
> 1. Comma missing. line 122 - 123. Line 339 - 342 and et.al.
>
> 2. Line 132: One column is missing the divergence expression.
>
> 3. Line 761-765: Where is the reference to the inequality?
>
> 4. Line 267-268: What is the statement of the post-processing theorem?
>
> `[A5]`.
> 1. We have fixed "comma missing" typos.
>
> 2. We have fixed this expression typo, $D_{e^\varepsilon}(\mathcal{M}(D_1) | \mathcal{M}(D_2))  \rightarrow D_{e^\varepsilon}(\mathcal{M}(D_1) | | \mathcal{M}(D_2))$.
>
> 3. We have added reference.
>
> 4. The post-processing theorem states that: If a mechanism $M$ satisfies $(\varepsilon,\delta)$-DP and $g$ is any (possibly randomized) mapping, then $g \circ M$ also satisfies $(\varepsilon,\delta)$-DP [1].
> ---
> ## References
> [1]. Cynthia Dwork and Aaron Roth. 2014. The Algorithmic Foundations of Differential Privacy. Found. Trends Theor. Comput. Sci. 9, 3–4 (Aug 2014), 211–407.

---

> > ### Comment · Reviewer_eVCt · 2025-11-27
> >
> > Thank you for resolving all my issues. The newly added experiments are convincing. I will keep my score.

---

### Author Response · Authors · 2025-12-02

Dear Area Chair and Reviewers,

We sincerely appreciate your time and effort in reviewing our manuscript. Below, we provide a summary of our key contributions and how every major concern have been addressed in our revision.
___
## Contributions
The reviewers reached a strong consensus on the novelty, timeliness, and theoretical rigor of our work. Here, we highlight key contributions:
- __Novelty__: Reviewers consistently recognized the novelty of our work. Reviewer BQou described HYPER-Q as the "first of its kind" to analyze privacy in hybrid models. Reviewers X27r and 2RVG commended the conceptual novelty of treating quantum noise as a "privacy-enhancing feature" (reviewer X27r) and a "useful resource" (reviewer 2RVG).
- __Timeliness and Relevance__: There is agreement that the paper addresses a "very important and timely problem" (reviewer 2RVG) for the QML community. Reviewer X27r emphasized that establishing how to formally account for privacy in hybrid models is a "valid direction" for the field. Reviewer BQou also complimented that our work "lights a torch towards a new direction of quantum noise-based differential privacy."
- __Rigorous Theories__: The reviewers praised the "high theoretical completion" (reviewer 2RVG) of the work. They highlighted that Theorems 1–3 provide "rigorous mathematical proofs" (reviewer BQou) and "rigorous theoretical guarantees" (reviewer eVCt). Reviewer BQou further appreciated the insight on POVM trace balancing as a concrete design rule for optimal amplification.
- __Solid Validation__: Reviewer eVCt found experiments "clear" and "well-grounded" with "sensible baselines."
___
## Responses to Concerns
In response to the reviewers’ concerns, we have significantly strengthened both the theoretical generality and the empirical robustness of our work. Below is a summary of key improvements.
- __Concern 1: Generalization to Realistic Asymmetric Quantum Noise__. Reviewers X27r, 2RVG, and BQou challenged us to generalize our theoretical framework beyond the symmetric depolarizing channel. In the revision, we address this directly by providing new theoretical results showing that our framework __extends to realistic, asymmetric quantum noise found on hardware__. Notably, we have added the following:
    - __New Theoretical Results__: We introduce __new Theorem 1.1__ and __Theorem 1.2__ (Section 4.2.3 and Appendix B.4), which establish explicit privacy-amplification bounds for two asymmetric noise channels: Generalized Amplitude Damping (GAD) and Generalized Dephasing (GD). These results show that the core mechanism underlying our work __persists beyond the depolarizing model and naturally extends to realistic noise.__
    - __New Key Insight__: Our extended analysis (Appendix B.4) yields a general principle: __any quantum channel that induces a non-trivial contraction of the quantum hockey-stick divergence inherently provides privacy amplification__. This shows that our framework is not tied to a particular noise model but instead captures a broad and physically meaningful class of quantum noise processes.
- __Concern 2: Hyperparameter Robustness__. Reviewers eVCt and X27r questioned how HYPER-Q behaves under different quantum-noise levels $\eta$ and if selecting $\eta$ requires costly tuning. In revision, we added __a new set of $\eta$-sweep experiments__ (Appendix F.5)(https://imgur.com/a/8LCj2dN). The results show a consistent unimodal “sweet spot” in the range $\eta \approx 0.05$–$0.15$, where utility is maximized. This behavior across datasets provides a low-overhead heuristic for selecting $\eta$ in real applications.
- __Concern 3: Scalability to High Dimensions__. Reviewers eVCt and X27r questioned if our privacy gains would persist as the Hilbert space dimension $d$ grows. We added __a new scaling experiment__ (Appendix F.3)(https://imgur.com/a/CMcltH7), which shows that the reduction in required classical noise remains __non-vanishing__ even at larger $d$.
- __Concern 4: Tightness of Utility Bound__. Reviewer X27r questioned if the utility bound in Theorem 3 might be loose. We added __a new tightness evaluation__ (Appendix F.4)(https://imgur.com/a/1CSfXG1), which confirms Theorem 3 effectively captures the worst-case utility error.
- __Concern 5: Comparison with additional baseline and dataset__. Reviewer eVCt requested comparisons against standard gradient-based DP methods while reviewer BQou requested validation on more complex dataset. In revision, we included __two new sets of experiments__:
    - __HYPER-Q vs. DP-SGD__: The new results show that HYPER-Q yielded superior utility-privacy trade-offs (Section 5)(https://imgur.com/a/6N9lpKS).
    - __Generalization to CIFAR-10__: The new results show that QML equipped with HYPER-Q maintained competitive utility on CIFAR-10 (Appendix F.6)(https://imgur.com/a/V37S1KQ).

In conclusion, we have completely addressed every major concern raised by reviewers. Thank you again for your thoughtful feedback.

Best,

Authors

---

### Meta-Review · Area_Chair_7gu9 · 2026-01-05

**Summary:**

This paper proposes  HYPER-Q as a new differentially private method for quantum machine learning. This is a borderline paper in my batch, with two 6s and two 4s in the original ratings. After careful reading of the main paper and the rebuttal, I tend to agree with the review comments given by Reviewer X27r and Reviewer BQou. As a theoretical paper, some theoretical insights are not fully supported by the empirical results. As an empirical paper, the paper falls short in the considered noise model and scalability. Overall, this paper requires significant modifications and another round of full review.

**Reviewer Concerns:**

The major concerns on the theoretical soundness and practicality remain unaddressed.

**Reviewer Scores:**

Unlikely to change, given that most reviewers already had some discussion with the authors.

---

### Decision · Program_Chairs · 2026-01-26

Reject